# Benchmarking DNA foundation models for genomic and genetic tasks

Haonan Feng[1], Lang Wu[2], Bingxin Zhao [3], Chad Huff[4], Jianjun Zhang [5], Jia Wu [6], Lifeng Lin [7], Peng Wei [1] ✉ & Chong Wu [1,8] ✉

The rapid evolution of DNA foundation models promises to revolutionize genomics, yet comprehensive evaluations are lacking. Here, we present a comprehensive, unbiased benchmark of five models (DNABERT-2, Nucleotide Transformer V2, HyenaDNA, Caduceus-Ph, and GROVER) across diverse genomic and genetic tasks including sequence classification, gene expression prediction, variant effect quantification, and topologically associating domain (TAD) region recognition, using zero-shot embeddings. Our analysis reveals that mean token embedding consistently and significantly improves sequence classification performance, outperforming other pooling strategies. Model performance varies among tasks and datasets; while general purpose DNA foundation models showed competitive performance in pathogenic variant identification, they were less effective in predicting gene expression and identifying putative causal QTLs compared to specialized models. Our findings offer a framework for model selection, highlighting the impact of architecture, pre-training data, and embedding strategies on performance in genomic and genetic tasks.

Led by the advances in Natural Language Processing (NLP) in recent years, foundation language models through self-supervised pre-training have been the paradigm of decoding information in sequences. By representing sequences as numerical embeddings, foundation language models can outperform previous methods in many downstream tasks such as sequence classification and sequence generation. As natural language-based foundation models like GPT-4[1], Llama 3[2], and Qwen3[3] have been proven successful, similar ideas have been extended to other domains by interpreting domain-specific languages with unique semantic rules, and examples include foundation models on programming codes, protein sequences and single-cell sequencing[4–7]. With the long-lasting interests in decoding DNA sequences to understand the epigenetic patterns, transcriptional regulations, and disease associations[8,9], DNA foundation language models have also emerged recently including DNABERT-2[10], Nucleotide Transformer[11], HyenaDNA[12], Caduceus-Ph[13], and GROVER[14]. These models are pre-trained on large genomic datasets such as the human reference genome[15], human whole-genome sequencing datasets like 1000 Genomes project datasets[16], and multi-species genome datasets[10,11]. After fine-tuning, they have shown promising results in DNA sequence classification tasks.

A critical aspect of DNA foundation models is the method used to generate sequence embeddings, with sentence-level summary token, mean token embeddings, and maximum pooling being three primary approaches. The comparative efficacy of these embedding methods in DNA sequence analysis remains understudied, despite their potential

[1]Department of Biostatistics, The University of Texas MD Anderson Cancer Center, Houston, TX 77030, USA. [2]Department of Interdisciplinary Oncology and Department of Genetics, LSU-LCMC Health Cancer Center, School of Medicine, Louisiana State University Health Sciences Center, New Orleans, LA 70112, USA. [3]Department of Statistics and Data Science, University of Pennsylvania, Philadelphia, PA 19104, USA. [4]Department of Epidemiology, The University of Texas MD Anderson Cancer Center, Houston, TX 77030, USA. [5]Department of Thoracic/Head and Neck Medical Oncology, The University of Texas MD Anderson Cancer Center, Houston, TX 77030, USA. [6]Department of Imaging Physics, Division of Diagnostic Imaging, The University of Texas MD Anderson Cancer Center, Houston, TX 77030, USA. [7]Department of Epidemiology and Biostatistics, University of Arizona, Tucson, AZ 85724, USA. [8]Institute for Data Science in Oncology, The UT MD Anderson Cancer Center, Houston, TX 77030, USA. ✉e-mail: pwei2@mdanderson.org; cwu18@mdanderson.org

impact on model performance across varying sequence lengths and biological contexts.

With the rapid evolution of DNA foundation models of various architectures and the wide range of genomic analysis tasks to be solved, there is a pressing need for effectively evaluating these models. However, most of the current evaluations on DNA foundation models are biased, as they are conducted after fine-tuning[10–12], which may introduce biases in model performance comparison. For instance, different models may have various levels of overfitting depending on which layers are selected to update during fine-tuning. The use of advanced parameter-efficient fine-tuning methods[17,18] further complicates this issue by introducing additional hyperparameters that could impact model fitting. A recent work leveraged this problem by directly comparing DNA foundation models based on their output embeddings[19], where the weights in all layers were frozen and a trainable convolutional neural network (CNN) was appended to the last layer. While this approach mitigates fine-tuning biases, the study scope was limited to several human genome analysis tasks. There is yet to be a comprehensive benchmark which addresses crucial challenges including model performance across diverse genomic tasks and species, scalability with sequence length, and ability to capture biologically significant features.

In this study, we first provide a comprehensive and unbiased evaluation of five state-of-the-art DNA foundation language models across 57 diverse datasets spanning four major categories: human genome region classification, multi-species genome region classification, human epigenetic trait classification, and multi-species epigenetic trait classification. Our benchmarking assesses models' performance across multiple species, sequence lengths, and genomic tasks including promoter identification, enhancer classification, transcription factor binding site prediction, and epigenetic modification detection. We systematically evaluate three different pooling methods (summary token, mean token, and maximum pooling) for each model, revealing substantial and consistent performance differences. Beyond classification tasks, we extend our evaluation to assess models' capabilities in gene expression prediction, where we evaluate how well zero-shot embeddings can predict tissue-specific expression levels from genomic sequences using GTEx v8 data. We further assess variant effect quantification, examining models' ability to distinguish pathogenic from common variants and identify functional quantitative trait loci (QTLs). Additionally, we investigate topologically associating domain (TAD) recognition to determine whether models inherently learn higher-order chromatin structures. We also conduct novel experiments to understand factors influencing model performance, including a controlled pre-training experiment comparing multi-species versus human-only training data. Our evaluation framework provides actionable insights for model selection, the strengths and limitations of different architectural choices, embedding strategies, and pre-training approaches across diverse genomic applications. All code and datasets are available at https://github.com/ChongWuLab/dna_foundation_benchmark.

## Results

### Sequence classification: pooling methods benchmark

We evaluated the zero-shot embeddings from DNA foundation models on 57 sequence classification datasets, including 52 binary classification datasets and 5 multi-class datasets (Section Methods: Benchmarking Datasets). For each dataset, we first generated zero-shot embeddings for all sequences, then split the samples into training and testing sets, and finally trained a classifier and reported its performance predicting the labels of each sequence in the test set from their zero-shot embeddings (Section Methods: Benchmarking Methods). The detailed workflow of sequence classification benchmarking can be found in Fig. 1.

First, we evaluated three output pooling methods: sentence-level summary-token ([CLS] or [SEP]) embedding, mean token embedding, and maximum pooling. We sought to determine the optimal method for generating sequence-level representations, as the choice of pooling method can significantly alter how genomic information is captured and passed to downstream analyses. The results revealed significant differences in the performance of pooling methods across all DNA foundation models. Using DeLong's test for statistical significance ($p < 0.01$), we systematically assessed which pooling method performed better for each dataset. Our analysis showed that mean token embedding consistently delivered superior performance against others (Supplementary Table 1). Specifically, mean token embedding delivered higher AUROC (AUC) with statistical significance than both summary-token embedding and maximum pooling, in 41 out of the 52 binary sequence classification datasets for DNABERT-2, 42 for NT-v2, 35 for HyenaDNA, 37 for Caduceus-Ph, and 41 for GROVER. On the contrary, maximum pooling or summary-token pooling rarely outperformed the other methods, being optimal in only a few specific datasets.

Figure 2 illustrates the distribution of AUC scores for all models when using different pooling methods. The average increase in AUC when switching from summary token to mean token embedding was 4.0% (interquartile range: 2.0%–5.5%) for DNABERT-2, 6.8% (interquartile range: 3.7%–9.6%) for NT-v2, 8.7% (interquartile range: 4.6%–12.9%) for HyenaDNA, 5.9% (interquartile range: 2.8%–9.1%) for Caduceus-Ph, and 1.4% (interquartile range: 0.7%–1.9%) for GROVER across all binary classification tasks. This consistent enhancement with statistical significance underscores the superiority of mean token embedding.

The improved performance with mean token embedding suggests that, by averaging the embeddings of all non-padding tokens, mean token embedding may provide a more comprehensive representation of the entire DNA sequence as opposed to relying on a single summary token. This finding is particularly relevant for some DNA sequence classification tasks, such as promoter and enhancer identification, where the discriminative features may be distributed throughout the sequence rather than concentrated in a specific region. For example, in the promoter identification task for the GM12878 cell line, mean token embedding improved the AUC from 0.964 to 0.986 for DNABERT-2, a 2.3% increase of statistical significance ($p < 0.01$ by DeLong's test). More strikingly, for the *B. amyloliquefaciens* genome, the improvement was from 0.689 to 0.864 for HyenaDNA, representing a 25.4% increase of statistical significance ($p < 0.01$ by DeLong's test). These examples illustrate how mean token embedding can capture distributed features more effectively across the entire sequence.

Additionally, we observed that the performance differences among the models were reduced when using mean token embedding, implying that this pooling method helps to mitigate the architectural variations across the models. Specifically, the range of average AUC scores across models decreased from 0.708 to 0.799 with summary-token pooling, to 0.795 to 0.822 with mean pooling. This observation further reinforces the value of mean pooling as a more robust approach that can help standardize model evaluation.

With the optimal pooling method established, we also evaluated our choice of the downstream classifier. We selected random forest as our primary model, as it requires minimal hyperparameter tuning and can inherently handle high-dimensional inputs without dimension reduction, in both ways helps avoid the introduction of additional evaluation bias. It also has the capacity to capture the complex, non-linear relationships potentially encoded in genomic sequences. To evaluate our approach, we run random forest, naïve Bayes, and elastic-net logistic regression on all binary sequence classification datasets (Supplementary Fig. 1). The results reconfirmed that mean token embedding was the superior pooling strategy across all foundation

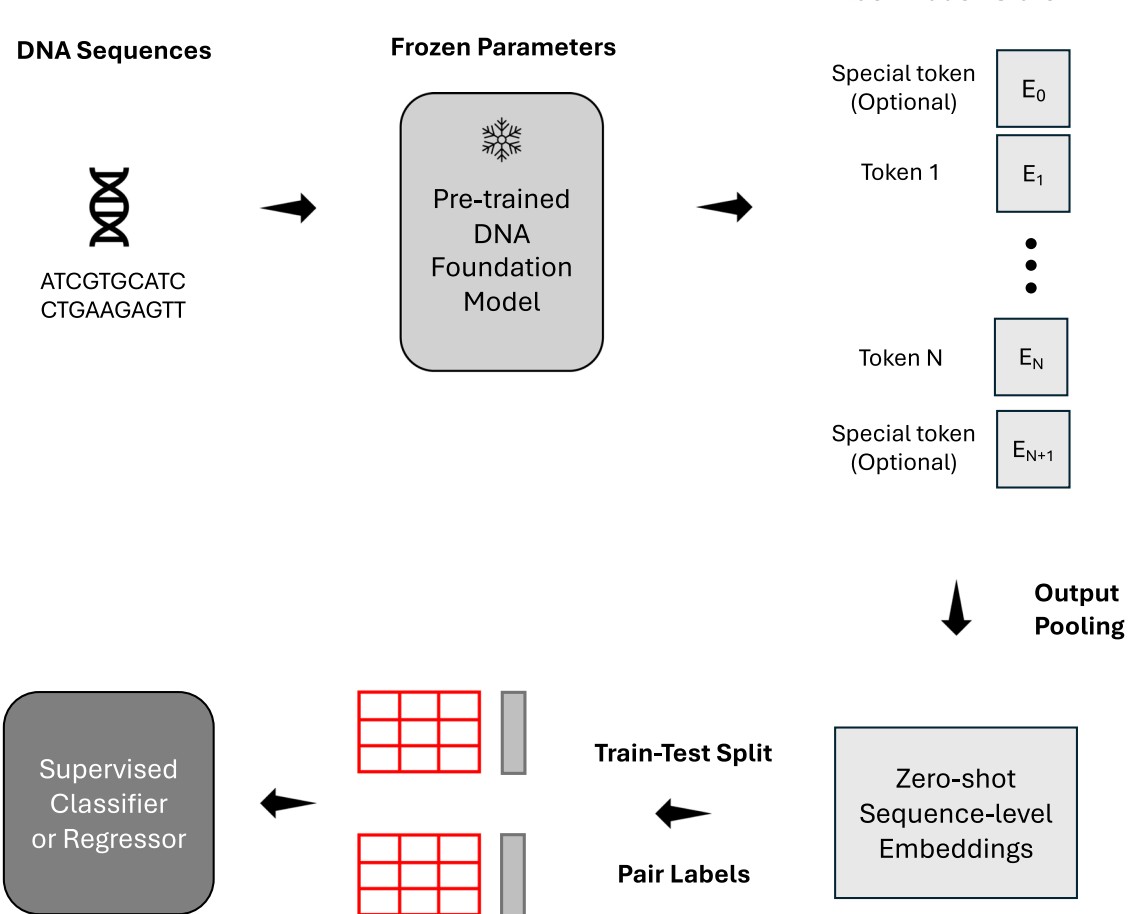

**Fig. 1 | Overview of sequence classification benchmark workflow.** DNA sequences are input into foundation models, generating token embeddings from the final layer. These embeddings undergo output pooling to produce high-dimensional representations of input sequences. A supervised classifier (random forest) is trained on these embeddings using labeled datasets. Model performance is evaluated on a independent test set using multiple metrics, with AUROC as the primary measure.

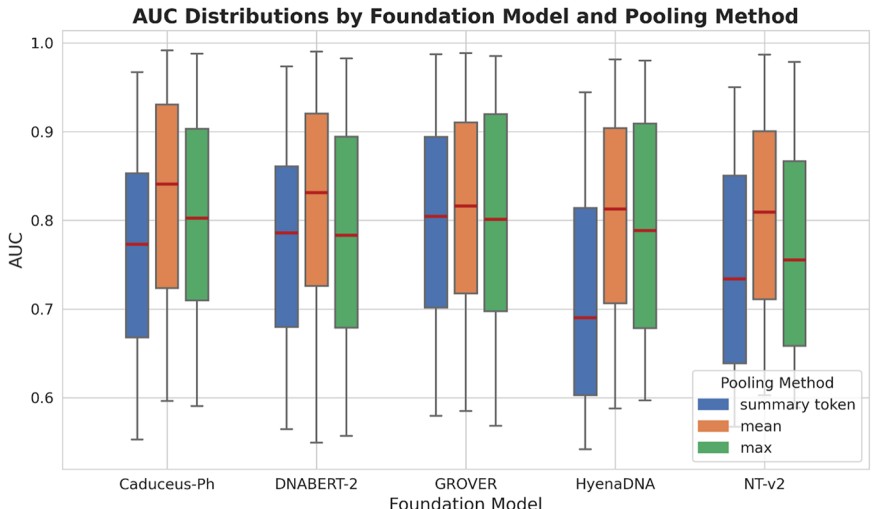

**Fig. 2 | Boxplots comparing the AUC scores distribution on the choice of using mean output pooling, summary-token pooling, or maximum pooling.** We calculate AUC scores from all 52 binary sequence classification datasets included in this study. Boxplots show the median (center line), interquartile range (box = 25th–75th percentiles), and whiskers correspond to 1.5 × interquartile range. minima and maxima correspond to the whisker ends.

**Table 1 | The AUC results for binary sequence classification tasks on human genome**

| Data | DNABERT-2 | NT-v2 | HyenaDNA | Caduceus-Ph | GROVER |
|---|---|---|---|---|---|
| DNase I Hypersensitive | 0.8666 | 0.8524 | 0.8295 | 0.8799 | 0.857 |
| Human TFBS 1 | 0.8382 | 0.8315 | 0.8301 | **0.8796** | **0.8618** |
| Human TFBS 2 | 0.821 | 0.809 | 0.8205 | **0.8687** | **0.8495** |
| Human TFBS 3 | 0.7896 | 0.7974 | 0.7875 | **0.8249** | **0.8158** |
| Human TFBS 4 | 0.726 | 0.7103 | 0.7149 | **0.7725** | **0.763** |
| Human TFBS 5 | 0.9204 | 0.9149 | 0.9159 | **0.9294** | **0.931** |
| Promoter GM12878 | 0.9856 | 0.9835 | 0.976 | **0.9865** | 0.9839 |
| Promoter HUVEC | **0.9903** | 0.987 | 0.9817 | **0.9896** | 0.9885 |
| Promoter Hela-S3 | **0.9886** | 0.9838 | 0.981 | **0.9871** | 0.9857 |
| Promoter NHEK | **0.9501** | 0.9323 | 0.9271 | **0.9567** | **0.9507** |
| Acceptor | **0.8969** | 0.7928 | 0.7946 | **0.8449** | 0.8041 |
| Coding | **0.9438** | 0.9289 | 0.9406 | **0.9735** | **0.9594** |
| Donor | **0.9056** | 0.8198 | 0.8128 | **0.8535** | 0.819 |
| Enhancer | **0.8717** | **0.8674** | 0.8339 | 0.8384 | 0.8554 |
| Enhancer Cohn | **0.8223** | 0.7894 | 0.7754 | **0.821** | **0.8161** |
| Enhancer Ensembl | 0.9369 | **0.9389** | 0.9356 | **0.9431** | 0.9382 |
| Open chromatin region | **0.7253** | 0.7183 | 0.7191 | **0.765** | **0.7455** |
| Promoter All 300 bps | 0.9426 | **0.9445** | 0.9394 | **0.9519** | 0.9402 |
| Promoter All 70 bps | 0.8311 | **0.8527** | 0.832 | **0.8748** | **0.8506** |
| Promoter NonTATA 251 bps | 0.9297 | 0.8905 | 0.928 | **0.9426** | **0.9395** |
| Promoter NonTATA 300 bps | **0.9765** | **0.9758** | 0.9662 | **0.9834** | 0.9728 |
| Promoter NonTATA 70 bps | 0.8531 | **0.8729** | 0.8516 | **0.8961** | **0.8704** |
| Promoter TATA 300 bps | 0.7646 | 0.7791 | **0.8077** | 0.76 | 0.78 |
| Promoter TATA 70 bps | 0.7781 | 0.7947 | 0.7827 | **0.8103** | 0.796 |

The tasks include promoter region identification (across multiple datasets), coding region detection, splice site donor and acceptor identification, enhancer identification (across multiple datasets), transcription factor binding site identification (across multiple datasets), and open chromatin region identification (across multiple datasets). Using mean token pooling method. Bolded: higher than at least two other AUCs, $p < 0.01$. P-values are calculated using one-sided DeLong's test.

models and all classifiers. Within mean pooling configuration, both random forest and elastic-net proved to be the more competitive classifiers. Given its strong performance and well-documented generalization capabilities, we selected random forest as the standard classifier for all analyses in our sequence classification benchmark, providing a reliable estimate of each foundation model's performance on similar tasks.

**Sequence classification: human genome regions**
We evaluated the performance of the five DNA foundation models, DNABERT-2, NT-v2, HyenaDNA, Caduceus-Ph, and GROVER, on a diverse set of human genome sequence region classification tasks. For these tasks, as shown in Table 1, when using mean token pooling, all five models achieved AUC scores above 0.8 on the majority of tasks, indicating their ability to capture meaningful semantic information from human DNA sequences. These results show that zero-shot embeddings generated by these models are sufficiently informative for supervised learning models, even without fine-tuning. Among the five models, Caduceus-Ph exhibited superior overall performance across multiple human genome classification tasks. For promoter identification in cell lines GM12878, HUVEC, Hela-S3 and NHEK, both DNABERT-2 and Caduceus-Ph achieved statistically significant superior performance. DNABERT-2 showed particular strength in splice site prediction, significantly outperforming other models in both donor and acceptor identification tasks with AUCs of 0.906 and 0.897, respectively. For transcription factor binding site (TFBS) prediction, Caduceus-Ph consistently outperformed all other models, demonstrating its ability to capture complex regulatory patterns in the human genome.

When compared to our baseline CNN model trained directly on DNA sequences, the DNA foundation models demonstrated clear

advantages in several key tasks (Supplementary Data 1,2). For example, for GM12878 and HUVEC cell lines promoter identification, DNABERT-2, GROVER, Caduceus-Ph, and NT-v2 all showed significant improvements over the baseline CNN. In enhancer identification tasks. Overall, these four DNA foundation models demonstrated statistically significant performance gains compared to the baseline CNN.

**Sequence classification: multispecies genome regions**
In multispecies genome classification tasks using mean token pooling (Table 2), HyenaDNA demonstrated unexpected advantage in Arabidopsis promoter identification, achieving statistically significant superior performance for both TATA and NonTATA promoters with AUC scores of 0.961 and 0.955 respectively, despite being pre-trained exclusively on human genomes. This suggests HyenaDNA's architecture enables effective transfer of learned representations to plant genomes. For the human versus worm classification task, Caduceus-Ph and GROVER demonstrated the strongest performance (AUCs: 0.992 and 0.984), with both models achieving statistically significant advantages over other models. In bacterial promoter identification, GROVER showed exceptional performance for R. Capsulatus (AUC: 0.715) while no model achieved clear statistical superiority for B. amyloliquefaciens, suggesting challenges in generalizing to more distantly related prokaryotic genomes.

Compared to the baseline CNN model, DNA foundation models generally underperformed for these multispecies tasks (Supplementary Data 1, 2). This pattern was particularly evident for the almost all datasets in Table 2, as most of the times all foundation models were outperformed by the baseline CNN. Such performance gap could be expected, as the baseline CNN is specifically optimized for each dataset with all parameters updated during training, while foundation models rely solely on their pre-training to generate embeddings. In

**Table 2 | The AUC results for binary sequence classification tasks which have multiple species involved, including promoter region prediction (first four rows), human vs worm classification and mouse transcription factor binding site (TFBS) identification**

| Data | DNABERT-2 | NT-v2 | HyenaDNA | Caduceus-Ph | GROVER |
|---|---|---|---|---|---|
| Promoter Arabidopsis NonTATA | 0.9457 | 0.9395 | **0.9547** | 0.9437 | 0.949 |
| Promoter Arabidopsis TATA | 0.951 | 0.95 | **0.9609** | 0.9372 | 0.9486 |
| Promoter B.Amyloliquefaciens | 0.8518 | 0.8225 | 0.8643 | 0.8686 | 0.8617 |
| Promoter R.Capsulatus | 0.6855 | 0.6746 | **0.7116** | 0.67 | **0.7154** |
| Human vs worm | 0.9799 | 0.9785 | 0.9502 | **0.9915** | **0.9843** |
| Mouse TFBS 1 | 0.711 | 0.704 | 0.5899 | 0.6841 | 0.6947 |
| Mouse TFBS 2 | **0.9072** | 0.9005 | 0.8996 | **0.9472** | **0.9093** |
| Mouse TFBS 3 | 0.9308 | 0.9269 | 0.8944 | 0.9351 | 0.9327 |
| Mouse TFBS 4 | **0.7622** | 0.6942 | 0.588 | 0.7047 | 0.6815 |
| Mouse TFBS 5 | 0.6783 | **0.7077** | 0.627 | **0.715** | 0.6822 |

Using mean token pooling method. Bolded: higher than at least two other AUCs, $p < 0.01$. $P$-values are calculated using one-sided DeLong's test.

**Table 3 | The AUC results for each model on 5mC, 6 mA detection in human; epigenetic marks detection in yeast, and 4mC detection in multiple species**

| Data | DNABERT-2 | NT-v2 | HyenaDNA | Caduceus-Ph | GROVER |
|---|---|---|---|---|---|
| Human 5mC | 0.685 | **0.7377** | 0.6843 | **0.783** | **0.7437** |
| Human 6 mA | 0.7351 | **0.7508** | 0.7377 | **0.7731** | **0.7671** |
| Yeast H3 | **0.9137** | 0.8951 | 0.8996 | **0.9285** | 0.9056 |
| Yeast H3K14ac | **0.7597** | **0.7407** | 0.7067 | 0.7297 | 0.7301 |
| Yeast H3K36me3 | **0.7989** | **0.7849** | 0.7395 | **0.7662** | 0.7533 |
| Yeast H3K4me1 | **0.7306** | **0.7115** | 0.6994 | 0.7071 | 0.696 |
| Yeast H3K4me2 | **0.7078** | 0.6847 | 0.6854 | 0.6895 | **0.6939** |
| Yeast H3K4me3 | **0.6813** | 0.6603 | 0.6486 | 0.6595 | 0.668 |
| Yeast H3K79me3 | **0.8565** | 0.8436 | 0.8215 | 0.845 | 0.8427 |
| Yeast H3K9ac | **0.7922** | 0.7687 | 0.7555 | **0.7779** | 0.7692 |
| Yeast H4 | **0.9314** | 0.9104 | 0.8983 | **0.9304** | 0.908 |
| Yeast H4ac | **0.7473** | 0.7263 | 0.6979 | 0.7235 | 0.7175 |
| A.Thaliana 4mC | 0.5994 | **0.6332** | 0.5941 | **0.6146** | 0.6026 |
| C.Elegans 4mC | 0.5985 | **0.6487** | 0.5964 | 0.5964 | **0.6057** |
| D.Melanogaster 4mC | 0.6147 | **0.6519** | 0.6096 | 0.6161 | 0.6167 |
| E.Coli 4mC | 0.5492 | 0.6028 | 0.6105 | **0.6283** | 0.5851 |
| G.Pickeringii 4mC | 0.5958 | 0.6302 | 0.6292 | 0.6348 | 0.6293 |
| G.Subterraneus 4mC | 0.5802 | 0.6145 | 0.609 | 0.6079 | 0.6061 |

Using mean token pooling method. Bolded: higher than at least two other AUCs, $p < 0.01$. $P$-values are calculated using one-sided DeLong's test.

tasks involving multispecies genomes, such discrepancy may have been amplified.

### Sequence classification: human & multispecies epigenetic modification

We further investigated the performance of the foundation models on several epigenetic modification prediction tasks. For human epigenetic modifications, specifically the detection of 5-methylcytosine (5mC) and N6-methyladenosine (6 mA), using mean token pooling, NT-v2, Caduceus-Ph, and GROVER consistently demonstrated strong performance (Table 3). All three models achieved statistically significant superiority for 5mC detection, with AUCs of 0.738 (NT-v2), 0.783 (Caduceus-Ph), and 0.744 (GROVER) respectively. A similar pattern was observed for 6 mA detection, where NT-v2, Caduceus-Ph, and GROVER again outperformed the other models.

On yeast epigenetic mark prediction, among the foundation models, DNABERT-2 showed especially robust performance across the full range of yeast epigenetic marks and performance on the 4 mC

detection tasks across multiple species is detailed in Table 3. As noted in our Methods, the interpretation of results for the eukaryotic 4mC datasets (*A. thaliana, C. elegans, D. melanogaster*) warrants particular consideration due to the nature of their annotations, and these tasks are thus viewed as exploratory for model capabilities on such data. With these considerations, the 4 mC detection tasks (Table 4) showed that NT-v2 achieved notable performance for A. thaliana (AUC: 0.633), C. elegans (AUC: 0.649) and D. melanogaster (AUC: 0.652). Caduceus-Ph demonstrated strong results for the E. coli dataset (AUC: 0.628), where 4mC annotations are more directly supported. For C. elegans, both NT-v2 and GROVER showed significantly better performance than other models.

Compared to the baseline CNN, DNA foundation models generally underperformed on the human and multi-species epigenetic tasks (Supplementary Data 1,2). However, this trend was reversed for yeast epigenetic mark prediction, Multiple DNA foundation models demonstrated a clear and significant advantage over the baseline CNN. In contrast to other epigenetic tasks, for most yeast datasets, several

**Table 4 | Overall performance of DNA foundation models in gene expression prediction using random forest regression**

| Model | Input Sequence Length | Average Prediction Correlation | Average Prediction MSE |
|---|---|---|---|
| DNABERT-2 | 6000 bp | 0.121 | 0.236 |
| NT-v2 | 6000 bp | 0.122 | 0.236 |
| HyenaDNA | 6000 bp | 0.122 | 0.235 |
| Caduceus-Ph | 6000 bp | 0.123 | 0.234 |
| GROVER | 2048 bp | 0.114 | 0.233 |
| HyenaDNA-450K* | 196K bp | 0.137 | 0.226 |
| Caduceus-Ph Long Sequence Input* | 131K bp | 0.127 | 0.227 |
| Enformer* | 196K bp | 0.129 | 0.227 |

Average prediction correlation across all genes and average prediction MSE across all genes are recorded. *Note that for long input sequences, only a subset of human genes are involved in analysis.

foundation models outperformed the task-specific CNN. For instance, in predicting Yeast H3, H3K4me1, and H3K79me3 marks, five and four foundation models respectively showed superior performance. This suggests that the features learned during pre-training are particularly effective for capturing the signals related to yeast histone modifications. Overall, epigenetic prediction proved more challenging for all models, with AUC scores typically 10–15% lower than for tasks distinguishing functional genomic regions.

Lastly, in the rest five classification tasks with more than two classes (Supplementary Table 2), we observed notable performance variations among the foundation models. HyenaDNA demonstrated exceptional performance in the Regulatory Region Type classification task, achieving an accuracy of 0.83, significantly outperforming the rest. Similarly, for Splice Site Type classification using NT's dataset, HyenaDNA achieved the highest accuracy of 0.563. For Covid Variants classification, DNABERT-2 and GROVER take the lead, while for Enhancer Strength prediction, GROVER, DNABERT-2, and Caduceus-Ph all performed comparably with accuracies over 0.7. These results suggest that different model architectures have varying strengths in multi-class discrimination tasks, with HyenaDNA's architecture appearing particularly well-suited for capturing complex patterns necessary for discriminating between multiple regulatory region types.

### Gene expression prediction

We evaluated DNA foundation models and revealed several key findings about their effectiveness in predicting gene expression from genomic sequences. Across all foundation models tested, random forest regression significantly outperformed XGBoost, achieving better correlation and lower mean squared error (MSE) between predicted and actual gene expression values (all $p < 0.001$) (Supplementary Table 3), regardless of whether they used shorter or longer input sequences. Based on these results, we use random forest regression to predict gene expression from DNA foundation model zero-shot embeddings, and our subsequent conclusions are all based on random forest regression outcomes.

On average, the Pearson correlation between predicted and actual gene expression was modest, ranging from 0.114 to 0.123 across models using 6k bp sequences inputs (or corresponding center-cropped sequences for GROVER) (Table 4). This suggests that zero-shot embeddings from DNA foundation models, without fine-tuning, provide limited information for predicting subject-specific tissue-level gene expression. Within the short sequence models, Caduceus-Ph and HyenaDNA exhibited the highest average correlations, significantly

outperforming DNABERT-2 and GROVER. GROVER, which used a shorter 2048 bp input, achieved a slightly lower average correlation than the other short-sequence models. Among long sequence models, Enformer achieved similar performance to Caduceus-Ph. HyenaDNA-450K demonstrated significantly better performance than Enformer ($p < 0.01$). When comparing HyenaDNA and Caduceus-Ph with their longer sequence input counterparts (HyenaDNA-450K and Caduceus-Ph with longer sequence inputs) on the same set of genes, we observed that extending the sequence length significantly improved performance for HyenaDNA ($p < 0.001$), while the improvement for Caduceus-Ph was not statistically significant (Supplementary Table 4).

The distribution of gene prediction correlations showed similar patterns across all models (Fig. 3). While average correlations were modest, certain genes were consistently predicted with substantially higher accuracy, with correlations exceeding 0.4 and reaching up to 0.9. This indicates that while zero-shot embeddings may not capture the regulatory complexity for all genes, they successfully identify and model genes whose expression is tightly controlled by their local sequence.

Analysis of the best-predicted genes revealed a remarkable consistency in model performance, identifying a core set of highly predictable genes regardless of model architecture (Supplementary Data 3). With 6k bp inputs, the gene *CUTALP* was consistently the top-predicted gene across all five short-sequence models, with correlations exceeding 0.89. Other highly predictable genes that consistently ranked among the top 10 across these models included *CPNE1*, *NT5C3B*, *DDX11*, and *PEX6*. For the long sequence models, which analyzed an extended genomic context of up to ~196 K bp, *DDX11* and *HLA-DRB5* consistently achieved the highest correlations across all three models. Other genes showing strong predictability with longer sequences included *DHFR* and *CD151*. Of particular relevance to blood tissue, *CD151* is a tetraspanin family member that plays important roles in platelet activation and aggregation, making it highly relevant to blood function[20]. *DHFR* (dihydrofolate reductase) is essential for DNA synthesis and is targeted by several drugs used to treat blood disorders[21], further highlighting the biological relevance of these predictions.

While our results identify specific genes whose expression patterns appear more predictable from sequence data, the overall performance of these zero-shot embeddings remains modest. Therefore, these findings should be considered preliminary indicators of model capabilities.

### Variant effect quantification

Our analysis of zero-shot embeddings for variant effect prediction reveals that model performance is highly task-dependent, creating a clear distinction between performance on pathogenic variant identification, and putative causal QTL classification. For the task of distinguishing pathogenic from common SNPs, the transformer-based foundation models NT-v2 and Caduceus-Ph surprisingly emerged as the top performers. NT-v2 was particularly dominant, achieving the highest average test AUC of 0.73 and the largest effect size (Cohen's d: 0.88), substantially outperforming all other models, including those trained on functional tracks like Enformer (AUC: 0.69, Cohen's d: 0.73) and Sei (Table 5). This suggests that the pre-training objectives of these specific foundation models are exceptionally well-suited for capturing the subtle, sequence-level patterns that differentiate pathogenic variants.

In contrast, for the putative causal QTL variant effect benchmark to distinguish putative causal QTLs from non-causal variants, models explicitly trained to predict genomic tracks (AlphaGenome[22], Enformer[23], and Sei[24]) held a clear and consistent advantage (Table 6). AlphaGenome was the standout model, achieving the highest AUC and Cohen's d scores across all four QTL types (e.g., AUC = 0.80 for sQTLs, AUC = 0.86 for ipaQTLs). Enformer and Sei also significantly

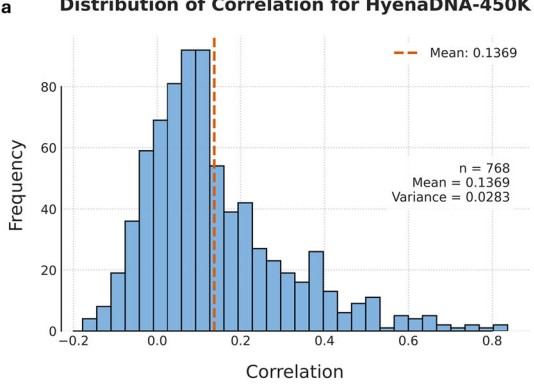

**Fig. 3 | The distribution of prediction correlation between predicted and actual gene expression values.** The histograms showing the prediction correlation on all genes using (**a**). HyenaDNA and (**b**). HyenaDNA-450 K which takes longer input sequences (196 K bps). Both models demonstrate similar distribution patterns with a slight positive skew in the histogram, indicating that while most genes have modest predictability, a subset of genes shows stronger sequence-expression relationships.

**Table 5 | Overall performance of DNA foundation models and other genomic models in the pathogenic versus common variant effect quantification task**

| Model | AUC | Cohen's d |
|---|---|---|
| Sei, hidden states* | 0.6598 | 0.5573 |
| Sei, output tracks* | 0.664 | 0.6046 |
| Enformer, hidden states* | 0.688 | 0.7269 |
| Enformer, output tracks* | 0.6662 | 0.6542 |
| DNABERT-2 | 0.538 | 0.1338 |
| NT-v2 | **0.7319** | **0.8813** |
| HyenaDNA | 0.612 | 0.3952 |
| HyenaDNA-450K, long sequence | 0.6261 | 0.4493 |
| Caduceus-Ph | **0.6959** | **0.7354** |
| Caduceus-Ph, long sequence | 0.6243 | 0.4615 |
| GROVER | 0.6029 | 0.3693 |

All metrics represent the average test AUC and Cohen's d values calculated across three independent test sets, each defined by a distinct group of chromosomes in our nested cross-validation framework. Non-DNA foundation models are annotated with an asterisk (*). Bolded: top 2 highest (absolute) value.

outperformed the general DNA foundation models; for example, in eQTL prediction, Enformer achieved an AUC of 0.77, compared to the best-performing general foundation model, Caduceus-Ph, at 0.65. A notable finding for both Enformer and Sei was that their internal hidden states were often as predictive as their final processed output tracks; for instance, in sQTL prediction, Sei's hidden state representation achieved an AUC of 0.65 compared to 0.63 for its output tracks.

Among the general-purpose foundation models, Caduceus-Ph again demonstrated the most consistent and robust performance, serving as a reliable baseline. The performance of the other foundation models was more erratic. This instability was particularly pronounced on the QTL tasks with the smaller sample sizes (ipaQTL, paQTL and sQTL) where several models, including DNABERT-2, HyenaDNA, Caduceus-Ph with long sequence input, and GROVER, may yield an average AUC slightly below 0.5. This result highlights the challenges of benchmarking on small datasets with strict chromosome-based splits. Our nested cross-validation design makes this variance explicit: the detailed results for each test chromosome group (Supplementary Data 4) reveal that performance on these small datasets is highly sensitive to the choice of holdout chromosomes.

Finally, a direct comparison between the short- and long-sequence versions of HyenaDNA and Caduceus-Ph did not suggest a clear advantage for using longer inputs in this context. For the pathogenic SNP task, the short-sequence version of Caduceus-Ph (AUC: 0.70) clearly outperformed its long-sequence counterpart (AUC: 0.62). This suggests that for a single nucleotide change, a larger genomic context may dilute the local signal when using this embedding subtraction methodology.

**Pre-training experiment**

We investigated the impact of pre-training dataset diversity by re-pretraining HyenaDNA on DNABERT-2's multi-species dataset comprising 135 species across 6 taxonomic categories, and examined their performance on the 57 sequence classification benchmark datasets. We maintained the hyperparameters comparable to the original HyenaDNA-1K checkpoint referring to their descriptions, though our implementation may slightly differ from the original model's training scale in number of training steps and batch size as these were not clearly stated.

The newly pre-trained model, which leverages the diverse multi-species dataset, demonstrated significant performance gains over the original human-genome-trained HyenaDNA-1K on most datasets. It achieved statistically significant improvements in 14 of the 49 evaluated datasets, particularly in areas requiring cross-species generalization and diverse epigenetic pattern recognition (Supplementary Data 5). For instance, in human epigenetic modification tasks, the AUC for 5mC detection increased from 0.707 to 0.749, and in the cross-species human versus worm genome classification, the AUC improved from 0.968 to 0.984. This advantage also extended to multi-species epigenetic datasets such as *C. elegans* 4mC detection and various promoter identification tasks. On the other hand, the original HyenaDNA-1K achieved superior AUC on 3 out of the 49 datasets, related to human enhancer identification, human open chromatin region identification and yeast epigenetic mark prediction. This suggests that specialized human genome pre-training still retain a few benefits from certain tasks.

These findings provide evidence that the architecture of HyenaDNA is fundamentally robust on training datasets, and multi-species pre-training may enhance its generalizability for diverse tasks and cross-species generalization. Our results suggest that multi-species pre-training should be considered a critical design choice. Future DNA foundation models can benefit from exploring optimal species sampling strategies and taxonomic diversity in pre-training.

**Table 6 | Overall performance of DNA foundation models and other genomic models in QTL variant effect quantification tasks**

|  | Model | eQTL | sQTL | paQTL | ipaQTL |
|---|---|---|---|---|---|
| AUC | AlphaGenome, output tracks* | **0.8029** | **0.7147** | **0.7543** | **0.8644** |
|  | Sei, hidden states* | 0.7561 | 0.6534 | 0.6189 | 0.6071 |
|  | Sei, output tracks* | 0.7497 | 0.6276 | 0.6553 | 0.606 |
|  | Enformer, hidden states* | **0.7744** | **0.6662** | **0.6737** | **0.6919** |
|  | Enformer, output tracks* | 0.7699 | 0.6174 | 0.666 | 0.6587 |
|  | DNABERT-2 | 0.5702 | 0.5795 | 0.5066 | 0.4694 |
|  | NT-v2 | 0.6091 | 0.5047 | 0.5251 | 0.6019 |
|  | HyenaDNA | 0.6117 | 0.5531 | 0.4699 | 0.448 |
|  | HyenaDNA-450K, long sequence | 0.6027 | 0.5262 | 0.5521 | 0.5093 |
|  | Caduceus-Ph | 0.6492 | 0.5666 | 0.5082 | 0.5678 |
|  | Caduceus-Ph, long sequence | 0.6265 | 0.5703 | 0.4649 | 0.5203 |
|  | GROVER | 0.5896 | 0.4742 | 0.4494 | 0.4759 |
| Cohen's d | AlphaGenome, output tracks* | **1.287** | **0.7872** | **0.9824** | **1.6347** |
|  | Sei, hidden states* | 1.0335 | 0.553 | 0.4538 | 0.4126 |
|  | Sei, output tracks* | 1.0116 | 0.4936 | 0.5503 | 0.4227 |
|  | Enformer, hidden states* | **1.1102** | **0.6115** | **0.6028** | **0.6576** |
|  | Enformer, output tracks* | 1.1085 | 0.4129 | 0.5457 | 0.5691 |
|  | DNABERT-2 | 0.2371 | 0.2825 | 0.024 | −0.0756 |
|  | NT-v2 | 0.3956 | −0.004 | 0.0658 | 0.3837 |
|  | HyenaDNA | 0.3877 | 0.2018 | −0.0768 | −0.2048 |
|  | HyenaDNA-450K, long sequence | 0.3605 | 0.0757 | 0.2068 | 0.0733 |
|  | Caduceus-Ph | 0.5484 | 0.2278 | 0.0456 | 0.2324 |
|  | Caduceus-Ph, long sequence | 0.4913 | 0.2464 | −0.1151 | 0.075 |
|  | GROVER | 0.319 | −0.0978 | −0.1393 | −0.072 |

All metrics represent the average AUC and Cohen's d values calculated across three independent test sets, each defined by a distinct group of chromosomes in our nested cross-validation framework. Non-DNA foundation models are annotated with an asterisk (*). Bolded: the top two highest (absolute) performances for each task.

## TAD region recognition

A key question for foundation models is whether their internal representations, such as self-attention patterns, learn biologically meaningful, higher-order genomic structures without explicit training. To investigate this, we assessed NT-v2's capability to recognize topologically associating domains (TADs) by analyzing attention patterns in the self-attention mechanism. Unfortunately, such analysis is currently constrained by model architecture; other foundation models evaluated in our study either lack accessible attention mechanisms (HyenaDNA and Caduceus-Ph), accessible attention matrices (DNABERT-2), or have input length limitations that preclude meaningful analysis of large genomic structures like TADs. We processed 1500 TAD-centered sequences and 1500 randomly selected background sequences through NT-v2, extracting and averaging attention matrices across all layers and attention heads. The attention weight at any coordinate $(x, y)$ indicates how strongly the query token at position y attends to the key token at position x during self-attention computation. Therefore, if NT-v2 can recognize TAD structures inherently, we would expect the attention weights to increase in the central region (approximately token positions 300–700), resulting in a vertical band of positive values in the average attention matrix when key tokens fall within the TAD region. But for random background sequences, the average attention matrix would not show any clear pattern. Hence the differences between these two attention matrices will also show a vertical band.

Supplementary Fig. 2 displays the heatmap of the difference between attention matrices for TAD-centered versus background sequences. We observed no distinct patterns in the attention difference matrix. The heatmap shows predominantly uniform values near zero, with only minimal variation along the diagonal where tokens attend to themselves. This suggests that NT-v2's self-attention

mechanism does not recognize TAD boundaries without specific fine-tuning.

Our analysis represents the current extent of attention mechanism interpretability for DNA foundation models associated with chromatin structures, and reveals current DNA foundation models' limitations on interpretability. This finding highlights a critical challenge and a vital area for future research, and extending such interpretability analyses is crucial for moving these models from "black box" predictors to tools that can generate essential biological insights.

## Runtime analysis

Our runtime analysis revealed computational efficiency profiles across the DNA foundation models on a single A100 GPU (Fig. 4). Despite the logarithmic scale of sequence lengths, none of the models demonstrated clear quadratic scaling, suggesting that attention computation was not the dominant factor within our experiment of batch size equal to 1. However, testing with larger batch sizes to potentially observe quadratic scaling would require multiple and more powerful GPUs, which we leave for future work. HyenaDNA models exhibited the most favorable scaling characteristics, particularly for longer sequences. The HyenaDNA-160 K model maintained relatively stable performance for shorter sequences (below 2 K nucleotides), while HyenaDNA-1M showed excellent scaling even at sequence lengths approaching 500 K nucleotides. This efficiency can be attributed to HyenaDNA's architecture that leverages long convolutions with implicit parameterization and data-controlled gating. DNABERT-2 demonstrated competitive runtime for sequences approaching its recommended length limit (~2 K nucleotides), beyond which we observed a dramatic runtime increase, indicated by the spike in the orange line. NT-v2 consistently required the highest runtime across all sequence lengths,

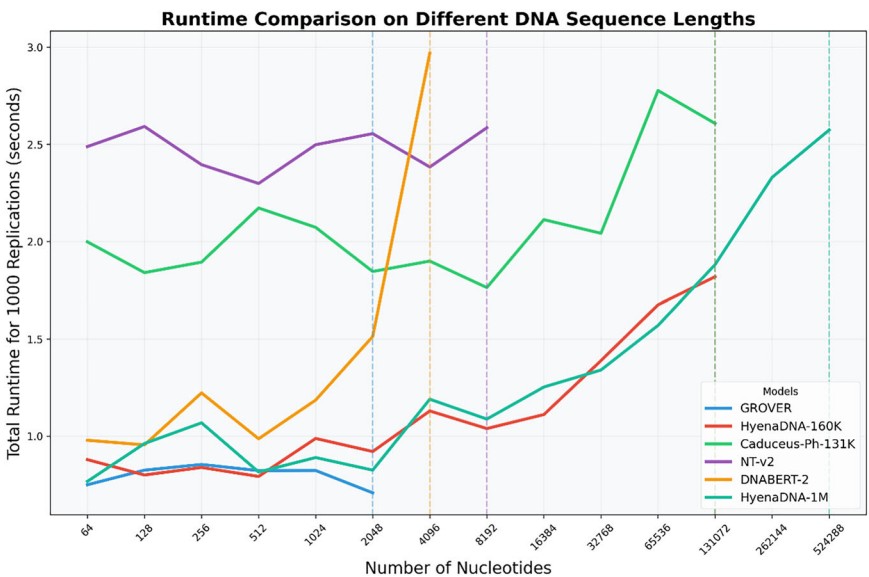

**Fig. 4 | Runtime comparison of DNA foundation models.** The sequence lengths range from 64 to 524,288 nucleotides, measured as total seconds for 1000 replications using batch size 1 on an A100 GPU. Dashed vertical lines indicate points where sequence length either exceeds the supported/recommended model input limit, or exceeds GPU memory capacity.

~2.5 s for 100 replications, which reflects its substantially larger model size of 500 M parameters, yet the runtime is highly stable. GROVER demonstrated the fastest runtime among all tested models for sequences within its supported length range (2 K nucleotides), while Caduceus-Ph-131K showed moderate performance with gradually increasing runtime as sequence length increased.

## Discussion

Our comprehensive study provides a systematic evaluation of five state-of-the-art DNA foundation models (DNABERT-2, NT-v2, HyenaDNA, Caduceus-Ph, and GROVER), focusing on the quality of their zero-shot embeddings across a diverse suite of genomic tasks. A key strength of our benchmark is the breadth and diversity of our comparisons, evaluating zero-shot embeddings across numerous genomic tasks, species, and sequence analysis paradigms, including classification, regression (gene expression prediction), and variant effect quantification. Our methodology, centered on using random forest classifiers for downstream tasks, was chosen to minimize inductive biases from complex classifier tuning, allowing a more direct assessment of the embeddings themselves. It is crucial to note, however, that direct performance comparison with results from original model publications can be nuanced due to differences in fine-tuning strategies, classifier choices, and pooling methods. A significant contribution of our study is the rigorous analysis of output pooling methods, where we demonstrated that mean token embedding consistently and significantly outperforms sentence-level summary tokens (average AUC improvements of 1.4%–8.7%) and maximum pooling, offering a clear path for similar tasks.

Our benchmark on sequence classification tasks reveals distinct strengths and weaknesses for each model. Caduceus-Ph established itself as the best-performing model for human genome classification, consistently outperforming others in transcription factor binding site prediction and promoter region identification. GROVER presented a balanced profile, showing generalized strengths across diverse tasks, from human epigenetic modification detection to bacterial promoter identification. DNABERT-2 also proved a strong and consistent performance, excelling notably in splice site prediction and showing remarkable performance in yeast epigenetic mark prediction. NT-v2's primary strength lies in epigenetic modification detection, where it

performed well on human 5mC/6 mA datasets and was the top performer for 4mC detection across several other species. Finally, HyenaDNA, while being generally modest in binary classification, demonstrated an exceptional and distinct capability in multi-class settings, particularly in distinguishing between different regulatory region types. A common trend was that performance was generally higher on human-centric tasks and for distinct genomic region classification compared to the more subtle signals of epigenetic modifications.

The implications of our findings extend beyond technical performance into biological and clinical research. The superiority of Caduceus-Ph in human TFBS prediction makes it a prime candidate for studies focused on dissecting human gene regulatory networks. The consistent, strong performance of DNABERT-2 in tasks like promoter, enhancer, and splice site identification positions it as a powerful and reliable tool for understanding gene regulation in human diseases. NT-v2's performance on epigenetics across species suggests its utility for comparative epigenomic studies across diverse organisms. Furthermore, HyenaDNA's strength in multi-class classification could be leveraged for genome-wide annotation tasks to distinguish between different functional elements, while its scalability makes it ideal for analyzing large-scale genomic rearrangements. By benchmarking these models across diverse genomic tasks, our study provides guidance for researchers to select the most appropriate model for their specific biological questions, ranging from developmental biology to personalized medicine.

Besides classification tasks, we also evaluated DNA foundation model performance on predicting gene expression levels from DNA sequences, which revealed both specific strengths and notable limitations of current zero-shot embeddings. While the prediction performance for most genes was modest on average, we identified a consistent subset of highly predictable genes across models, such as *CUTALP* and *DDX11* achieving strong prediction correlation (>0.80). This indicates that for these certain genes, zero-shot embeddings can effectively capture sequence-based regulatory signals. However, when extending the input sequence context from 6 K to ~196 K base pairs, the overall improvement in average prediction performance was not uniformly substantial across all models, though some architectures like HyenaDNA did show small gains.

A critical insight from our study is the impact of pre-training data diversity on model performance. We investigated this by re-pretraining HyenaDNA in the roughly identical setting, but on DNA-BERT-2's multi-species dataset comprising 135 species across 6 taxonomic categories. The newly pre-trained model demonstrated significant performance gains over the original human-genome-trained HyenaDNA, achieving statistically significant improvements in 14 of 49 evaluated datasets, particularly in cross-species generalization and epigenetic pattern recognition. However, the original human-genome pre-trained model retained advantages in 3 datasets, suggesting that species-specialized pre-training data can still retain benefits for certain tasks. These findings demonstrate that multi-species pre-training enhances generalizability for diverse genomic tasks and that pre-training data composition should be considered a critical design choice for future DNA foundation models.

In a comprehensive variant effect quantification benchmark, we observed a clear and fascinating distinction between a model's ability to identify pathogenic variants and its ability to predict the functional impact of quantitative trait loci (QTLs). In the task of distinguishing pathogenic from common SNPs, the transformer-based models NT-v2 and Caduceus-Ph were the surprising top performers, substantially outperforming all other models, including those specifically designed to predict functional genomic tracks like Enformer. The dominant performance of NT-v2, which achieved an AUC of 0.73, is a profound finding. It suggests that pre-training on a vast and diverse collection of genomes enables the model to learn an intrinsic, fundamental grammar of DNA sequence "fitness," allowing it to recognize sequence-level patterns indicative of deleterious mutations without any explicit training on variant pathogenicity. In sharp contrast, on the QTL benchmarks (e.g., eQTLs, sQTLs), the specialized models such as AlphaGenome and Enformer were the clear and consistent winners. This dichotomy illustrates a critical concept: foundation models have learned a general, context-free understanding of what makes a DNA sequence "broken," but specialized models excel at understanding the specific, context-dependent regulatory grammar that determines whether a variant is "functional" in a particular tissue.

In summary, while we focused on benchmarking zero-shot embeddings in our current study, our findings provide several concrete directions for optimizing fine-tuning approaches to enhance DNA foundation model abilities. First, our systematic comparison of pooling methods revealed that mean token pooling significantly outperforms summary token pooling and maximum pooling, suggesting that downstream fine-tuning may leverage aggregated sequence representations rather than relying on single summary tokens. Second, our controlled pre-training experiment demonstrated that the choice of pre-training data substantially influences model performance on specific tasks. This finding suggests that researchers may select foundation models with appropriate pre-training data distributions that align with their target application domains prior to fine-tuning.

Our benchmark also revealed important trade-offs and limitations of DNA foundation models. When benchmarked against a baseline convolutional neural network, DNA foundation models showed advantages in some human genome classification tasks but underperformed in several multispecies and epigenetic tasks, given that we used solely zero-shot embeddings from frozen DNA foundation models while the CNN was fully trained on each specific task. This highlights the balance between general-purpose pre-training and task-specific optimization. Our comparison between DNA foundation models and specialized genomic models (such as Enformer and AlphaGenome in our variant effect tasks) also reveals a fundamental trade-off between versatility and task-specific optimization. While foundation models demonstrate remarkable adaptability, specialized models incorporate inductive biases that can yield superior performance within their targeted domains. This complementarity suggests promising directions for hybrid architectures that combine pre-

trained genomic representations with task-specific components. The limitations in interpretability were also apparent. Our TAD region recognition experiment found no evidence that NT-v2 inherently recognizes higher-order chromatin structures in its zero-shot attention patterns, and the lack of accessible attention mechanisms in models like HyenaDNA and Caduceus-Ph poses further challenges.

## Limitations

Despite the comprehensive nature of our analysis, we acknowledge several limitations that warrant further exploration. First, our sequence classification benchmark was relied on public datasets which were mostly built from smaller, curated windows around regulatory elements, rather than assays that exhaustively sample the genome. These datasets focus on well-defined biological traits and contain samples of genomic regions to highlight local regulatory signals. Benchmarking on these datasets enables reliable comparisons but inevitably captures only a part of the regulatory landscape. Consequently, long-context models such as Enformer and Sei were not run on these small-window sequences. Another limitation is that, while fine-tuning may introduce biases in evaluating foundation models, there is still a need to investigate the fine-tuning potential of different models appropriately. For instance, NT-v2, with its larger size (500 M parameters) compared to other foundation models, may exhibit more significant improvement when fine-tuned for specific applications. Finally, exploring model ensembles that leverage the complementary strengths of different DNA foundation models may be proven even more effective, in addressing the diverse challenges in genomic sequence analysis.

In conclusion, our study provides a comprehensive evaluation framework for DNA foundation models, offering insights into their strengths, limitations, and potential areas for improvement. The findings presented here can guide researchers in selecting appropriate models for specific genomic tasks and highlight promising directions for future development in this rapidly evolving field.

## Methods

### DNA foundation language models

To evaluate DNA foundation language models comprehensively, we identified the three most recent state-of-the-art DNA foundation language models, including DNABERT-2[10], Nucleotide Transformer version-2[11], HyenaDNA[12], Caduceus[13], and GROVER[14]. These foundation models take DNA sequence as input, tokenize into sequence of tokens, and generate embeddings of fixed dimension for each token after passing multiple layers. In the following, we will briefly describe these three models.

DNABERT-2[10] has the network architecture similar to Bidirectional Encoder Representations from Transformers (BERT)[25], which usually contains a positional embedding layer added to input embeddings, and a series of encoders each consisting of a multi-head self-attention layer and a feedforward network. It is pre-trained using the masked language modeling approach on genomes from 135 species, including the human reference genome. DNABERT-2 tokenizes DNA sequences by the Byte Pair Encoding (BPE) method, which is an iterative algorithm that searches for nucleotides combinations and builds the vocabulary at the same time; it makes no assumption on fixed words and grammars, so each input sequence is independently tokenized merely based on its pattern. It is worth noting that the number of tokens in the tokenized sequence is not fixed in DNABERT-2. DNABERT-2 modifies the architecture of BERT by using Attention with Linear Biases (ALiBi) instead of positional embedding layer. DNABERT-2 has about 117 million trainable parameters, the output embedding dimension is 768. There is no hard limit on the input sequence length, although the runtime is still quadratically increasing with sequence length.

Nucleotide Transformer Version 2 (NT-v2)[11] is also based on the BERT architecture, and it is pre-trained using the masked language

modeling approach on genomes from 850 species, including the human reference genome. To tokenize DNA sequence, NT-v2 employs the 6 mers tokenization method that uses a sliding window of size 6 and reads every 6 nucleotides; if there are leftover elements at the end of sequence, nucleotides will be tokenized individually into {A, T, C, G, N}. Therefore, the number of tokens produced by the tokenizer will be ~1/6 of DNA sequence length. NT-v2 modifies BERT by replacing the learned positional embeddings with the rotary embeddings, which rotates the embeddings output by each attention layer based on the token's position, and the Swish activation without bias. These modifications reduce the number of model parameters in Nucleotide Transformer Version 1, and thus reduce the computation cost. The largest NT-v2 model has around 500 million trainable parameters, the output embedding dimension is 1024, and the input sequence length limit is 12,000 nucleotides.

HyenaDNA[12] differs from the architectures of DNABERT-2 and NT-v2 by eschewing the attention mechanism in favor of a decoder-based architecture. HyenaDNA is pre-trained exclusively on the human reference genome using a next nucleotide prediction approach. The key component of this model is the Hyena operators, which integrate long convolutions with implicit parameterization and data-controlled gating. Benefiting from this architecture, HyenaDNA can process very long DNA sequences with fewer model parameters than attention-based transformer. This enables a straightforward tokenization approach in HyenaDNA, where each nucleotide is treated as an individual token. HyenaDNA can also perform in-context learning such as soft-prompting[12,26], and details can be found in its original article. The largest HyenaDNA model has around 30 million trainable parameters, the output embedding dimension is 256, and the input sequence length limit is one million nucleotides.

Caduceus-Ph[13] is a long-range DNA foundation language model that leverages the MambaDNA architecture—an extension of selective state space models (SSMs)—to capture the unique structural and contextual properties of genomic sequences. It introduces two key innovations: bi-directional sequence modeling via the BiMamba block, which enables the incorporation of both upstream and downstream genomic context, and reverse complement (RC) equivariance achieved through a dedicated module that respects the intrinsic reverse complementary nature of DNA. In our implementation, RC handling is performed via a post-hoc conjoining approach, simplifying embedding extraction and facilitating the ensembling of predictions across strands at inference time. Unlike traditional attention-based architectures, Caduceus-Ph operates at nucleotide resolution and scales efficiently to long sequences without incurring quadratic computational overhead. The largest Caduceus-Ph configuration comprises ~35 million trainable parameters, features an output embedding dimension of 256, and maximum input sequence of 131 thousand nucleotides.

GROVER[14] is a DNA foundation language model designed to learn the intrinsic "grammar" of the human genome by capturing both local token properties and long-range contextual dependencies. Built on a transformer encoder architecture akin to BERT, GROVER comprises 12 transformer layers that include multi-head self-attention, feedforward networks, and normalization layers. Its novelty lies in the use of byte-pair encoding (BPE) to tokenize genomic sequences—the vocabulary is iteratively optimized over 600 cycles to balance token frequency and capture the heterogeneous sequence composition of the genome. Trained with a masked token prediction objective, GROVER not only reconstructs masked regions but also develops token embeddings that encapsulate key genomic features such as nucleotide frequency, GC content, and sequence length. The GROVER configuration comprises ~117 million trainable parameters, features an output embedding dimension of 768. It supports an input sequence length limit of 512 tokens. Due to the variable-length nature of BPE tokens which can range from single nucleotides to longer 6 mers, the corresponding

nucleotide sequence length can vary, typically averaging around two thousand.

## Benchmarking datasets

**Sequence classification datasets.** To unbiasedly evaluate the foundation models, we collected and curated datasets from five distinct sequence classification sources. These datasets were selected to reflect a wide range of potential downstream tasks, ensuring they are both challenging and achievable. While these datasets are not genome-wide in terms of exhaustive coverage, they have been carefully curated by focusing on sub-sampled genomic regions that capture local regulatory signals within controlled windows, each with specific sampling strategies to reduce noise and redundancy.

DNase-I hypersensitive sites detection[27]: The datasets used in the study consist of positive DNA sequences for the 280 Dnase I hypersensitive sites (DHS), and negative sequences for the 737 non-Dnase I hypersensitive sites, originally collected by ref. 28 and has been widely used for benchmarking the detection of DNase I hypersensitive sites[29,30]. The positive sequences are experimentally identified DNase I hypersensitive sites (DHSs), which are markers for open chromatin and active regulatory elements. The 737 negative sequences are genomic regions verified as non-DHS. Identification of the DNA sequences containing DHS is crucial for detecting DNA regulatory regions, as DHS is indicative of genomic regulatory regions. The datasets consist of a curated set of human genomic regions representing a specific challenge set designed for direct comparison against prior computational methods. sequence length ranges from 225–275 base pairs.

5mC and 6 mA modifications detection in human[31]: We use the two benchmark datasets to predict 5-methylcytosine (5mC) and N6-methyladenosine (6 mA) modifications in human (Homo sapiens) DNA in this study. The datasets were originally collected by ref. 32 which have been subsequently adopted by relevant tasks[33]. These datasets consist of short, 41 bp sequences centered on specific cytosine or adenine sites in the human genome. Both the positive (methylated) and negative (non-methylated) samples are derived from experimental profiling studies. Sequences with high similarity were computationally filtered out to reduce redundancy and potential bias.

Promoter identification in multiple species[34]: We use 8 datasets from a study that established a benchmark for identifying promoters across multiple species, covering 4 distinct organisms including human (4 different cell lines of GM12878, NHEK, HeLa-S3, HUVEC), *B. amyloliquefaciens*, *R. capsulatus*, and *Arabidopsis* (TATA and non-TATA). These datasets focus on genomic regions proximal to transcription start sites (TSS). The positive samples are promoter sequences derived from experimentally-based ENCODE annotations for human cell lines, as well as established promoter sequences for bacterial and plant species. The negative samples for this task were computationally generated by identifying non-promoter genomic regions that have the highest sequence similarity to the corresponding positive promoter sequences, creating a challenging classification problem. Sequence lengths vary significantly, from under 100 bp in bacteria to over 2000 bp in human cell lines.

N4-methylcytosine (4 mC) site detection in multiple species[35]: For this task, we utilized six distinct datasets that the authors collected to benchmark, corresponding to *E. coli*, *C. elegans*, *G. pickeringii*, *G. subterraneus*, *D. melanogaster*, and *A. thaliana*, consist of 41 bp sequences centered on potential 4 mC sites. The positive samples are experimentally identified 4 mC sites collected from public databases. The negative samples (non-4 mC sites) were generated by randomly sampling sequences from the respective genomes. It is important to note that while 4mC is a well-established modification in prokaryotes like E. coli and the Geobacter species, for eukaryotic species (*A. thaliana*, *C. elegans*, *D. melanogaster*), 4mC site annotations are often reliant on computational predictions rather than direct, genome-wide high-stringency chemical mapping due to ongoing research into its

prevalence and detection []. This distinction is important for the interpretation of model performance on predicting sequences containing 4 mC sites.

Genomic Benchmarks dataset collection[36]: We also downloaded a suite of datasets from the Genomic Benchmarks collection, which provides benchmarks for classifying key regulatory elements. The tasks from this collection include identifying promoters, enhancers, and open chromatin regions in humans, as well as distinguishing between coding and intergenic DNA and differentiating sequences between species (human and *C. elegans*). For these datasets, positive samples are derived from experimentally-validated sources like the FANTOM5 project and the Ensembl Regulatory Build. The negative samples consist of randomly sampled, length-matched sequences from non-overlapping genomic regions, and sequence lengths typically range from 200 to over 2000 base pairs.

We also adopted a large collection of datasets from the downstream task benchmarks in DNABERT-2 and NT-v2. These datasets cover a diverse range of tasks including promoter (proximal and core), transcription factor binding site, splice site, and epigenetic mark prediction across species like human, mouse, and yeast. The majority of these tasks are region-specific, focusing on sequences of a few hundred base pairs centered on the functional element. A common paradigm in their construction is the use of experimentally-derived positive samples (e.g., from ENCODE ChIP-seq), while negative samples are sometimes generated through methods like dinucleotide shuffling, random sampling with matched GC-content, or even adversarial example generation to increase task difficulty. Since these datasets were originally pre-processed into specific splits, we re-shuffled them to ensure a fair and rigorous comparison in our framework.

With all the datasets we collected, there are in total 57 datasets included in this study, including 52 binary classification datasets and 5 multi-class datasets. A key characteristic of these 57 classification tasks is their focus on short sequences (41 to ~2000 bp), a design intended to test local signal recognition. Detailed descriptions and our naming of all 57 datasets can be found in Supplementary Note 1. In Supplementary Note 2, we also annotated each dataset on whether the sequences are relied on experimentally validated or golden standard sources. The specific training size, testing size, class label distribution, and details of sequence lengths for all datasets used in our study can be found in Supplementary Data 6.

To facilitate a systematic analysis, we categorized these 57 datasets into four distinct categories based on the nature of their respective classification tasks: (1) Human Genome Sequence Region Classification: This category encompasses tasks such as identification of transcription factor binding sites, promoter regions, and other functional elements within the human genome. (2) Multi-Species Genome Sequence Region Classification: These tasks involve distinguishing genomic regions across different species, for example, differentiating between human and Caenorhabditis elegans (worm) genome sequences. (3) Human Genome Epigenetic Trait Classification: This group includes tasks related to identifying epigenetic modifications specific to the human genome, such as detection of N4-methylcytosine (4mC) sites. (4) Multi-Species Genome Epigenetic Trait Classification: These tasks focus on identifying and classifying epigenetic traits across multiple species' genomes. We present and analyze our findings separately for each of these four dataset categories, allowing for a comprehensive assessment of the models' capabilities and limitations in various genomic classification scenarios. It is important to note that the input sequences for all 57 classification datasets, which range from 41 bp to ~2000 bp, fall well within the maximum input length of all foundation models evaluated in these tasks. Therefore, a direct comparison was possible without sequence truncation, ensuring that each model was evaluated on identical, biologically relevant inputs for these tasks.

**Gene expression prediction datasets.** To evaluate DNA foundation models in predicting gene expression, we used the GTEx v8 dataset[37], which provides gene expression measurements across multiple human tissues. We focused on whole blood (610 subjects, 21,004 genes) tissue. We generated subject-specific DNA sequences by incorporating individual genetic variants from whole-genome sequencing VCF data into the human reference genome (GRCh38/hg38). For each gene, we extracted sequences spanning 3000 base pairs upstream and downstream of the transcription start site (TSS), resulting in ~6000-nucleotide sequences that include subject-specific SNPs and insertion/deletion variants. The variants have been phased so both alleles are generated. Consequently, for each gene we have generated {subject-specific sequences on both alleles, subject gene expression} pairs.

To consider models which are capable of processing longer sequences, we created an extended-context dataset with sequences spanning TSS ± 98 K base pairs. We included sequences of such length for two key reasons: (1) Some versions of HyenaDNA and Caduceus can process sequences significantly longer than 6000 bps, and we wanted to assess whether they generate improved zero-shot embeddings with extended genomic context; and (2) Enformer[23], another transformer model that has demonstrated success in predicting gene expression, accepts sequences up to 196,608 bps long, allowing us to benchmark it against DNA foundation models.

For each gene, we regressed the gene expression on covariates provided by GTEx v8, including demographic variables such as sex, probabilistic estimation of expression residuals (PEER) factors, and top genotype principal components used to account for population structure[37]. The residuals from this regression, representing covariate-corrected gene expression levels, served as the quantitative trait for our prediction models to learn from the sequence data. The 6000 bps dataset includes all genes. For the long sequence dataset, due to computational constraints, we randomly selected 1000 genes and excluded genes not present in whole blood tissue dataset or genes whose TSS ± 98 K exceed chromosome boundaries, making the final total of 768 genes.

**Variant effect quantification datasets.** To directly assess the capacity of DNA foundation models to represent genetic variants within their zero-shot embeddings, we developed two distinct benchmarks: one focused on differentiating pathogenic from common variants, and another on identifying functional quantitative trait loci (QTLs).

Our first benchmark utilizes data from the Genomics Long-Range Benchmark[38], which provides two distinct categories: pathogenic single nucleotide polymorphisms (SNPs) associated with disease phenotypes and common SNPs frequently observed in the population. Following the methodology of generating paired sequences, we constructed a reference sequence and a corresponding alternative sequence for each SNP by replacing the central nucleotide. This process resulted in a dataset of {reference sequence, alternative sequence} pairs, where each pair corresponds to a single SNP and is labeled as either pathogenic or common. To accommodate models with different input length capabilities, we created two versions of the dataset. For models supporting longer contexts, we extracted 196,608 bp sequences, resulting in 22,222 pathogenic and 17,374 common sequences. For models with shorter input limits, we used 6000 bp sequences, yielding 22,239 pathogenic and 17,398 common sequences. In both cases, a small number of SNPs were omitted where the required sequence length exceeded chromosome boundaries. The purpose of this benchmark is to evaluate if the change between the reference and alternate allele embeddings is distinct enough to distinguish pathogenic from benign genetic variation.

As a second benchmark, we adopted the curated putative causal QTL dataset from the Borzoi[39] study. The dataset contains high-confidence causal variants, was originally derived from GTEx v8 and

has been statistically fine-mapped. We focus on QTLs from the whole blood tissue. In total, there are 1896 expression QTLs (eQTLs), 540 splicing QTLs (sQTLs), 116 intronic-polyadenylation QTLs (ipaQTLs), and 142 polyadenylation QTLs (paQTLs). A key strength of this dataset is its rigorously constructed negative set, where each putative causal QTL is paired with a carefully matched non-causal variant based on characteristics such as distance to the relevant functional site and the expression level of the associated gene. For each variant, this resulted in a {reference sequence, alternative sequence} pair, labeled according to its specific QTL classification (e.g., eQTL, sQTL) or as a non-causal variant. Similar to the pathogenic versus common SNP benchmark, we constructed sequence datasets of both long (196,608 bp) and short (6000 bp) sequences, again filtering out a small number of SNPs due to chromosome length limits.

**TAD region dataset.** To evaluate DNA foundation models' ability to recognize topologically associating domains (TADs), we utilized the processed dataset of TAD boundaries measured in IMR90 cells following the Enformer study[23]. The dataset contains TAD regions of 2400 base pairs across the human genome, each with an estimated boundary strength score reflecting the magnitude of interaction changes at the boundary. We filtered the dataset to include only TAD regions with boundary strength in the top 5%, and then selected 1500 TAD regions with the strongest boundaries, similar to the workflow in Enformer study. For each selected TAD region, we extracted 6000 bp sequences from the human reference genome, positioning the 2400 bp TAD region in the center with flanking context on both sides. This generated a total of 1500 TAD-centered sequences. As a control, we randomly sampled 1500 background sequences of identical length (6000 bp) from the reference genome, representing genomic regions without known TAD boundaries. This balanced design allowed for direct comparison between TAD-centered sequences and background sequences.

The objective of this dataset was to compare the attention matrices generated by foundation models when processing TAD-centered sequences versus background sequences. Since NT-v2 is the only model in our evaluation that both returns attention matrices and supports sufficiently long input sequences, we focused exclusively on NT-v2 for this task.

## Benchmarking methods

**Sequence classification benchmark.** For each classification task, we first established a clear data partitioning strategy. We maintained the training and testing splits from the original works where they were available and clearly defined. For datasets lacking pre-defined partitions, we applied a random 70:30 split for training and testing. For datasets adopted from the downstream task benchmarks in DNABERT-2 and NT-v2, we performed a new random 70:30 split maintaining the original class label distributions, ensuring a standardized and fair comparison across all models in our framework. This re-shuffling is always performed unless there was clear evidence that the original datasets were not manually processed by the authors of DNA foundation models. It should also be noted that due to the pre-processed nature of many classification datasets lacking complete chromosomal information, a uniform whole-chromosome holdout strategy was not feasible.

On the choice of supervised classifiers, we focus on classifiers that require minimal hyperparameter tuning to ensure that performance reflects the quality of DNA foundation model embeddings rather than classifier optimization. Additionally, we select classifiers that inherently handle high-dimensional inputs without requiring dimensionality reduction, as such preprocessing introduces another layer of tuning and potential confounding. Therefore, we included random forest[40], Naïve Bayes, and elastic-net logistic regression. During training, we performed 5-fold cross-validation that divides the training set into

non-overlapping train-validation pairs for hyperparameters tuning, and then reported the testing performance on the test set. The hyperparameter grids for all these classifiers are detailed in the Supplementary Table 5. We evaluated model performance on the test set using AUROC (AUC) and accuracy. AUC serves as our primary measure of performance throughout this work, with the accuracy providing supplementary information.

On the choice of baseline models for comparison against DNA foundation models, we noted that the short-sequence nature of the benchmark datasets is by definition incompatible with genomic models requiring longer and fixed-length inputs, such as Enformer, Sei, and AlphaGenome. Forcing these short sequences into a long-context model via excessive 'N' padding would create additional artificial signal structures to handle. Therefore, to establish a relevant, task-specific baseline, we implemented a simple CNN consisting of three convolution layers with pooling and a classification head. The CNN architecture takes 5-dimensional one-hot encoded input corresponding to nucleotides A,T,C,G,N and comprises three 1D convolutional layers with 64, 128, and 256 channels respectively, max pooling after the first two layers, adaptive global max pooling, and a final linear layer for classification. Unlike the foundation models where embeddings remained frozen during training, the CNN was trained directly on one-hot encoded DNA sequences with all parameters updated during training. This comparison aimed to identify when pre-trained DNA foundation models could outperform task-specific neural networks.

To evaluate statistical differences in performance between models, we applied the DeLong's test[41] for comparing AUC values. For each dataset (task), pairwise comparisons were conducted between all model pairs using one-sided tests with a significance threshold of $\alpha = 0.01$. A model's performance was considered significantly better only when its AUC value was higher and the corresponding test yielded $p < 0.01$. For the five datasets involving multi-class classification tasks, where DeLong's test become less adaptable, we used classification accuracy as the primary metric instead.

**Sequence classification benchmark: pooling methods.** For our model-wise comparison in sequence classification benchmark, we ensured fair evaluation by using the same pooling method across all models. However, given the lack of comprehensive studies on optimal pooling methods for DNA foundation models, we also conducted a systematic comparison of different pooling methods for each individual model. In foundation language models, output pooling methods refer to techniques for generating a single, fixed-dimensional embedding to represent an entire sequence[25]. Despite their importance, there has not yet been a comprehensive study on which output pooling methods are most effective for DNA foundation models. All DNA foundation models in this study create additional special tokens during tokenization, aligning with common practice in natural language processing[42-44]. We focused on three common output pooling strategies:

Sentence-level summary token method: In the original BERT architecture, a special "CLS" token is appended to the start of every tokenized sequence to serve as a sentence-level summary. After processing through multiple self-attention layers, this token captures the context of the entire input sequence. DNABERT-2, NT-v2, and GROVER implement this approach. Similarly, HyenaDNA and Caduceus-Ph append an end-of-sequence "SEP" token that can be used to represent the whole sequence[12,13].

Mean pooling method: This approach calculates the average of all non-padding token embeddings in the sequence to create a unified representation. This potentially captures information from all parts of the sequence equally.

Maximum pooling method: This method selects the maximum value across each dimension from all token embeddings, potentially highlighting the most salient features.

For these comparisons, we employed the same statistical testing framework described earlier, using the DeLong's test with a significance threshold of 0.01 to determine significant differences in performance.

**Gene expression prediction benchmark.** To evaluate DNA foundation models in predicting gene expression, we employed a regression-based approach using data from the GTEx v8 dataset. As described in the dataset subsection, we worked with two sequence lengths: shorter 6000 nucleotides (applied to all 610 subjects and all 21,004 genes) and longer sequences of up to 196 K nucleotides (applied to a subsample of 768 genes).

For the shorter sequences, we generated zero-shot embeddings using DNABERT-2, NT-v2, HyenaDNA, Caduceus-Ph, and GROVER. Due to GROVER's architecture, its input is limited to 512 tokens generated via Byte Pair Encoding (BPE). Since the average BPE token length is approximately four nucleotides, this corresponds to an input of roughly 2048 base pairs. To accommodate this, we extracted the central 2048 bp from each 6000 bp sequence, ensuring the TSS remained centered. GROVER's performance on this shorter input was then directly compared against the other models that utilized the full 6000 bp sequence. For the longer sequences, we generated embeddings using HyenaDNA-450K, Caduceus-Ph, and Enformer. Caduceus-Ph was provided with the central 131 K nucleotides due to its input length limit. For Enformer, we used the model's final hidden state as its sequence embedding and performed mean pooling over all regions. For each subject and gene across all models, we first generated zero-shot embeddings on both alleles separately, then calculated the mean of these embeddings to represent the joint genetic information from the two alleles.

Since gene expression is represented as a continuous value, we approached this as a regression task. We experimented with random forest and XGBoost[45] as regression models. For each gene, we train a separate regression model, with the subjects split into training and testing sets at a ratio of 75:25, respectively. During training, we performed 5-fold cross-validation using mean squared error as the optimization metric for hyperparameter tuning, with the complete hyperparameter grid detailed in Supplementary Table 6. This regression framework was applied to the whole blood tissue with 610 subjects and 21,004 genes.

For evaluation, we used both mean squared error (MSE) and Pearson correlation coefficient between predicted and actual expression values for each gene, which quantifies how effectively each DNA foundation model could capture the genomic features that influence gene expression levels. We conducted multiple comparative analyses: First, we compared performance metrics between different DNA foundation models on short sequences to assess their relative effectiveness. Second, we compared metrics between Enformer and DNA foundation models on longer sequences to evaluate the benefits of specialized architectures. Third, we performed paired comparisons on the same genes between HyenaDNA (shorter sequences) versus HyenaDNA-450 K (longer sequences), and Caduceus-Ph on shorter versus longer sequences, to determine whether extended genomic context improves prediction accuracy. To assess statistical significance in these comparisons, we applied the paired Wilcoxon signed-rank test, which allows for robust non-parametric comparison of paired samples. Additionally, we identified and analyzed the subset of genes that were best predicted by each model to understand potential model-specific strengths in capturing particular genomic features or regulatory mechanisms.

**Variant effect quantification benchmark.** To directly assess the capacity of DNA foundation models to represent genetic variants within their zero-shot embeddings, we developed a unified robust quantitative framework. Our general methodology, inspired by the Enformer[23] study, was to first compute a unified effect vector for each variant by subtracting the zero-shot embedding of the reference sequence from that of the alternative sequence (embedding(alt) - embedding(ref)). This standardized approach was applied across all models in our comparison, including the general DNA foundation models (DNABERT-2, NT-v2, GROVER, HyenaDNA, Caduceus-Ph), as well as Sei[24], Enformer[23], and, AlphaGenome[22]. We then trained a random forest classifier on these high-dimensional effect vectors to distinguish between the variant labels (e.g., pathogenic vs. common).

To prevent data leakage from effects such as linkage disequilibrium, we employed a strict chromosome-based holdout strategy for evaluation. However, recognizing that a single arbitrary chromosome split could lead to high variance in performance estimates, especially for smaller datasets like ipaQTL and paQTL, we implemented a nested cross-validation scheme to ensure a more robust and stable evaluation. We established three distinct test sets comprised of non-overlapping chromosome groups: Group 1 (chromosomes 3, 6, 9, 12, 16, 18, 19, 21), Group 2 (chromosomes 2, 5, 11, 14, 17, 20, 22, and X), and Group 3 (chromosomes 1, 4, 7, 8, 10, 13, 15). For each of these three main folds, the corresponding group served as the test set. The remaining chromosomes served as a training-validation set, on which we performed a 4-fold inner cross-validation, also partitioned by chromosome, to tune the random forest hyperparameters. Once the optimal parameters were selected, the classifier was retrained on the entire training-validation set and evaluated on the held-out test chromosomes. The final reported test AUC and Cohen's d scores are the average of the performances across these three independent test sets.

For the short sequence (6000 bp) datasets, we compared DNABERT-2, NT-v2, GROVER, Caduceus-Ph, HyenaDNA, and Sei. Due to model-specific constraints, GROVER received the central 2048 nucleotides and Sei received the central 4096 nucleotides. For Sei, we evaluated two distinct representations: (1) its direct output across 21,907 predictive tracks, and (2) its final hidden state (dimension 15,360) after the spline transformation layer. For the long sequence (196,608 bp) datasets, we compared Caduceus-Ph, HyenaDNA (450 K checkpoint), and Enformer, with Caduceus-Ph using the central 131,072 nucleotides. For Enformer, we assessed both its final hidden state and its 5313 human-specific output tracks. For the long-sequence QTL datasets, we also included AlphaGenome, providing it the central 131,072 nucleotides of the input and averaging its output tracks over the central 2048 bp to derive the effect vector.

For all experiments, no dimensionality reduction was performed on the effect vectors. However, to ensure stable and robust training for models generating relatively high dimensional embeddings (e.g., the hidden states of Sei and Enformer), we adjusted the hyperparameter search space for the random forest. Specifically, we removed the "sqrt" option from the max features tuning grid (Supplementary Table 7), thereby prioritizing the more restrictive "logarithm" option. This standard practice for high-dimensional data helps decorrelate the trees, which stabilizes the training process, reduces the risk of overfitting, and ensures a more reliable performance estimate.

**TAD region recognition benchmark.** We assessed NT-v2's capability to recognize topologically associating domains by analyzing attention patterns across TAD-centered and background sequences. We processed 1500 TAD-centered sequences and 1500 control sequences through NT-v2 and extracted attention matrices from all layers and heads. For each sequence type, we averaged these matrices across all layers, heads, and sequences to produce composite attention patterns.

Our analysis focused on the center 400 tokens (corresponding to the 2400 bp TAD region) in each averaged attention matrix. We compared attention values within this central region between TAD-centered and background sequences to determine whether NT-v2

naturally allocates higher attention to genomic regions containing strong TAD boundaries.

**Pre-training dataset experiment.** To investigate the effect of pre-training dataset diversity on zero-shot embedding quality, we pre-trained HyenaDNA using DNABERT-2's multi-species dataset. This was made possible thanks to the HyenaDNA's reproducible pre-training pipeline, allowing us to pre-train the model on different datasets while maintaining the same training settings for fair comparison.

For this experiment, we used DNABERT-2's multi-species dataset, which encompasses genomes from 135 species across 6 categories, totaling 32.49 billion nucleotide bases (approximately 12 times larger than the human genome dataset). Since the DNABERT-2 dataset contains processed sequences of 1000 bps, we used the HyenaDNA architecture with input length 1000 (HyenaDNA-1K[12]). We strictly maintained the pre-training settings as specified in the original work, including all model hyperparameters, optimizer configurations. We roughly maintained their training steps, and batch sizes.

We compared our multi-species pre-trained model against the official human-genome pre-trained HyenaDNA-1K on 49 of the 57 sequence classification tasks (excluding 8 datasets with sequences >1000 bps). This comparison allowed us to directly assess whether species diversity in pre-training data improves zero-shot embedding quality while controlling for all other variables, including model architecture and training methodology.

**Runtime benchmark.** To evaluate the computational efficiency of DNA foundation models, we measured the time required to generate zero-shot embeddings across varying sequence lengths. Experiments were conducted on a single A100 GPU, measuring the average forward pass time for randomly generated DNA sequences of length $2^k$ (k ranging from 6–19, spanning 64 to ~500,000 nucleotides). We recorded performance for each model until it reached its sequence length limit or exceeded the A100 memory capacity.

All measurements used a consistent batch size of 1 to accommodate the substantial memory requirements of longer sequences. To reduce measurement error, we performed 100 replications and recorded the total runtime. For DNABERT-2 and GROVER, which use Byte Pair Encoding in tokenization with variable token counts, we estimated the number of tokens as (sequence length / 4). This approximation may introduce slight measurement variations due to potential sequence truncation or padding requirements.

### Model configuration selection
For the DNA foundation models we benchmarked, we selected specific model checkpoints based on their capabilities and our experimental requirements. For NT-v2[11], we used the largest NT-v2-500M model, which showed optimal performance in the original study. With HyenaDNA, we included different checkpoints for different tasks: Hyena-160K[12] for default choice of benchmarks, Hyena-450K[12] in specifically gene expression prediction and variant effect quantification, to leverage its ability to process longer sequences, and Hyena-1K[12] for pre-training experiments to match our pre-processing format. For Caduceus-Ph, we consistently employed Caduceus-Ph-131K[13] across all tasks to maximize context length while maintaining efficiency. DNABERT-2[10] and GROVER[14] only have one model checkpoint. The embedding dimensions of all DNA foundation models are detailed in Supplementary Table 8. For Enformer[23], we used the widely-accepted official PyTorch implementation[46]. Sei[24] only has one model checkpoint. For AlphaGenome[22], we used their provided API to access the regular model checkpoint.

### Reporting summary
Further information on research design is available in the Nature Portfolio Reporting Summary linked to this article.

## Data availability
The processed benchmarking datasets generated in this study (excluding the Gene Expression Prediction benchmark) have been deposited in the Hugging Face Hub under https://huggingface.co/datasets/hfeng3/dna_foundation_benchmark_dataset. The Gene Expression Prediction benchmark involves human participant data and is available under restricted access to protect participant privacy; raw individual whole genome sequencing data can be obtained by application via the GTEx Protected Data Access portal (https://gtexportal.org/home/protectedDataAccess); the public aggregated gene-expression releases are available at https://www.gtexportal.org/home/downloads/adult-gtex/qtl. The original datasets used for the sequence-classification benchmarks are available from the sources cited in the main text. The Variant Effect Quantification (pathogenic versus common variant) benchmark dataset is available from the Genomics Long-Range Benchmark repository on the Hugging Face Hub (https://huggingface.co/datasets/InstaDeepAI/genomics-long-range-benchmark/tree/main/variant_effect_pathogenic). The QTL benchmark files were accessed from the Borzoi paper repository (Google Cloud Storage: https://console.cloud.google.com/storage/browser/borzoi-paper/qtl). The TAD region recognition benchmark files were accessed from the Basenji Hi-C repository (Google Cloud Storage: https://console.cloud.google.com/storage/browser/basenji_hic/insulation).

## Code availability
The code used to develop the model and perform the analyses and generate results in this study is publicly available and has been deposited at GitHub (https://github.com/ChongWuLab/dna_foundation_benchmark) under MIT License. The specific version of the code associated with this publication is archived in Zenodo and is accessible via 10.5281/zenodo.17349484[47].

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

## Acknowledgements

This work is partially supported by grants from the National Institutes of Health R01CA263494, U01CA293883, P01CA296429, R21HL170213, P30CA016672 and P50CA217674, and Cancer Prevention & Research Institute of Texas (CPRIT) grant RP230166. The content is solely the responsibility of the authors and does not necessarily represent the official views of the National Institutes of Health and Cancer Prevention & Research Institute of Texas.

## Author contributions

C.W. and P.W. conceived and designed the study. H.F. and C.W. curated the benchmarking datasets. H.F. performed the benchmarking analysis and summarized the results. H.F. and C.W. wrote the initial version of the manuscript. L.W., B.Z., C.H., J.Z., J.W., and L.L. provided critical feedback and contributed to the interpretation of the results. All authors wrote and proofread the manuscript.

## Competing interests

The authors declare no competing interests.
