## [Transparent Peer Review file · Nature Communications]

Benchmarking DNA Foundation Models for Genomic and Genetic Tasks

Corresponding Author: Dr Chong Wu

Version 0:

Reviewer comments:

Reviewer #1

(Remarks to the Author)

This paper presents a collection of genomic sequence classification benchmark tasks designed to measure the performance of DNA foundation language models. The study aims to assist future researchers in selecting and optimizing DNA language models. The results showcase the strengths and weaknesses of the three models tested and the comparative efficacy of using different sequence embedding methods. The benchmark is well-conceived and covers four DNA classification tasks over 57 multi-species datasets. This breadth ensures that the benchmark provides a holistic view of the capabilities of DNA language models across various classification challenges. Such a diverse and extensive benchmark is unparalleled in recent genomic benchmarks. Several detailed comments about the work are listed in the following:

The scope of models used in this comparison seems limited. The most recent DNA foundation models benchmark paper, Bend (Marin, F. I. et al., 2024), used 15 versions of various models. Including more models, such as GROVER, DEEPSEA, or even newer models like MAMBA, could further validate the evaluation methods and conclusions of the study. Moreover, adding a simple downstream baseline, such as a shallow two-layered CNN, could help validate the proposed benchmarking and provide a direct and obvious comparison to the complex language models.

The benchmark performance of models on DNA modification seems to be much lower in accuracy than other methylation prediction/imputation models (e.g., DeepCpG). So, in general, how do foundation models compare to other task-specific models in performance?

Lack of comparison with other classifiers: The study argues that random forest should be utilized due to its minimal tuning requirement (compared with CNN used for zero-shot embeddings in Bend (Marin, F. I. et al, 2024), which requires much more hyperparameters) and its superior performance over XGBoost. However, other simple machine learning models, such as SVM, logistic regression, KNN or Naïve Bayes also generally require minimal parameter tuning. Providing a detailed comparison of more classifier models could help to fully capture the representation in the last hidden embedding, as well as solidify the argument in the superiority of choosing Random Forest over other models, including the CNN used in past papers.

Models are inconsistently pre-trained on different datasets. To better understand why one model excels another on certain tasks, it could be useful to train all language models from scratch using the same dataset. So we can gain a better insight into the origins of advantages and disadvantages. DNABERT2 and NT-V2 are originally pre-trained on different multi-species sequence datasets. HyenaDNA is pre-trained purely on the human reference genome. Training all models on the same dataset under the same conditions would allow for a more controlled comparison and better assessment of the generalizability of the evaluation methods.

Alternative pooling methods should be tested and used in a context-dependent manner. The study thoroughly compared the effects of two pooling methods over different models. It demonstrates that mean pooling is generally more effective than summary-level token pooling in generating zero-shot embeddings. However, some models might be intrinsically designed to favor certain pooling methods (for example, NT-v2 with its final averaging pooling layer in pre-training). Since the pooling method can be model-specific, task-specific, and sensitive to sequence length, testing more pooling methods (such as max-pooling, attention-based pooling, or hybrid approaches) could facilitate more intriguing analyses.

The study could be more impactful and cutting-edge by extending the test datasets to include longer sequences. This would

allow in-depth exploration of the relationship between model performance and sequence length. Transformer-based models are known to be inefficient with long sequences. Test-time training models and MAMBA that scale up to million-bps-long sequences may be compared and discussed in this context.

Although this study focuses on zero-shot embedding to avoid potential bias introduced by fine-tuning, fine-tuning is still widely used and could significantly alter or improve the model's performance. Future research on how fine-tuning affects performance across different tasks could provide a more complete picture of each model's capability.

While the study is comprehensive in its evaluation of classification tasks, real-life research questions are not limited to this one type of task. Incorporating more types of tasks, such as the more complex regression-based tasks, could contribute to the generalizability of the proposed method and make this work more impactful.

(Remarks on code availability)

Reviewer #2

(Remarks to the Author)

The manuscript benchmarks three DNA foundation models—DNABERT-2, Nucleotide Transformer version-2 (NT-v2), and HyenaDNA—for genomic sequence classification using zero-shot embeddings. While the authors' approach to comparing the performance of different sequence embeddings is generally appropriate and well-designed, its effectiveness is undermined by the poor quality of datasets chosen for model evaluation. Specifically, the human genome datasets are very limited in size, and some are not based on experimental data. I strongly advise the authors to carefully review the datasets selected for evaluation and select genome-wide, high-quality experimental data. Additionally, to make the manuscript's contribution more impactful, the model comparison should include other models that focus on regulatory signal predictions from genomic sequences.

Major points:

Although the datasets cover a diverse range of tasks, the size of each individual human dataset is small and inadequate for a comprehensive comparison of DNA foundation models. A more appropriate approach would be to use whole-genome datasets, as this would provide a more accurate assessment of the models' ability to generalize across diverse genomic sequences.

The authors should clarify why their performance results differ from those in previous studies. For example, the original NT-v2 paper (The Nucleotide Transformer: Building and Evaluating Robust Foundation Models for Human Genomics) showed that NT-v2 predicts promoters (All, non-TATA, and TATA) better than DNABERT-2. But in this paper, the authors found that DNABERT-2 performed better for some of the promoter classification tasks. The authors should explain what might have caused this difference.

One of the datasets the authors used is 4-methylcytosine (4mC), which is not widely accepted as an epigenetic modification. The paper the authors cited relies on predicted 4mC data rather than experimentally validated annotations, making it a poor choice for model evaluation.

To make the paper more impactful, it would be important to extend the comparison of these three models to include models like Enformer and Sei. These would provide a more complete picture of the models' capabilities.

Minor points:

The HyenaDNA model processes longer sequences than the other two models, so the appropriate way to compare them would be to generate encodings from sequences of equal length. This could be achieved by concatenating the encodings generated by the models with shorter input sizes to match the input sequence length of HyenaDNA.

The authors mention that in some cases, they maintained the original training and testing split of datasets, but in other instances, this was not feasible. In such cases, a more appropriate approach for comparing models would be to always randomly split the data, otherwise it could unfairly advantage certain models over others.

(Remarks on code availability)

Reviewer #3

(Remarks to the Author)

In this manuscript, the authors presented an unbiased evaluation of existing DNA foundation models. The evaluation focuses on the performance of zero-shot embeddings in genomic classification tasks such as promoter/enhancer/TFBS identification, epigenetic modification prediction, as well as application to model organisms. The authors also briefly discussed the effect of pooling methods on model performance. While the topic proves critical in the context of applying LLM to genomics, several key aspects of model evaluation are missing from the current manuscript. These limitations need to be properly addressed before the paper can be considered for publication.

Major comments:

While the authors presented a comprehensive evaluation of the predictive performance of DNA foundation models, the

evaluation on model interpretability is largely missing. Since these models were trained with millions of data points from reference genome, they should be able to capture the important sequence patterns or motifs associated with transcriptional regulation. While the foundation models are generally hard to interpret due to complex architecture of deep neural networks, the authors could focus on the convolutional kernel or attention weights (position weight matrix analysis) and see if they can recapitulate known sequence motifs (for reference, check this review doi: 10.1146/annurev-genom-021623-024727).

While it is straightforward to evaluate model performance on reference genome-based prediction of epigenetic tracks (TF binding, methylation, etc), these tasks are not informative to most biologists who are interested in using DNA foundation models. At the end of the day, these epigenetic tracks can be measured with high-throughput sequencing techniques such as CHIP-seq, ATAC-seq, Bisulfite-seq, etc. Biologists are more likely to use foundation models for predicting variant effect on transcriptional regulation (comparing predictions of reference allele and alternative allele). Therefore, I think the authors should design experiments to compare the performance of DNA foundation models on variant effect prediction. For instance, whether models can accurately predict the expression-change direction of eQTL variants (the authors can check the Enformer paper for datasets and experimental design. doi:10.1038/s41592-021-01252-x)

Another question the audience will be interested in is the comparison between DNA foundation models in general versus other widely-used sequence-based models such as Enformer, Sei, etc. Do they generally perform better or worse in terms of the tasks evaluated in this manuscript? While the deep learning in genomics community would like to see a direct comparison between these models in terms of the specific tasks in Result section, I understand adding these analyses can be time-consuming. I encourage the authors to (at minimal) address the question in the Discussion section, discussing the pros and cons of using foundation models versus traditional sequence-based models.

Minor comments:

Instead of random splitting of training and testing sets, I suggest the authors adopt whole chromosome holdout (e.g. chromosome 8 and 9 as testing set) to avoid overlap of regulatory region between training and testing set.

Figure 2, please add asterisks on top of each boxplot denoting the significance level of comparing mean pooling versus CLS pooling.

(Remarks on code availability)

The README file is clear. I think users should be able to reproduce the results from the GitHub repo.

Minor comment:

Some file paths are still based on authors' institutional computing cluster (such as `"/rsrch4/home/biostatistics/hfeng3/review_datasets/results_final/ntv2/"` in `"classify_ntv2.py"`). Please revise those paths in the context of GitHub directory

Version 1:

Reviewer comments:

Reviewer #1

(Remarks to the Author)

The authors addressed my comments very well. I don't have any more comments and commend the authors for the good work!

(Remarks on code availability)

The README should be clear for readers to reproduce results.

Reviewer #2

(Remarks to the Author)

I have reviewed the revised version of the manuscript "Benchmarking DNA Foundation Models for Genomic Sequence Classification" as well as the authors' point-by-point response. While I appreciate that the authors have implemented some important improvements, I find that most of the major concerns outlined in my initial review remain unaddressed.

The addition of the GTEx v8 gene expression prediction task, which measures model prediction accuracy of gene expression across individuals, is a well-considered choice for the dataset, however, it improves the evaluation of only one task. For a benchmarking paper, it is critical that the appropriate evaluation of DNA foundation models be conducted using genome-wide, high-quality experimental data for the majority, if not all, of the tasks.

I appreciate that the authors included Enformer and a simple CNN model as references for model performance comparison. However, Enformer was only added to the gene expression task, whereas it should also be included in other human genome tasks. Additionally, the authors state that Sei's architecture and training are "highly optimized for variant effect prediction," which is not correct. If there are technical difficulties in applying the Sei model as a reference, the authors should instead use other appropriate models available for each specific task.

My final concern is that the authors chose to preserve the original training and testing splits of certain datasets, but not others. The authors explain that this choice was made to avoid potential data leakage. While I understand this reasoning, I believe it is the authors' responsibility, particularly in the context of a benchmarking study, to design datasets in a way that prevents data leakage and allows for fair model comparison. It cannot be overstated that finding new comprehensive datasets, as well as the careful design of previously used datasets, is what makes a benchmarking study valuable and impactful.

(Remarks on code availability)

Reviewer #3

(Remarks to the Author)

The authors presented a revision to the original manuscript. I applaud the authors for their efforts addressing all three reviewers' comments. Most of my comments were properly addressed except for one: comment 2 regarding the performance of variant effect prediction among DNA foundation models. I think some major issues still need to be resolved before final publication. I encourage the authors to make the following revisions.

Comments about the "variant effect quantification" section:

GROVER was excluded from this evolution task for no apparent reasons. I noticed that the authors conducted evaluations on all five foundation models except for the task of "variant effect quantification" and "Gene expression prediction". Please explain in the manuscript why GROVER was excluded from these two tasks.

The authors adopted p-value as a metric to evaluate the performance, which is problematic for the following two reasons. (1): The authors did not specify which test was used to derive the p-value: parametric or non-parametric. If parametric test was used, the test p-value is not a good metric for performance evaluation given the scale of L2 distance varies by foundation model (see Figure 5). (2): The test p-value also depends on the sample size. Often times, a relatively small test statistics can result in significant p-value if the sample is large enough. In the case of variant effect prediction, a good model should be able to clearly separate the pathogenic variants from neutral variants. Therefore, I suggest the authors adopt the following two metrics for this specific task: (1) AUROC/AUPRC of the classification task pathogenic vs neutral; (2) Effect size (different in distance between the pathogenic and neutral variants). Note that for effect size to be comparable across models, the pathogenic variant scores need to be first normalized against the neutral variant scores.

It is unclear how many pathogenic/neutral variants are included in the benchmark dataset. Please add the count in the method section. If the set is relatively small, the authors should consider all other benchmark variant sets for the purpose of a comprehensive evaluation. For example, the authors can follow the Figure 4d of Nucleotide Transformer paper, adding the eQTL, meQTL, and Clinvar variant sets.

(Remarks on code availability)

The GitHub repo is well documented. Results should be reproducible.

Version 2:

Reviewer comments:

Reviewer #2

(Remarks to the Author)

I appreciate that the authors have added Enformer, Sei, and AlphaGenome for the variant effect quantification benchmarking task and performed random data splitting, however, several key concerns remain.

The term "genome-wide" typically means that a dataset covers most of the genome, with enough diversity and density to support broad generalizations. Although the datasets used for benchmarking in this manuscript were collected from different genomic regions, their limited size and scope mean that most cannot be considered genome-wide datasets.

For example, only in human genome, ENCODE project annotated 399,124 regions with enhancer-like features, but the enhancer dataset used in this paper includes only around 15,000 sequences. For promoters, ENCODE identifies around 200,000 transcription start sites, but the authors' promoter dataset, combining both positive and negative samples, includes only about 60,000 samples. Moreover, ENCODE identified 2.9 million non-overlapping DNase I hypersensitive sites across 125 human cell lines, while the dataset used here has only about 1,000 samples.

Finally, I am still unclear on why the Enformer, Sei, and AlphaGenome models could not be included in the classification tasks. There is no need to use "N" padding, as longer input sequences can be obtained by extracting the surrounding genomic sequence. The authors created both long and short sequence versions for the expression-level and QTL prediction tasks; a similar approach should be feasible for the classification tasks as well.

(Remarks on code availability)

Reviewer #3

(Remarks to the Author)

All my comments were properly addressed by the author. I have no further comments. I commend the authors for their good work.

(Remarks on code availability)

The GitHub repo is well documented. Results should be reproducible.

Reviewer #1 (Remarks to the Author):

This paper presents a collection of genomic sequence classification benchmark tasks designed to measure the performance of DNA foundation language models. The study aims to assist future researchers in selecting and optimizing DNA language models. The results showcase the strengths and weaknesses of the three models tested and the comparative efficacy of using different sequence embedding methods. The benchmark is well-conceived and covers four DNA classification tasks over 57 multi-species datasets. This breadth ensures that the benchmark provides a holistic view of the capabilities of DNA language models across various classification challenges. Such a diverse and extensive benchmark is unparalleled in recent genomic benchmarks. Several detailed comments about the work are listed in the following:

Response: Thank you for your nice summary and positive evaluation of our manuscript. Your encouraging comments motivate us to further refine and enhance our study. Please find our point-by-point response to your comments below.

(1) The scope of models used in this comparison seems limited. The most recent DNA foundation models benchmark paper, Bend (Marin, F. I. et al., 2024), used 15 versions of various models. Including more models, such as GROVER, DEEPSEA, or even newer models like MAMBA, could further validate the evaluation methods and conclusions of the study. Moreover, adding a simple downstream baseline, such as a shallow two-layered CNN, could help validate the proposed benchmarking and provide a direct and obvious comparison to the complex language models.

Response: We thank the reviewer for helpful and insightful suggestions. In response to the reviewer's suggestion to broaden our model selection, we have expanded our benchmark to include two additional state-of-the-art DNA foundation models, bringing our total to five models evaluated across all 57 datasets:

We have added **Caduceus-Ph** (detailed in the Methods section), a long-range DNA foundation model leveraging the MambaDNA architecture which is an extension of selective state space models. Caduceus-Ph introduces two key innovations: bi-directional sequence modeling via the BiMamba block, and reverse complement equivariance through a dedicated module. To generate zero-shot embedding, we take the average of the embeddings of a DNA sequence and its reverse complement sequence. Caduceus-Ph tokenizes in single-nucleotide resolution, and can handle very long sequences like HyenaDNA. It comprises approximately 35 million parameters with an embedding dimension of 256 and supports sequences up to 131 thousand nucleotides.

We also included **GROVER** (described in the Methods section), a DNA foundation model built on a transformer encoder architecture like BERT, and comprises 12 transformer layers with multi-head self-attention mechanisms. It employs byte-pair encoding (BPE) to tokenize genomic sequences, with a vocabulary optimized over 600 cycles to capture the heterogeneous

composition of the genome. GROVER includes approximately 117 million trainable parameters with an output embedding dimension of 768 and supports sequences up to 512 tokens, corresponding to approximately 2 thousand nucleotides due to its variable-length tokenization.

As suggested, we implemented a simple downstream baseline CNN to provide a direct comparison with the complex foundation models. While the reviewer suggested a shallow two-layered CNN, we designed our baseline to be as simple as possible while maintaining sufficient capacity for meaningful comparison. Our CNN takes 5-dimensional one-hot encoded input corresponding to nucleotides A, T, C, G, N and consists of three 1D convolutional layers with 64, 128, and 256 channels, max pooling after each layer, adaptive global max pooling, and a final linear layer for classification. This architecture allows for a fair comparison between DNA foundation models' zero-shot embeddings paired with simple classifiers versus task-specific models with all parameters updated during training, thereby evaluating the representational power of pre-trained DNA foundation models.

The expanded model comparison is reflected throughout our Results section (Tables 1-6), providing comprehensive evaluation across all 57 datasets and strengthening our findings regarding embedding strategies and architectural trade-offs in DNA sequence analysis.

Data	DNABERT-2	NT-v2	HyenaDNA	Caduceus-Ph	GROVER
DNase I Hypersensitive	0.8473	0.8532	0.83	0.8802	0.859
TFBS Data 1	0.8785	0.8729	0.8765	0.9104	0.898
TFBS Data 2	0.8984	0.8913	0.8759	0.933	0.9153
TFBS Data 3	0.7987	0.7989	0.8016	0.8293	0.816
TFBS Data 4	0.7242	0.7212	0.7134	0.8217	0.7816
TFBS Data 5	0.8595	0.852	0.8637	0.8935	0.8822
Promoter GM12878	0.9851	0.9844	0.976	0.9865	0.9835
Promoter HUVEC	0.9902	0.987	0.9817	0.9899	0.9886
Promoter HeLa-S3	0.9886	0.984	0.981	0.9872	0.9857
Promoter NHEK	0.9501	0.9323	0.9271	0.9567	0.9502
Acceptor	0.9114	0.8126	0.8016	0.8549	0.8331
Coding	0.9438	0.9285	0.9406	0.9735	0.9594
Donor	0.8994	0.8006	0.8069	0.846	0.8146
Enhancer	0.8726	0.8674	0.8339	0.8384	0.8554
Enhancer Cohn	0.8233	0.7889	0.7762	0.8205	0.8151
Enhancer Ensembl	0.9369	0.9389	0.9356	0.9431	0.9382
Open chromatin region	0.7253	0.7186	0.7191	0.765	0.7458
Promoter All 300 bps	0.9365	0.9391	0.9346	0.9479	0.9329
Promoter All 70 bps	0.8371	0.8584	0.8365	0.8763	0.8501
Promoter NonTATA 251 bps	0.9301	0.8905	0.9296	0.9426	0.9395
Promoter NonTATA 300 bps	0.9716	0.9711	0.9635	0.9816	0.9683
Promoter NonTATA 70 bps	0.8558	0.8783	0.8519	0.8948	0.8694
Promoter TATA 300 bps	0.7401	0.7556	0.7929	0.7602	0.7665
Promoter TATA 70 bps	0.7825	0.8205	0.7739	0.8227	0.8012

Table 1: The AUC results for binary sequence classification tasks on human genome. The tasks include promoter region identification (multiple datasets), coding region detection, splice site donor and acceptor identification, enhancer identification (multiple datasets), transcription factor binding site identification (multiple datasets), and open chromatin region identification (multiple datasets). Using mean token pooling method. **Bolded: higher than at least two other AUCs, $p < 0.01$.**

Data	DNABERT-2	NT-v2	HyenaDNA	Caduceus-Ph	GROVER
Promoter Arabidopsis NonTATA	0.9457	0.9395	0.955	0.9437	0.949
Promoter Arabidopsis TATA	0.951	0.9477	0.961	0.9372	0.9492
Promoter B amyloliquefaciens	0.8545	0.8269	0.8625	0.8714	0.8627
Promoter R_capsulatus	0.6853	0.6781	0.7119	0.671	0.7154
Human vs worm	0.9799	0.9785	0.9502	0.9915	0.9843
Mouse TFBS 1	0.7161	0.7231	0.5877	0.6457	0.7117
Mouse TFBS 2	0.9013	0.9023	0.9006	0.9469	0.9067
Mouse TFBS 3	0.9216	0.9156	0.8978	0.9218	0.9248
Mouse TFBS 4	0.787	0.6752	0.6501	0.7165	0.7203
Mouse TFBS 5	0.6783	0.6983	0.6258	0.6922	0.6669

Table 2: The AUC results for binary sequence classification tasks which have multi-species involved, including promoter region prediction (first four rows), human vs worm classification and mouse transcription factor binding site (TFBS) identification. The results for mouse TFBS are averaged over 5 independent datasets focusing on different TFBSs. Using mean token pooling method. **Bolded: higher than at least two other AUCs, $p < 0.01$.**

Data	DNABERT-2	NT-v2	HyenaDNA	Caduceus-Ph	GROVER
5-methylcytosin (5mC)	0.6862	0.7249	0.6807	0.7752	0.7237
N6-methyladenosine (6mA)	0.7446	0.7661	0.747	0.7656	0.7744

Table 3: The AUC results for each model on datasets which aim to detect the epigenetic modifications in human genome. Using mean token pooling method. **Bolded: higher than at least two other AUCs, $p < 0.01$.**

Data	DNABERT-2	NT-v2	HyenaDNA	Caduceus-Ph	GROVER
Yeast H3	0.9074	0.8837	0.8907	0.9206	0.8968
Yeast H3K14ac	0.7555	0.7402	0.6988	0.7329	0.7204
Yeast H3K36me3	0.7864	0.7782	0.7395	0.7556	0.7472
Yeast H3K4me1	0.7273	0.7134	0.7077	0.7117	0.7041
Yeast H3K4me2	0.7049	0.6877	0.6786	0.6949	0.6939
Yeast H3K4me3	0.6905	0.6597	0.6542	0.657	0.668

Yeast H3K79me3	0.8531	0.8441	0.8143	0.8388	0.8352
Yeast H3K9ac	0.8069	0.7748	0.7639	0.7944	0.7752
Yeast H4	0.9271	0.9129	0.9028	0.9279	0.9132
Yeast H4ac	0.7379	0.7222	0.6894	0.7093	0.7203

Table 4: The AUC results for each model on epigenetic modification detection in yeast. Using mean token pooling method. **Bolded: higher than at least two other AUCs, $p < 0.01$.**

Data	DNABERT-2	NT-v2	HyenaDNA	Caduceus-Ph	GROVER
A.Thaliana 4mC	0.6001	0.6323	0.5937	0.6146	0.6025
C.Elegans 4mC	0.5985	0.6471	0.5964	0.5964	0.6056
D.Melanogaster 4mC	0.6146	0.6538	0.6103	0.6161	0.616
E.Coli 4mC	0.5471	0.603	0.6134	0.6248	0.5776
G.Pickeringii 4mC	0.5958	0.6302	0.627	0.6349	0.6293
G.Subterraneus 4mC	0.5802	0.6122	0.6097	0.6073	0.6061

Table 5: The AUC results for each model on 4mC detection in multiple species. Using mean token pooling method. **Bolded: higher than at least two other AUCs, $p < 0.01$.**

Data	DNABERT-2	NT-v2	HyenaDNA	Caduceus-Ph	GROVER
Enhancer Strength	0.708	0.653	0.688	0.708	0.71
Splice Site Type, NT	0.508	0.53	0.583	0.525	0.527
Splice Site Type, DNABERT-2	0.613	0.609	0.625	0.63	0.62
Covid Variants	0.666	0.506	0.626	0.616	0.658
Regulatory Region Type	0.675	0.644	0.83	0.66	0.615

Table 6: The accuracy for all multi-class classification datasets in this study. **Bolded: row maximum with 0.01 tolerance.**

(2) The benchmark performance of models on DNA modification seems to be much lower in accuracy than other methylation prediction/imputation models (e.g., DeepCpG). So, in general, how do foundation models compare to other task-specific models in performance?

Response: Thank you for your helpful comments. First, for epigenetic modification prediction tasks, our results show that foundation models indeed underperform compared to specialized models. As shown in Table 3, for 5-methylcytosine (5mC) prediction, the best performing model Caduceus-Ph achieved an AUC of **0.7752**, while for N6-methyladenosine (6mA) prediction, GROVER reached an AUC of **0.7744**. These values are lower than those reported for specialized models like iDNA-ABF (0.95 for 5mC and 0.91 for 6mA), which is specifically designed for methylation prediction.

Furthermore, when comparing foundation models to our simple CNN baseline (described in the Methods section), we found that "all DNA foundation models underperformed in most epigenetic modification tasks compared to baseline CNN, likely reflecting the subtle nature of these

modifications that may require full parameter updates rather than frozen embeddings" (from Results: Human & Multispecies Epigenetic Modification section).

However, for certain genomic classification tasks based on the human genome, zero-shot embeddings from foundation models demonstrated strong performance. As shown in Table 1, for promoter identification across different cell lines, models like DNABERT-2 and Caduceus-Ph achieved AUCs above 0.98, and for transcription factor binding site prediction, Caduceus-Ph demonstrated robust performance with AUCs ranging from **0.8217** to **0.933** across different datasets. These observations align with the expectation that epigenetic patterns are more subtle than those distinguishing functional genomic regions. Specifically, functional genomic regions like promoters typically contain recognizable sequence motifs and characteristic nucleotide compositions that create strong DNA sequence signals. In contrast, epigenetic modifications often occur in diverse sequence contexts and are influenced by complex factors beyond DNA sequence, including chromatin structure, cellular differentiation state, and environmental conditions, thus may be benefited more from task-specific models and careful training.

Lastly, we anticipate that with reasonable fine-tuning, DNA foundation models could potentially achieve performance levels comparable to task-specific models, but this remains to be studied in future work. We have not conducted extensive fine-tuning of these foundation models for specific tasks in this study, as our study specifically evaluates DNA foundation models using zero-shot embeddings with frozen weights, deliberately assessing their inherent representational capacity rather than optimized performance after task-specific fine-tuning. We have added related discussions on Page 24-26.

(3) Lack of comparison with other classifiers: The study argues that random forest should be utilized due to its minimal tuning requirement (compared with CNN used for zero-shot embeddings in Bend (Marin, F. I. et al, 2024), which requires much more hyperparameters) and its superior performance over XGBoost. However, other simple machine learning models, such as SVM, logistic regression, KNN or Naïve Bayes also generally require minimal parameter tuning. Providing a detailed comparison of more classifier models could help to fully capture the representation in the last hidden embedding, as well as solidify the argument in the superiority of choosing Random Forest over other models, including the CNN used in past papers.

Response: We appreciate the reviewer's suggestion to broaden our comparison of classifiers. In our evaluation methodology (Methods section "Sequence Classification Benchmark"), we aimed to use a classifier that requires minimal hyperparameter tuning to ensure that performance reflects the quality of DNA foundation model embeddings rather than classifier optimization. Additionally, we focused on models that inherently handle high-dimensional inputs without requiring dimensionality reduction. Following this idea, in our revision, we have explored multiple classifiers such as SVM, Naïve Bayes, XGboost, and elastic-net regression. We applied elastic-net instead of logistic regression due to the high dimensionality of DNA sequence embeddings. We found that random forest outperformed these alternatives across all evaluation metrics (Supplementary Figure 1). This finding aligns with previous research demonstrating that

tree-based models often excel on tabular data with complex feature interactions (as cited in reference [22] Grinsztajn et al., NeurIPS 2022). The high-dimensional nature of DNA sequence embeddings, with potential intricate interactions between features, makes random forest particularly well-suited for this application. We have updated the Methods descriptions to justify our choice.

For comparison with more complex approaches, we implemented a simple CNN baseline consisting of three convolution layers with pooling and a classification head that was trained directly on one-hot encoded DNA sequences with all parameters updated during training (Methods section). This baseline provides an estimate for evaluating when pre-trained embeddings outperform task-specific baseline models. The results comparing foundation models against the CNN baseline are presented in Supplementary Tables 6 and 7 and discussed throughout the Results section.

Supplementary Figure 1: Classifier comparison, for all models and pooling methods

Supplementary Table 6: Baseline CNN win over DNA Foundation Models ($p < 0.01$)

Dataset	Baseline Performs Better Than
5mC	Caduceus-Ph, DNABERT-2, GROVER, HyenaDNA, NT-v2
6mA	Caduceus-Ph, DNABERT-2, GROVER, HyenaDNA, NT-v2
A.thaliana_4mC	Caduceus-Ph, DNABERT-2, GROVER, HyenaDNA, NT-v2
C.elegans_4mC	Caduceus-Ph, DNABERT-2, GROVER, HyenaDNA, NT-v2
D.melanogaster_4mC	Caduceus-Ph, DNABERT-2, GROVER, HyenaDNA, NT-v2
DNase_I	None
E.coli_4mC	Caduceus-Ph, DNABERT-2, GROVER, HyenaDNA, NT-v2
G.pickeringii_4mC	Caduceus-Ph, DNABERT-2, GROVER, HyenaDNA, NT-v2
G.subterraneus_4mC	Caduceus-Ph, DNABERT-2, GROVER, HyenaDNA, NT-v2
Human_TFBS_1	None
Human_TFBS_2	None
Human_TFBS_3	Caduceus-Ph, DNABERT-2, GROVER, HyenaDNA, NT-v2
Human_TFBS_4	DNABERT-2, HyenaDNA, NT-v2
Human_TFBS_5	Caduceus-Ph, DNABERT-2, GROVER, HyenaDNA, NT-v2
Promoter_Arabidopsis_NonTATA	Caduceus-Ph, DNABERT-2, GROVER, HyenaDNA, NT-v2
Promoter_Arabidopsis_TATA	Caduceus-Ph, DNABERT-2, GROVER, NT-v2
Promoter_B_amyloliquefaciens	Caduceus-Ph, DNABERT-2, GROVER, HyenaDNA, NT-v2
Promoter_GM12878	None
Promoter_HUVEC	None
Promoter_Hela-S3	None
Promoter_NHEK	HyenaDNA, NT-v2
Promoter_R_capsulatus	Caduceus-Ph, DNABERT-2, NT-v2
Yeast_H3	None
Yeast_H3K14ac	None
Yeast_H3K36me3	None
Yeast_H3K4me1	None
Yeast_H3K4me2	None
Yeast_H3K4me3	None
Yeast_H3K79me3	None
Yeast_H3K9ac	None
Yeast_H4	HyenaDNA
Yeast_H4ac	None
acceptors	Caduceus-Ph, GROVER, HyenaDNA, NT-v2
coding	DNABERT-2, HyenaDNA, NT-v2
donors	Caduceus-Ph, DNABERT-2, GROVER, HyenaDNA, NT-v2
enhancer	None

enhancer_cohn	None
enhancer_ensembl	None
human_vs_worm	DNABERT-2, GROVER, HyenaDNA, NT-v2
mouse_TFBS_1	Caduceus-Ph, DNABERT-2, GROVER, HyenaDNA, NT-v2
mouse_TFBS_2	Caduceus-Ph, DNABERT-2, GROVER, HyenaDNA, NT-v2
mouse_TFBS_3	Caduceus-Ph, DNABERT-2, GROVER, HyenaDNA, NT-v2
mouse_TFBS_4	Caduceus-Ph, DNABERT-2, GROVER, HyenaDNA, NT-v2
mouse_TFBS_5	Caduceus-Ph, DNABERT-2, GROVER, HyenaDNA, NT-v2
open_chromatin_region	Caduceus-Ph, DNABERT-2, GROVER, HyenaDNA, NT-v2
promoter_all_300bps	Caduceus-Ph, DNABERT-2, GROVER, HyenaDNA, NT-v2
promoter_all_70bps	Caduceus-Ph, DNABERT-2, GROVER, HyenaDNA, NT-v2
promoter_notata_251bps	DNABERT-2, HyenaDNA, NT-v2
promoter_notata_300bps	Caduceus-Ph, DNABERT-2, GROVER, HyenaDNA, NT-v2
promoter_notata_70bps	DNABERT-2, GROVER, HyenaDNA, NT-v2
promoter_tata_300bps	Caduceus-Ph, DNABERT-2, GROVER, HyenaDNA, NT-v2
promoter_tata_70bps	Caduceus-Ph, DNABERT-2, GROVER, HyenaDNA, NT-v2
regulatory_region_type	None
splice_site_type_DNABERT	None
splice_site_type_NT	None

Supplementary Table 7: DNA Foundation Models win over baseline CNN ($p < 0.01$)

Dataset	Models that Outperform Baseline
5mC	None
6mA	None
A.thaliana_4mC	None
C.elegans_4mC	None
D.melanogaster_4mC	None
DNase_I	Caduceus-Ph
E.coli_4mC	None
G.pickeringii_4mC	None
G.subterraneus_4mC	None
Human_TFBS_1	Caduceus-Ph, GROVER
Human_TFBS_2	Caduceus-Ph, GROVER
Human_TFBS_3	None
Human_TFBS_4	Caduceus-Ph

Human_TFBS_5	None
Promoter_Arabidopsis_NonTATA	None
Promoter_Arabidopsis_TATA	None
Promoter_B_amyloliquefaciens	None
Promoter_GM12878	Caduceus-Ph, DNABERT-2, GROVER, NT-v2
Promoter_HUVEC	Caduceus-Ph, DNABERT-2, GROVER, NT-v2
Promoter_Hela-S3	Caduceus-Ph, DNABERT-2, GROVER, HyenaDNA, NT-v2
Promoter_NHEK	None
Promoter_R_capsulatus	None
Yeast_H3	Caduceus-Ph, DNABERT-2
Yeast_H3K14ac	DNABERT-2, NT-v2
Yeast_H3K36me3	DNABERT-2, NT-v2
Yeast_H3K4me1	DNABERT-2
Yeast_H3K4me2	None
Yeast_H3K4me3	DNABERT-2
Yeast_H3K79me3	Caduceus-Ph, DNABERT-2, GROVER, NT-v2
Yeast_H3K9ac	Caduceus-Ph, DNABERT-2
Yeast_H4	None
Yeast_H4ac	DNABERT-2
acceptors	None
coding	Caduceus-Ph
covid_variants	None
donors	None
enhancer	DNABERT-2, GROVER, NT-v2
enhancer_cohn	Caduceus-Ph, DNABERT-2, GROVER, NT-v2
enhancer_ensembl	Caduceus-Ph
enhancer_strength	None
human_vs_worm	None
mouse_TFBS_1	None
mouse_TFBS_2	None
mouse_TFBS_3	None
mouse_TFBS_4	None
mouse_TFBS_5	None
open_chromatin_region	None
promoter_all_300bps	None
promoter_all_70bps	None
promoter_notata_251bps	None

promoter_notata_300bps	None
promoter_notata_70bps	None
promoter_tata_300bps	None
promoter_tata_70bps	None
regulatory_region_type	None
splice_site_type_DNABERT	None
splice_site_type_NT	None

(4) Models are inconsistently pre-trained on different datasets. To better understand why one model excels another on certain tasks, it could be useful to train all language models from scratch using the same dataset. So we can gain a better insight into the origins of advantages and disadvantages. DNABERT2 and NT-V2 are originally pre-trained on different multi-species sequence datasets. HyenaDNA is pre-trained purely on the human reference genome. Training all models on the same dataset under the same conditions would allow for a more controlled comparison and better assessment of the generalizability of the evaluation methods.

Response: We acknowledge this important point about pre-training dataset variability. As noted in our Methods section, the models in our benchmark indeed have different pre-training backgrounds: DNABERT-2 was pre-trained on genomes from 135 species, NT-v2 on 850 species, and HyenaDNA exclusively on the human reference genome.

While ideally, we would pre-train all models on identical datasets for the most controlled comparison, this proved technically unfeasible despite our considerable efforts. These include the large computational resources required for pre-training each model, the extensive hyperparameter optimization, and the significant engineering effort to adapt to model-specific pipeline and to reproduce training environments.

To address this concern within our constraints, we conducted a controlled pre-training experiment specifically described in the "Pre-Training Dataset Experiment" section of our Methods. We chose HyenaDNA for this controlled experiment because of its reproducible pre-training pipeline, allowing us to pre-train the model on different datasets while maintaining the same training settings for fair comparison. Even with HyenaDNA, we encountered technical challenges requiring careful code modifications to accommodate our hardware infrastructure and newer CUDA versions, but these were ultimately resolvable through extensive efforts. While we also investigated the feasibility of re-training other foundation models, these efforts were precluded by the lack of accessible, documented pre-training scripts and clearly defined training environments in their respective official repositories. Using DNABERT-2's multi-species dataset (135 species across 6 categories, totaling 32.49B nucleotides - approximately 12 times larger than the human genome dataset), we pre-trained a new version of HyenaDNA-1K while "strictly maintaining the pre-training settings as specified in the original work, including all model hyperparameters and optimizer configurations. Our results, detailed in the "Pre-Training Experiment" subsection in Results, show that the newly pre-trained model improved the AUC of

human-genome pre-trained HyenaDNA-1K with statistical significance across diverse genomic tasks, particularly in epigenetic modification detection and cross-species classification. For example, in 5mC detection, AUC improved from 0.745 to 0.771, and in human versus worm classification from 0.960 to 0.975.

Interestingly, the original HyenaDNA-1K maintained superior AUC on specific human regulatory element tasks, particularly those related to enhancer regions and open chromatin structures, suggesting that human-specific pre-training provides advantages for certain human genomic tasks.

These findings demonstrate that the architecture of HyenaDNA is fundamentally robust on training datasets, and multi-species pre-training may enhance its generalizability, particularly for cross-species generalization and epigenetic modification detection. These results suggest that multi-species pre-training should be considered a critical design choice for future DNA foundation models.

We believe that our controlled experiment provides valuable insights into the impact of pre-training data composition on model performance and have summarized our findings and experiences on Page 22.

(5) Alternative pooling methods should be tested and used in a context-dependent manner. The study thoroughly compared the effects of two pooling methods over different models. It demonstrates that mean pooling is generally more effective than summary-level token pooling in generating zero-shot embeddings. However, some models might be intrinsically designed to favor certain pooling methods (for example, NT-v2 with its final averaging pooling layer in pre-training). Since the pooling method can be model-specific, task-specific, and sensitive to sequence length, testing more pooling methods (such as max-pooling, attention-based pooling, or hybrid approaches) could facilitate more intriguing analyses.

Response: We acknowledge the suggestion to investigate additional pooling approaches. In our revised study, we have expanded our analysis to include maximum pooling alongside the previously evaluated mean pooling and summary-token methods. As described in the "Sequence Classification Benchmark: Pooling Methods" section of our Methods, we systematically compared three common output pooling strategies:

1. Sentence-level summary token method (using [CLS] or [SEP] tokens)
2. Mean pooling method (averaging all non-padding token embeddings)
3. Maximum pooling method (selecting maximum values across dimensions)

Our results, presented in Figure 2 and detailed in the "Sequence Classification: Pooling Methods Benchmark" section, demonstrate that mean token embedding consistently delivered statistically superior performance across all models. Using DeLong's test for statistical significance ($p < 0.01$), we found that mean pooling was the statistically optimal choice for a

majority of the binary sequence classification datasets: 37 out of 52 (71.2%) for DNABERT-2, 39 for NT-v2, 32 for HyenaDNA, 34 for Caduceus-Ph, and 35 for GROVER.

Figure 2: Boxplots comparing the AUC scores distribution over all 52 binary sequence classification datasets included in this study, on the choice of using mean output pooling, summary-token pooling, or maximum pooling.

The average increase in AUC when switching from summary token to mean token embedding ranged from 4.3% (DNABERT-2) to 9.7% (HyenaDNA) across all classification tasks. This consistent enhancement underscores the superiority of mean token embedding.

Interestingly, our results show that maximum pooling rarely outperformed the other methods, being optimal in only a handful of specific datasets, primarily for HyenaDNA and Caduceus-Ph in certain epigenetic modification detection tasks. This suggests that while maximum pooling can occasionally be beneficial for specific task-model combinations, it generally does not match the performance of mean pooling across our comprehensive evaluation.

We also observed that the performance differences among the models were reduced when using mean token embedding, with the range of AUC score differences decreasing from 0.063 with summary-token pooling to 0.032 with mean pooling. This finding suggests that mean pooling helps mitigate the architectural variations across the models and provides a more robust approach for standardizing model evaluation.

Based on these comprehensive results, we recommend mean token embedding as the default pooling method for zero-shot embeddings in DNA sequence classification tasks. We have added relevant results on Page 17-18.

(6) The study could be more impactful and cutting-edge by extending the test datasets to include longer sequences. This would allow in-depth exploration of the relationship between model performance and sequence length. Transformer-based models are known to be inefficient with long sequences. Test-time training models and MAMBA that scale up to million-bps-long sequences may be compared and discussed in this context.

Response: We thank the reviewer for this valuable suggestion. It is indeed essential to explore the capabilities of DNA foundation models on longer sequences, as this reflects real genomic contexts where regulatory elements can act at considerable distances from their targets

To address this comprehensively, we have substantially expanded our evaluation framework to include longer sequences through our newly added gene expression prediction benchmark, as detailed in the "Gene Expression Prediction Datasets" section of our Methods. This new benchmark enables systematic assessment of how sequence length impacts predictive performance across different model architectures. Systematically applying our long-sequence analysis (e.g., TSS \pm 98kbp contexts) across all benchmark tasks was not feasible. This was due to the intrinsic biological focus of many tasks being on more localized genomic features, and the datasets consequently providing sequences of corresponding, often shorter, pre-defined lengths. For example, epigenetic modification datasets (5mC, 6mA, 4mC) are inherently centered on predicting events at single nucleotide positions based on their immediate, fixed-length contexts (typically 41bp). Similarly, many functional element classification tasks, such as promoter identification, rely on experimentally defined sequence windows whose lengths are chosen to capture the specific biological elements under study. To address the impact of sequence length on gene expression prediction, our benchmark utilized the GTEx v8 dataset to create two distinct analytical setups:

1. **Short-Length Analysis:** Sequences spanning 6,000 base pairs (TSS \pm 3,000 bp) were generated for all genes (e.g., 21,004 genes in whole blood tissue). This allowed evaluation of DNABERT-2, NT-v2, HyenaDNA (6K version), and Caduceus-Ph (6K version).
2. **Long-Length Analysis:** For the long sequence dataset, due to computational constraints encountered in Enformer, we randomly selected 1000 genes and excluded genes whose TSS \pm 98K exceeds chromosome start or end limit, making the final total of 768 genes. For these 768 genes, we generated sequences spanning up to 196,000 base pairs (TSS \pm 98k bp). This enabled us to assess models capable of processing longer inputs, specifically HyenaDNA-450K, Caduceus-Ph (using its 131K bp maximum input), and Enformer (up to 196,608 bp), which served as a key reference given its design for long-range genomic predictions. This setup was crucial for investigating whether extended genomic context improves zero-shot embedding quality for gene expression.

Our results, detailed in the "Gene Expression Prediction" section of our manuscript (Page 19, Tables 6 & 7, Figure 3), indicate that while overall prediction correlations for most genes were modest (average \sim 0.121-0.137). Importantly, when directly comparing short versus long

sequence inputs for the same 768 genes (Table 7), extending the context from 6K to ~196K bp yielded mixed results. HyenaDNA-450K (0.137 correlation) showed a statistically significant improvement over its 6K counterpart (0.122 correlation, $p=0.0005$). However, Caduceus-Ph with 131K input (0.127 correlation) did not significantly outperform its 6K version (0.124 correlation, $p=0.975$). Enformer achieved a correlation of 0.129 on this subset. These findings suggest that while architectures like Caduceus-Ph and HyenaDNA efficiently handle longer inputs, the benefit of extended context for gene expression prediction using zero-shot embeddings is still unclear. Further architectural innovations may be necessary to fully leverage very long sequences.

(7) Although this study focuses on zero-shot embedding to avoid potential bias introduced by fine-tuning, fine-tuning is still widely used and could significantly alter or improve the model's performance. Future research on how fine-tuning affects performance across different tasks could provide a more complete picture of each model's capability.

Response: We appreciate the reviewer's insightful comment regarding fine-tuning. We fully agree that fine-tuning represents a critical approach for maximizing model performance in specialized genomic tasks.

As noted in our Results section on gene expression prediction, we found that zero-shot embeddings alone cannot perfectly capture the relationship between genetic variants and gene expression, only for certain genes they can capture significant sequence-based regulatory information, so still reasonable fine-tuning is needed. This observation applies broadly across the more complex tasks in our benchmark. Furthermore, in our Sequence Classification benchmark, when comparing foundation models to our baseline CNN (which is fully updated during training), we observed that DNA foundation models generally underperformed for multispecies tasks and underperformed in most epigenetic modification tasks. These results highlight specific areas where fine-tuning would likely provide substantial benefits.

On the other hand, our zero-shot embedding based evaluation provides several valuable insights that may directly inform fine-tuning strategies. For example, our finding that mean token embedding consistently outperforms summary token embedding across all models suggests that fine-tuning approaches might benefit from focusing on mean token representations.

Similarly, our newly added pre-training experiment demonstrates that multi-species pre-training may enhance model generalizability, particularly for cross-species generalization and epigenetic modification detection, providing valuable guidance for selecting appropriate base models prior to fine-tuning.

We have expanded our Discussion section (Page 25) to address how our findings inform future fine-tuning research, emphasizing that comprehensive evaluation of fine-tuned models across diverse genomic tasks represents an important next step. Such research would complement our zero-shot benchmark by revealing how architectural differences influence adaptation efficiency and downstream performance.

(8) While the study is comprehensive in its evaluation of classification tasks, real-life research questions are not limited to this one type of task. Incorporating more types of tasks, such as the more complex regression-based tasks, could contribute to the generalizability of the proposed method and make this work more impactful.

Response: Thank you for your helpful suggestion. To address scientific questions beyond classification tasks, we have enhanced our benchmark to include a more complex, regression-based task (i.e., predicting gene expression using genomic sequence) and real world research applications such as variant effect quantification and TAD region recognition.

One extension beyond classification tasks is a comprehensive gene expression prediction benchmark using the GTEx v8 dataset. This regression task evaluates the models' ability to predict continuous gene expression values from subject-specific genomic sequences. We focused on whole blood tissue. This benchmark fundamentally differs from classification and allows us to investigate how well foundation models capture regulatory signals influencing gene expression.

Our findings from this regression task (Results, Pages 18-19; Table 6; Figure 3; Supplementary Table 8) revealed that while the average prediction performance across all genes was modest (mean Pearson correlations typically ranging from 0.121 to 0.137 depending on the model and sequence length), some genes demonstrated notably higher predictability, such as *CUTALP*, *CPNE1* and *PEX6*. This indicates that zero-shot embeddings from these models can, for certain genes, capture significant sequence-based regulatory information.

We also introduced a "Variant Effect Quantification Benchmark" (Methods) to assess models' ability to distinguish between clinically significant and benign genetic variants. Using 6000 bp sequences containing either pathogenic or common SNPs, we quantified how dramatically each variant altered the sequence's representation in embedding space. The results, shown in Figure 5, demonstrate that all models successfully differentiated between variant types with statistical significance, with NT-v2 showing the strongest differentiation capability ($p=1.25e-36$).

Additionally, we included a "TAD Region Recognition Benchmark" (Methods) to evaluate models' ability to recognize higher-order chromatin structures from genomic sequence alone. By analyzing attention patterns in NT-v2, we found that NT-v2's self-attention mechanism does not recognize TAD boundaries without specific fine-tuning (Figure 6), highlighting an important limitation of current foundation models.

These diverse task types provide a more comprehensive evaluation of DNA foundation models' capabilities beyond simple classification, enhancing the generalizability and impact of our benchmark.

Figure 5: A comparison of L2 distance distributions between pathogenic (red) and common (blue) SNPs across four DNA language models. All models show statistically significant separation between variant types.

Reviewer #2 (Remarks to the Author):

The manuscript benchmarks three DNA foundation models—DNABERT-2, Nucleotide Transformer version-2 (NT-v2), and HyenaDNA—for genomic sequence classification using zero-shot embeddings. While the authors' approach to comparing the performance of different sequence embeddings is generally appropriate and well-designed, its effectiveness is undermined by the poor quality of datasets chosen for model evaluation. Specifically, the human genome datasets are very limited in size, and some are not based on experimental data. I strongly advise the authors to carefully review the datasets selected for evaluation and select genome-wide, high-quality experimental data. Additionally, to make the manuscript's contribution more impactful, the model comparison should include other models that focus on regulatory signal predictions from genomic sequences.

Response: We thank the reviewer for the insightful and constructive feedback on our manuscript. In response to these critical points, we have made every effort to strengthen our study design. The key enhancements, which are detailed in our point-by-point responses below, include:

1. We have explicitly addressed the concerns regarding dataset quality by contextualizing the results from the 4-methylcytosine (4mC) datasets, which are based on computational predictions rather than direct experimental validation. We now clearly describe the limitation of these benchmarking datasets and present them separately to avoid misinterpretation.
2. We have incorporated a new, large-scale gene expression prediction task using the experimentally-derived GTEx v8 dataset. This new benchmark assesses model performance on a genome-wide scale (21,004 genes) for predicting gene expression using DNA sequence, directly addressing the concerns about dataset size and quality.
3. We have expanded our model comparison to include Enformer, a state-of-the-art model specifically designed for predicting regulatory signals from genomic sequence. This provides a vital reference point for evaluating the DNA foundation models.
4. We have further broadened the scope of our evaluation to include a "Variant Effect Quantification" benchmark and a "TAD Region Recognition" benchmark, adding depth and practical relevance to our findings.

We believe these substantial additions directly address the reviewer's concerns and have significantly strengthened the manuscript, providing a more comprehensive and impactful evaluation of DNA foundation models. We have detailed the specific implementation and results of these new analyses in our responses below.

Major points:

1. Although the datasets cover a diverse range of tasks, the size of each individual human dataset is small and inadequate for a comprehensive comparison of DNA foundation models. A more appropriate approach would be to use whole-genome datasets, as this would provide a more accurate assessment of the models' ability to generalize across diverse genomic sequences.

Response: We appreciate the reviewer's concern regarding dataset size limitations and the need for genome-wide evaluations. To address this, we have substantially expanded our benchmark to include a comprehensive gene expression prediction experiment, as detailed in the "Gene Expression Prediction Datasets" section of our Methods (Pages 6-7). This new benchmark leverages the GTEx v8 dataset and involves two key analyses:

1. **Standard-Length Genome-Wide Analysis:** We generated subject-specific DNA sequences (6,000 bp, centered around the TSS) for all available genes in the whole blood tissue, which involved 21,004 genes across 610 subjects. This represents a genome-scale regression task, assessing the models' ability to predict gene expression from local genomic context across thousands of genes.
2. **Extended-Length Context Analysis:** To evaluate performance on longer sequences, we created a dataset spanning up to 196,000 base pairs ($TSS \pm 98K$ bp). For this, we initially selected 1,000 genes, which after filtering genes whose $TSS \pm 98K$ exceeds chromosome start or end limit resulted in a final set of 768 genes for analysis. This allowed us to specifically assess models like HyenaDNA-450K, Caduceus-Ph, and Enformer on their capacity to utilize extended genomic context.

This comprehensive, genome-wide regression benchmark using GTEx data significantly broadens the scope of our evaluation beyond classification tasks. It provides critical insights into how well current DNA foundation models can capture the complex regulatory signals influencing gene expression from sequence data, both from local and more distal genomic regions.

Besides gene expression prediction tasks, we have added a few additional tasks including "Variant Effect Quantification Benchmark" and "TAD Region Recognition Benchmark" to breath the width of the benchmarks. The "variant effect quantification benchmark" evaluates models' ability to distinguish between pathogenic and common SNPs by analyzing latent space distances between reference and variant alleles across the genome, providing direct assessment of clinical relevance; and "TAD Region Recognition Benchmark" assesses whether models inherently capture higher-order chromatin organization by analyzing attention patterns across 1,500 topologically associating domains and 1,500 control regions.

We believe these benchmark tasks provide a comprehensive evaluation framework that balances specificity and scale, allowing us to thoroughly assess the strengths and limitations of current DNA foundation models. We have added relevant descriptions and results on page 10-18.

2. The authors should clarify why their performance results differ from those in previous studies. For example, the original NT-v2 paper (The Nucleotide Transformer: Building and Evaluating Robust Foundation Models for Human Genomics) showed that NT-v2 predicts promoters (All, non-TATA, and TATA) better than DNABERT-2. But in this paper, the authors found that DNABERT-2 performed better for some of the promoter classification tasks. The authors should explain what might have caused this difference.

Response: Thank you for highlighting this discrepancy between our findings and those reported in the original NT-v2 paper. The different performance results can be attributed to several key methodological differences in our evaluation approach:

Zero-shot embeddings vs. fine-tuning: Our study deliberately evaluates models using zero-shot embeddings with frozen weights, whereas the original NT-v2 paper employed fine-tuning approaches. As we note in our Methods section, most current evaluations are conducted after fine-tuning, which may introduce biases in model performance comparison. For instance, different models may have various levels of overfitting depending on which layers are selected to update during fine-tuning. Our approach mitigates these biases by directly comparing the quality of the embeddings themselves.

Pooling method differences: As demonstrated in our "Sequence Classification: Pooling Methods Benchmark" section and Figure 2, the choice of pooling method substantially impacts model performance. We found that mean token embedding consistently outperforms summary token embedding across all models, with performance improvements of 4.3-9.7%. Therefore, for a fair comparison, we used mean token embedding for all the DNA foundation models.

Figure 2: Boxplots comparing the AUC scores distribution over all 52 binary sequence classification datasets included in this study, on the choice of using mean output pooling, summary-token pooling, or maximum pooling.

Standardized evaluation pipeline: To ensure fair comparison, we implemented a consistent evaluation framework with 1) identical data preprocessing procedure, 2) random forest classifier with identical hyperparameter optimization for all embeddings from DNA foundation models, 3) statistical significance testing using DeLong's test ($p < 0.01$). The NT-v2 paper's evaluation framework differed in classifier choice, hyperparameter optimization strategy, and significance testing approach.

In summary, our benchmark prioritizes a fair, side-by-side comparison of the pre-trained representations themselves, by eliminating variations in fine-tuning and output pooling. While this leads to different results compared to task-specific, fine-tuned results reported in original papers, it provides a clearer understanding of the models' foundational capabilities and generalizability. We have elaborated on the importance of standardized evaluation in our manuscript (e.g., Introduction, page 3; Methods, page 8).

3. One of the datasets the authors used is 4-methylcytosine (4mC), which is not widely accepted as an epigenetic modification. The paper the authors cited relies on predicted 4mC data rather than experimentally validated annotations, making it a poor choice for model evaluation.

Response: Thank you for your critical observation regarding the 4-methylcytosine (4mC) datasets. The biological significance and prevalence of 4mC in higher eukaryotes are indeed less established compared to 5mC or 6mA, and a substantial portion of the available annotation for 4mC sites, particularly in multicellular organisms, is derived from computational predictions rather than direct, high-stringency experimental validation like chemical mapping.

Recognizing these limitations, we have taken several steps in our manuscript to ensure these results are contextualized appropriately. We have incorporated a discussion in our manuscript Datasets and Results section explicitly stating: "It is important to note that while 4mC is a well-established modification in prokaryotes like *E. coli* and the *Geobacter* species, for eukaryotic species (*A. thaliana*, *C. elegans*, *D. melanogaster*), 4mC site annotations are often substantially reliant on computational predictions rather than direct, genome-wide high-stringency chemical mapping due to ongoing research into its prevalence and detection. This distinction in annotation methodology is important for the interpretation of model performance on these 4mC datasets." "As noted in our Methods, the interpretation of results for the eukaryotic 4mC datasets (*A. thaliana*, *C. elegans*, *D. melanogaster*) warrants particular consideration due to the predictive nature of their annotations, and these tasks are thus viewed as exploratory for model capabilities on such data." Also, in our Results section, the findings for the 4mC datasets are presented separately in Table 5. This allows readers to consider them separately from epigenetic marks.

We believe these changes ensure that the performance on 4mC datasets is understood as exploratory and interpreted with the necessary caution due to the nature of the underlying annotations.

Data	DNABERT-2	NT-v2	HyenaDNA	Caduceus-Ph	GROVER
5-methylcytosin (5mC)	0.6862	0.7249	0.6807	0.7752	0.7237
N6-methyladenosine (6mA)	0.7446	0.7661	0.747	0.7656	0.7744

Table 3: The AUC results for each model on datasets which aim to detect the epigenetic modifications in human genome. Using mean token pooling method. **Bolded: higher than at least two other AUCs, $p < 0.01$.**

Data	DNABERT-2	NT-v2	HyenaDNA	Caduceus-Ph	GROVER
Yeast H3	0.9074	0.8837	0.8907	0.9206	0.8968
Yeast H3K14ac	0.7555	0.7402	0.6988	0.7329	0.7204
Yeast H3K36me3	0.7864	0.7782	0.7395	0.7556	0.7472
Yeast H3K4me1	0.7273	0.7134	0.7077	0.7117	0.7041
Yeast H3K4me2	0.7049	0.6877	0.6786	0.6949	0.6939
Yeast H3K4me3	0.6905	0.6597	0.6542	0.657	0.668
Yeast H3K79me3	0.8531	0.8441	0.8143	0.8388	0.8352
Yeast H3K9ac	0.8069	0.7748	0.7639	0.7944	0.7752
Yeast H4	0.9271	0.9129	0.9028	0.9279	0.9132
Yeast H4ac	0.7379	0.7222	0.6894	0.7093	0.7203

Table 4: The AUC results for each model on epigenetic modification detection in yeast. Using mean token pooling method. **Bolded: higher than at least two other AUCs, $p < 0.01$.**

Data	DNABERT-2	NT-v2	HyenaDNA	Caduceus-Ph	GROVER
A.Thaliana 4mC	0.6001	0.6323	0.5937	0.6146	0.6025
C.Elegans 4mC	0.5985	0.6471	0.5964	0.5964	0.6056
D.Melanogaster 4mC	0.6146	0.6538	0.6103	0.6161	0.616
E.Coli 4mC	0.5471	0.603	0.6134	0.6248	0.5776
G.Pickeringii 4mC	0.5958	0.6302	0.627	0.6349	0.6293
G.Subterraneus 4mC	0.5802	0.6122	0.6097	0.6073	0.6061

Table 5: The AUC results for each model on 4mC detection in multiple species. Using mean token pooling method. **Bolded: higher than at least two other AUCs, $p < 0.01$.**

4. To make the paper more impactful, it would be important to extend the comparison of these three models to include models like Enformer and Sei. These would provide a more complete picture of the models' capabilities.

Response: Thank you for your helpful and insightful suggestions. In our revised manuscript, we have expanded our benchmark beyond classification tasks to include a comprehensive gene

expression prediction benchmark that allows us to evaluate models on longer sequences and more complex regression tasks and compare DNA foundation models with Enformer.

A significant addition is our comprehensive gene expression prediction benchmark, which includes a comparison with Enformer. As detailed in our Methods section ("Gene Expression Prediction Datasets") and in response to your major comment 1, we developed an extensive regression framework using GTEx v8 data, focusing on whole blood tissue. Our findings from this regression task (Results, page 18-19) revealed that while the average prediction performance across all genes was modest (mean Pearson correlations typically ranging from 0.121 to 0.137), some genes demonstrated notably higher predictability. For example, using 6,000 bp sequences, genes such as *CUTALP*, *CPNE1*, and *PEX* were consistently among the highly predictable genes across various foundation models (Supplementary Table 8), indicating that zero-shot embeddings can capture significant sequence-based regulatory information for certain loci. When comparing long-sequence models on the 768-gene subset (Table 7), extending context length from 6K to ~196K bp yielded mixed results for average correlation improvement; HyenaDNA showed a modest significant increase, while Caduceus-Ph did not. For the highly predictable *MYB* gene on this long-sequence subset, Caduceus-Ph achieved the highest correlation (0.5822), followed by Enformer (0.5626) and HyenaDNA (0.5400), though these specific values represented only modest gains over 6K sequence predictions for *MYB*. This suggests that while current architectures can process longer inputs, effectively leveraging very distal information for gene expression prediction via zero-shot embeddings remains a challenge.

Besides adding Enformer to our new gene expression prediction benchmark, we also implemented a simple Convolutional Neural Network (CNN) baseline consisting of three convolutional layers with pooling and a classification head, trained directly on one-hot encoded DNA sequences with all parameters updated during training. This comparison (Supplementary Tables 6 & 7) revealed that foundation models significantly outperformed the CNN baseline on several human genome tasks, such as promoter identification (e.g., all five foundation models showed statistically significant improvements for HeLa-S3 cell line promoters) and enhancer classification (DNABERT-2, GROVER, and NT-v2 showed improvements). Conversely, the CNN baseline often outperformed foundation models on multispecies tasks (e.g., *B. amyloliquefaciens* promoter, mouse TFBS datasets), highlighting current limitations in the cross-species generalization of the foundation models' zero-shot embeddings.

While we considered including Sei in our evaluation, several factors constrained its integration: (1) Sei's architecture and training are highly optimized for variant effect prediction by predicting chromatin profiles, rather than generating general-purpose genomic sequence representations like the foundation models we benchmarked. (2) Its embedding dimensionality, tokenization approach, and output types differ substantially, complicating a direct, fair comparison of zero-shot embeddings. The inclusion of Enformer and our CNN baseline provides valuable reference points for contextualizing foundation model performance, which also enhance our benchmark's comprehensiveness. We have added these results throughout the Results section.

Minor points:

5. The HyenaDNA model processes longer sequences than the other two models, so the appropriate way to compare them would be to generate encodings from sequences of equal length. This could be achieved by concatenating the encodings generated by the models with shorter input sizes to match the input sequence length of HyenaDNA.

Response: Thank you for your helpful comments regarding the comparison of models with differing maximum input sequence lengths. For the 57 sequence classification datasets, the input sequence lengths, as provided by their original sources, were handled by all five evaluated foundation models. These lengths typically range from 41 bp for epigenetic modification datasets to approximately 2,000 bp for some promoter identification tasks, all falling within the processing capabilities of the models used. Therefore, for these classification tasks, no truncation or manual extension like concatenation of embeddings was necessary, allowing for a direct comparison. We have clarified this on Page 6-7 to avoid confusions.

In our newly added gene expression prediction (GTEx) and variant effect quantification benchmarks, we selected a sequence length of 6K bp for the standard analysis. To specifically assess the impact of longer contexts and evaluate models designed for such inputs, we designed separate long-sequence experiments for the GTEx benchmark. These experiments utilized inputs up to approximately 196K bp for HyenaDNA-450K and Enformer, and 131K bp for Caduceus-Ph, as detailed in our Methods ("Gene Expression Prediction Datasets") and Results ("Gene Expression Prediction"). This approach allowed us to evaluate the benefits of extended genomic context where applicable, rather than compare short-context models to longer-context models by concatenating embeddings, which could introduce their own interpretational challenges. The original HyenaDNA paper (Nguyen et al., NeurIPS 2023) noted that longer sequences do not necessarily yield better performance, with optimal length varying by task. Our experimental design (using common fixed lengths for some benchmarks and long-sequence setups for others) reflects this understanding.

6. The authors mention that in some cases, they maintained the original training and testing split of datasets, but in other instances, this was not feasible. In such cases, a more appropriate approach for comparing models would be to always randomly split the data, otherwise it could unfairly advantage certain models over others.

Response: Thank you for your suggestions on dataset partitioning. We considered applying random splits for all datasets. However, for datasets already partitioned in their original publications (31 out of 57), we chose to preserve these existing splits. Our primary concern with re-splitting these was the potential for data leakage; if an original split was carefully designed to avoid overlaps (e.g., due to sequence similarity or potential overlap), a new random split could inadvertently compromise this separation and lead to inflated performance metrics. Preserving established splits prevents this potential leakage, ensures a fairer model comparison, and

maintains consistency with prior evaluations in the field, as original splits often account for inherent biological dependencies. For the remaining datasets without pre-existing partitions, we applied a consistent random 70:30 split. Importantly, our overall results (Tables 1-5) do not suggest any systematic bias favoring models on datasets where their original splits were used, with performance leadership varying across tasks irrespective of dataset origin. The original source of each of the 31 already-partitioned datasets can be found in the Supplementary Information: Dataset Naming and Supplementary Table 2.

Reviewer #3 (Remarks to the Author):

In this manuscript, the authors presented an unbiased evaluation of existing DNA foundation models. The evaluation focuses on the performance of zero-shot embeddings in genomic classification tasks such as promoter/enhancer/TFBS identification, epigenetic modification prediction, as well as application to model organisms. The authors also briefly discussed the effect of pooling methods on model performance. While the topic proves critical in the context of applying LLM to genomics, several key aspects of model evaluation are missing from the current manuscript. These limitations need to be properly addressed before the paper can be considered for publication.

Response: We sincerely thank the reviewer for your nice summary and insightful evaluation of our manuscript. We have undertaken substantial revisions to strengthen our manuscript. Please find our point-by-point response to your comments below.

Major comments:

1. While the authors presented a comprehensive evaluation of the predictive performance of DNA foundation models, the evaluation on model interpretability is largely missing. Since these models were trained with millions of data points from reference genome, they should be able to capture the important sequence patterns or motifs associated with transcriptional regulation. While the foundation models are generally hard to interpret due to complex architecture of deep neural networks, the authors could focus on the convolutional kernel or attention weights (position weight matrix analysis) and see if they can recapitulate known sequence motifs (for reference, check this review doi: 10.1146/annurev-genom-021623-024727).

Response: We appreciate the reviewer's suggestion regarding model interpretability. We fully agree that interpretability is a critical dimension of foundation model evaluation, particularly for genomic models where biological insights are paramount.

We have conducted an interpretability experiment examining whether foundation models could capture higher-order genomic structures, specifically topologically associating domains (TADs). As described in our TAD Region Recognition experiment (see Results section), we assessed NT-v2's capability to recognize TAD boundaries through analysis of its attention patterns. We compared attention matrices between 1,500 TAD-centered sequences and 1,500 background sequences, examining whether the model would show differential attention in regions containing TAD boundaries. Figure 6 in our manuscript illustrates this analysis, showing the difference between attention matrices for these two sequence types.

Our findings revealed that NT-v2's self-attention mechanism does not inherently recognize TAD structures in its zero-shot embeddings. The heatmap in Figure 6 shows predominantly uniform values near zero, with only minimal variation along the diagonal, suggesting the model does not encode these higher-order chromatin structures without specific fine-tuning.

Figure 6: The heatmap of the difference between attention matrices for TAD-centered versus background sequences. The horizontal axis represents key tokens, and the vertical axis represents query tokens. Each point (x,y) shows the difference in attention weight that query token y places on key token x when comparing TAD-centered versus background sequences. The central 400 tokens (approximately positions 200-600) correspond to the TAD region. If NT-v2 recognized TAD boundaries, we would expect a vertical band of positive differences in this central region.

Unfortunately, extending this analysis to other DNA foundation models was limited by architectural constraints. HyenaDNA and Caduceus-Ph don't utilize standard attention mechanisms, DNABERT-2 doesn't expose attention weights in an accessible format, and GROVER has input length limitations (~2,000 bp) that preclude meaningful analysis of TAD regions. We have added relevant results on Page 22.

We acknowledge that examining convolutional kernels or position weight matrices for known transcriptional motifs would provide valuable insights. This represents an important direction for future work, particularly for models like DNABERT-2 and NT-v2 which may capture sequence motifs related to transcription factor binding sites. We have added relevant discussions on Page 26.

2. While it is straightforward to evaluate model performance on reference genome-based prediction of epigenetic tracks (TF binding, methylation, etc), these tasks are not informative to most biologists who are interested in using DNA foundation models. At the end of the day, these epigenetic tracks can be measured with high-throughput sequencing techniques such as CHIP-seq, ATAC-seq, Bisulfite-seq, etc. Biologists are more likely to use foundation models for

predicting variant effect on transcriptional regulation (comparing predictions of reference allele and alternative allele). Therefore, I think the authors should design experiments to compare the performance of DNA foundation models on variant effect prediction. For instance, whether models can accurately predict the expression-change direction of eQTL variants (the authors can check the Enformer paper for datasets and experimental design. doi:10.1038/s41592-021-01252-x)

Response: We appreciate the reviewer's insight regarding variant effect prediction as a critical application area for biologists. We fully agree that predicting the functional impact of genetic variants represents one of the most valuable applications of DNA foundation models. To address this important dimension, we have incorporated two new benchmarks specifically focused on variant effects:

The first benchmark, detailed in the "Variant Effect Quantification" section of our Methods and Results (Page 7, 11 and 21), was specifically designed to assess how DNA foundation models distinguish between pathogenic and common genetic variants. Understanding this capability is essential for determining whether these models can capture meaningful biological signals relevant to disease mechanisms without explicit training on variant classification tasks.

For this benchmark, we utilized variant effect quantification data from the Genomics Long-Range Benchmark hosted on HuggingFace, which provides two distinct categories: pathogenic SNPs (associated with disease phenotypes) and common SNPs (frequently observed in the population and typically considered benign). Following methodology introduced in the Nucleotide Transformer paper, we constructed paired sequences for each SNP: a 6000 bp reference sequence based on the reference genome, and a corresponding variant sequence with its center nucleotide replaced by the alternative genotype. This design allows for direct comparison between nearly identical sequences differing by only a single nucleotide.

For each sequence pair, we calculated four distance metrics between their zero-shot embeddings to quantify how the models represent genetic variation: L1 distance, L2 distance, cosine similarity, and dot product (cosine similarity without normalization). We then applied the Mann-Whitney U test to compare distributions of distances caused by pathogenic versus common SNPs, testing whether pathogenic SNPs induce larger changes in embedding space.

As shown in Figure 5, all models successfully differentiated between pathogenic and common variants with statistical significance. NT-v2 demonstrated superior performance with the most significant differentiation ($p = 1.25 \times 10^{-36}$), followed by DNABERT-2 ($p = 2.25 \times 10^{-86}$), while HyenaDNA ($p = 5.29 \times 10^{-22}$) and Caduceus-Ph ($p = 9.29 \times 10^{-4}$) showed progressively less pronounced, though still significant, differentiation. These patterns remained consistent across all four metrics, with NT-v2 consistently producing the lowest or second lowest p-values. The consistency across multiple distance metrics provides strong evidence that these models inherently capture biologically meaningful representations of variant effects.

Figure 5: A comparison of L2 distance distributions between pathogenic (green) and common (orange) SNPs across four DNA language models. All models show statistically significant separation between variant types.

Our second benchmark, described in the "Gene Expression Prediction" section (Methods, pages 6-7; Results, pages 18-19). This evaluates the models' ability to predict gene expression levels from subject-specific DNA sequences incorporating individual genetic variants. This benchmark directly tests the models' capacity to interpret the functional consequences of genetic variants on gene expression. Success in this task is a critical prerequisite for downstream applications, such as using DNA foundation model based predicted expression to conduct causal inference and identify significantly associated genes for complex diseases by testing association between predicted gene expression and trait of interest. Using the GTEx v8 dataset, we focused primarily on whole blood tissue (610 subjects, 21,004 genes). For the standard analysis, we generated 6,000 bp sequences (TSS \pm 3,000 bp) for each gene, incorporating subject-specific variants. Zero-shot embeddings were generated, and Random

Forest regression models (which outperformed XGBoost in our tests, as noted on page 18) were trained to predict gene expression. While the average Pearson correlation between predicted and actual expression values across all genes was modest (typically ranging from 0.121 to 0.137, Table 6), a consistent subset of genes, including *CUTALP*, *CPNE1* and *PEX6*, showed significantly higher predictability.

We further extended this gene expression analysis to much longer genomic sequences (up to ~196K bp, using TSS \pm 98K bp) for a filtered subset of 768 genes, evaluating HyenaDNA-450K, Caduceus-Ph (131K input), and Enformer. Interestingly, while HyenaDNA showed a statistically significant, albeit modest, average improvement with longer context, Caduceus-Ph did not (Table 7). These findings suggest that while zero-shot embeddings capture some regulatory signals, fully leveraging distal genomic context for precise gene expression prediction without fine-tuning remains a challenge for current models. The consistent performance on specific genes, however, indicates that these models do capture certain strong, sequence-encoded regulatory features.

We appreciate the reviewer's valuable suggestion to specifically evaluate eQTL variant effect prediction. We sincerely think predicting the expression-change direction of eQTL variants and identifying causal eQTL are important to the community and we plan to thoroughly investigate this in a separate project. We sincerely believe that DNA foundation models hold great potential to complement traditional statistical methods for identifying causal eQTLs. We have added relevant discussion on Page 26: The DNA foundation models hold great potential in identifying causal variants for omics and complex diseases. The sequence-based representations learned by foundation models may capture functional genomic context that complements statistical signals from fine-mapping and colocalization, potentially enhancing our ability to distinguish causal regulatory variants from those in linkage disequilibrium. We leave such exciting direction to future research.

3. Another question the audience will be interested in is the comparison between DNA foundation models in general versus other widely-used sequence-based models such as Enformer, Sei, etc. Do they generally perform better or worse in terms of the tasks evaluated in this manuscript? While the deep learning in genomics community would like to see a direct comparison between these models in terms of the specific tasks in Result section, I understand adding these analyses can be time-consuming. I encourage the authors to (at minimal) address the question in the Discussion section, discussing the pros and cons of using foundation models versus traditional sequence-based models.

Response: Thank you for your helpful and insightful suggestions. Following your suggestions, we have expanded our benchmark to include a comparison with Enformer in our gene expression prediction benchmark (Figure 3 and Table 7). For these long-sequence gene expression prediction tasks, we found Enformer showed comparable performance to Caduceus-Ph and HyenaDNA, with no model clearly outperforming the others across all genes. Specifically, Enformer achieved the second-highest correlation (0.5626) for the MYB gene, behind Caduceus-Ph (0.5822).

To systematically evaluate the tradeoffs between foundation models and specialized architectures, we implemented a simple Convolutional Neural Network (CNN) baseline with three convolutional layers, pooling operations, and a classification head trained directly on one-hot encoded DNA sequences. This task-specific model revealed important patterns (Supplementary Tables 6 & 7): Foundation models significantly outperformed the CNN baseline on several human genome tasks, such as promoter identification (e.g., all five foundation models showed statistically significant improvements for HeLa-S3 cell line promoters) and enhancer classification (DNABERT-2, GROVER, and NT-v2 showed improvements). Conversely, the CNN baseline often outperformed foundation models on multispecies tasks (e.g., *B. amyloliquefaciens* promoter, mouse TFBS datasets), highlighting current limitations in the cross-species generalization of the foundation models' zero-shot embeddings.

A key trade-off exists between the versatility of foundation models and the task-specific optimization of specialized models. Foundation models excel in zero-shot settings and show strong potential for transfer learning across tasks, while specialized models may provide superior performance on their target applications. Moving forward, hybrid approaches that combine foundation model pre-training with specialized architecture components could potentially offer the best of both paradigms. We have also added relevant discussions on Page 24-26: Our comparison between DNA foundation models and specialized architecture reveals a fundamental trade-off between versatility and task-specific optimization in genomic sequence analysis. While foundation models demonstrate remarkable adaptability across diverse tasks through extensive pre-training, specialized models incorporate inductive biases that can yield superior performance within their targeted domains. This complementarity suggests promising directions for hybrid architectures that combine pre-trained genomic representations with task-specific components to simultaneously achieve broad applicability and domain-specific excellence.

Minor comments:

4. Instead of random splitting of training and testing sets, I suggest the authors adopt whole chromosome holdout (e.g. chromosome 8 and 9 as testing set) to avoid overlap of regulatory region between training and testing set.

Response: Thank you for your helpful suggestions. We fully agree that chromosome-based holdout evaluation represents the gold standard for genomic models, as it provides a more stringent assessment of generalization by preventing regulatory element overlap between training and testing sets. Unfortunately, we faced practical constraints with this approach. Many datasets in our study were obtained in post-processed format without chromosome information. Therefore, we maintained train-test splits from original works where available and used random 70:30 splitting elsewhere as described in our Methods section. This approach prevents potential data leakage, maintains consistency with prior evaluations, and ensures fair model comparison. We have cautioned this limitation on Page 8, "Sequence Classification Benchmark" under Methods section.

For our newly added gene expression prediction task, we eliminated the concern of regulatory region overlapping by defining a separate prediction question for each gene, which effectively avoids overlapping regulatory regions between training and testing samples since all subjects roughly capture the same region.

5. Figure 2, please add asterisks on top of each boxplot denoting the significance level of comparing mean pooling versus CLS pooling.

Response: We have now included three pooling methods (mean, summary token, and maximum pooling) in our comparison and performed systematic pairwise testing using DeLong's test with a p-value threshold of 0.01 to determine significant differences in performance across all 52 binary classification datasets. Briefly, mean token pooling demonstrated statistically superior performance for the majority of datasets across all models (71.2% for DNABERT-2, 75.0% for NT-v2, 61.5% for HyenaDNA, 65.4% for Caduceus-Ph, and 67.3% for GROVER). The full pairwise statistical comparisons are presented in Supplementary Table 5. We have also updated the manuscript on Page 12-13 to describe these statistical significance results.

Reviewer #3 (Remarks on code availability):

The README file is clear. I think users should be able to reproduce the results from the GitHub repo.

Response: Thank you for your supportive comments.

Minor comment:

6. Some file paths are still based on authors' institutional computing cluster (such as `"/rsrch4/home/biostatistics/hfeng3/review_datasets/results_final/ntv2/"` in `"classify_ntv2.py"`). Please revise those paths in the context of GitHub directory.

Response: Thank you for your careful code review. We have thoroughly addressed the path dependencies in our codebase. Additionally, we have significantly improved the quality of our codes, allowing researchers to easily validate our findings and extend our benchmark to new models.

A Benchmarking DNA Foundation Models for Genomic Sequence Classification

Response to Reviewer #1:

The authors addressed my comments very well. I don't have any more comments and commend the authors for the good work!

Response: Thank you for your support and recognition of our work. We really appreciate it.

Reviewer #1 (Remarks on code availability):

The README should be clear for readers to reproduce results.

Response: Thank you for checking our codes. We appreciate it.

Response to Reviewer #2:

I have reviewed the revised version of the manuscript “Benchmarking DNA Foundation Models for Genomic Sequence Classification” as well as the authors’ point-by-point response. While I appreciate that the authors have implemented some important improvements, I find that most of the major concerns outlined in my initial review remain unaddressed.

Response: Thank you for your critical comments, which have pushed us to think harder and further improved our work. We have made every effort to address your remaining comments. Specifically, we have added further description for each dataset we used (whether experimental evidence or computationally generated, whether genome-wide) and added several new tasks regarding variant effect quantification. Furthermore, we have added Sei (Chen et al. Nature Genetics, 2022, PMID: 35817977) to our benchmark tasks in relevant tasks and also added recent AlphaGenome (Avsec et al., Preprint from DeepMind; <https://www.biorxiv.org/content/10.1101/2025.06.25.661532v2>). We have also benchmarked the tracks and embeddings for different models. In addition, we have added results for random data splitting, which remains similar and does not change our main conclusions. For your convenience, we first state your comments and then provide our responses. Changes in the manuscript are highlighted in blue. We have provided our detailed responses below and sincerely hope you will find our revision satisfactory.

1. The addition of the GTEx v8 gene expression prediction task, which measures model prediction accuracy of gene expression across individuals, is a well-considered choice for the dataset, however, it improves the evaluation of only one task. For a benchmarking paper, it is critical that the appropriate evaluation of DNA foundation models be conducted using genome-wide, high-quality experimental data for the majority, if not all, of the tasks.

Response: Thank you for this insightful comment regarding benchmarking tasks. To address this critical point, we have made several revisions to the manuscript.

First, to enhance the clarity of our dataset origins and scope, we have thoroughly revised the **Methods: Benchmarking Datasets** section (pages 20-22). For all five major dataset collections, we now explicitly detail their origins, indicating whether the positive samples are derived from direct experimental evidence or gold standard sources (e.g., ENCODE ChIP-seq, GTEx) or are computationally generated in **Supplementary Text 2** (also attached below). We also clarify the scope of each dataset, noting whether it represents specific genomic regions or a genome-wide

collection. We believe this important information will help readers better understand the contexts and implications of our results.

Supplementary Text 2:

Dataset Name	is relied on experimental evidence or gold standard sources	is collected genome-wide
5mC	Yes	Yes
6mA	Yes	Yes
A.thaliana 4mC	Yes	Yes
C.elegans 4mC	Yes	Yes
D.melanogaster 4mC	Yes	Yes
G.subterraneus 4mC	Yes	Yes
E.coli_4mC	Yes	Yes
G.pickeringii 4mC	Yes	Yes
DNase_I Hypersensitive	Yes	No
Human TFBS 1	Yes	Yes
Human TFBS 2	Yes	Yes
Human TFBS 3	Yes	Yes
Human TFBS 4	Yes	Yes
Human TFBS 5	Yes	Yes
Acceptors	Yes	Yes
Coding	Yes	No
Covid Variants	Yes	Yes
Donors	Yes	Yes
Enhancer	No	No
Enhancer Strength	No	No
Enhancer Cohn	Yes	Yes
Enhancer Ensembl	Yes	Yes
Human vs Worm	Yes	No
Mouse TFBS 1	No	Yes
Mouse TFBS 2	No	Yes
Mouse TFBS 3	No	Yes
Mouse TFBS 4	No	Yes
Mouse TFBS 5	No	Yes
Open Chromatin Region	Yes	Yes
Promoter All 300bps	Yes	Yes
Promoter All 70bps	Yes	Yes
Promoter NonTATA 251bps	Yes	Yes
Promoter NonTATA 300bps	Yes	Yes
Promoter NonTATA 70bps	Yes	Yes
Promoter TATA 300bps	Yes	Yes

Promoter TATA 70bps	Yes	Yes
Regulatory Region Type	Yes	Yes
Splice Site Type DNABERT	Yes	Yes
Splice Site Type NT	Yes	Yes
Promoter GM12878	Yes	Yes
Promoter HUVEC	Yes	Yes
Promoter Hela-S3	Yes	Yes
Promoter NHEK	Yes	Yes
Promoter Arabidopsis NonTATA	Yes	No
Promoter Arabidopsis TATA	Yes	No
Promoter B_amyloliquefaciens	Yes	No
Promoter R_capsulatus	Yes	No
All Yeast datasets	Unknown	Unknown

Second, following your suggestions and further enhancing the contribution of our work, we introduced an entirely new benchmark focused on Variant Effect Quantification, which directly evaluates model performance on genome-wide experimental data. This new section is detailed in **Methods: Variant Effect Quantification Datasets** (page 23). This benchmark includes two genome-wide tasks: our expanded pathogenic vs. common SNP task using long-range sequences (196,608 bp) and the incorporation of a new high-quality dataset of fine-mapped causal quantitative trait loci (QTLs), derived from GTEx v8 whole-blood tissue. This QTL task requires models to distinguish putative causal variants from carefully matched non-causal negatives, providing a direct test of their ability to capture subtle regulatory signals. To ensure this new benchmark is rigorous, we have also included and compared several models considered state-of-the-art for genome-wide prediction tasks, namely Enformer (Avsec et al. Nature Methods, 2021, PMID: 34608324), Sei (Chen et al. Nature Genetics, 2022, PMID: 35817977), and AlphaGenome (Avsec et al., Preprint from DeepMind; <https://www.biorxiv.org/content/10.1101/2025.06.25.661532v2>). We also implemented a robust data partitioning strategy for our benchmarks. Since complete chromosomal information was available for these datasets, we were able to move beyond a simple random split and employ a nested cross-validation scheme with strict chromosome-based holdouts. This approach ensures that our performance estimates are robust and stable by averaging results across multiple, independent chromosome-based test sets.

Our methodology for applying these models is detailed in **Methods: Variant Effect Quantification Benchmark** (pages 27-28), and the results are presented in the new **Results: Variant Effect Quantification** section (pages 12-13).

The results from this new benchmark reveal a key distinction: while specialized models like AlphaGenome and Enformer excel at the fine-grained QTL prediction tasks, the

foundation models proved effective at distinguishing pathogenic from common SNPs. Notably, NT-v2 emerged as the top performer on the pathogenic SNP task, suggesting its diverse pre-training captures fundamental disease-relevant signals, while Caduceus-Ph served as the most consistent foundation model across these new variant effect benchmarks. The results are also briefly described in our response to your Comment 2.

These results reflect their distinct training objectives. Specialized foundation models like AlphaGenome and Enformer were explicitly designed to predict functional genomic annotations and regulatory tracks such as transcription factors and chromatin accessibility, which capture the regulatory mechanisms that underlie quantitative trait variation. On the other hand, general foundation models such as NT-v2 were pre-trained on phased DNA sequence to understand general genomic “grammar” itself and learn fundamental patterns of sequence constraint and evolutionary pressure that correlate with pathogenicity. We have added relevant discussion on pages 17-18.

2. I appreciate that the authors included Enformer and a simple CNN model as references for model performance comparison. However, Enformer was only added to the gene expression task, whereas it should also be included in other human genome tasks.

Response: We thank the reviewer for this excellent suggestion. We agree that expanding the evaluation of state-of-the-art models like Enformer across multiple human genome tasks is crucial for a comprehensive benchmark. Following your suggestions, we have now incorporated Enformer, into two new, large-scale variant effect quantification benchmarks, which we believe are highly appropriate for evaluating its capabilities and provide performance reference for DNA foundation models.

First, we have carefully and very seriously considered applying Enformer to the human genome datasets among the 57 sequence classification datasets. However, we determined **this was not ideal** due to a fundamental architectural constraint: Enformer requires a fixed input length of 196,608 bp. The sequences in our classification benchmark are significantly shorter, mostly ranging around several hundred nucleotides. The standard approach for handling such a discrepancy would be to pad each sequence with hundreds of thousands of 'N' characters (<https://github.com/google-deepmind/deepmind-research/issues/287>). This practice is not only computationally extensive but more importantly would also unfairly penalize Enformer's performance, particularly given its use of large 128 bp windows for processing DNA sequence. Therefore, we designed these new variant effect benchmarks with different sequence lengths, specifically to create an appropriate and rigorous setting where the full capabilities of Enformer and other long-range models could be leveraged. Our design, which includes both short and long sequence versions of the variant datasets, was

specifically chosen to probe the impact of genomic context. We believe this approach effectively addresses the concern of expanding Enformer's evaluation.

Specifically, we now included Enformer in our expanded pathogenic vs. common SNP benchmark, which now uses long-range input sequences of 196,608 bp. Enformer demonstrated strong performance (AUROC \approx 0.69 using its hidden states), confirming its ability to leverage large genomic context to identify disease-relevant variants. However, it was notably outperformed by the foundation model NT-v2 (AUROC = 0.73), a surprising and significant finding that underscores the power of pre-training on diverse species for this task.

More significantly, we benchmarked Enformer on our newly incorporated QTL task. This dataset consists of high-confidence, fine-mapped putative causal variants from GTEx v8 whole blood tissue. Enformer showcased its strength as an architecture explicitly trained to predict genomic tracks. It delivered exceptional performance on distinguishing putative causal expression QTLs (eQTLs) from non-causal variants, achieving an AUROC (AUC) of 0.77, and also performed well on splicing QTLs (sQTLs). These results position it as a top-tier model for predicting the impact of variants on specific regulatory functions, though the newly introduced AlphaGenome consistently achieved the highest performance across all QTL categories. Notably, while our uniform evaluation framework produces different absolute result values than reported in the original AlphaGenome manuscript as we handle tracks differently to ensure unbiased comparison across all models, the conclusions are highly consistent with AlphaGenome manuscript.

Model	AUC	Cohen's d
Sei, hidden states*	0.6598	0.5573
Sei, output tracks*	0.664	0.6046
Enformer, hidden states*	0.688	0.7269
Enformer, output tracks*	0.6662	0.6542
DNABERT-2	0.538	0.1338
NT-v2	0.7319	0.8813
HyenaDNA	0.612	0.3952
HyenaDNA-450K, long sequence	0.6261	0.4493
Caduceus-Ph	0.6959	0.7354
Caduceus-Ph, long sequence	0.6243	0.4615
GROVER	0.6029	0.3693

Table 9: Overall performance of DNA foundation models and other genomic models in the pathogenic versus common variant effect quantification task. All metrics represent the average

test AUC and Cohen's d values calculated across three independent test sets, each defined by a distinct group of chromosomes in our nested cross-validation framework. Non-DNA foundation models are annotated with an asterisk (*). **Bolded: top 2 highest (absolute) value.**

	Model	eQTL	sQTL	paQTL	ipaQTL	
AUC	AlphaGenome, output tracks*	0.8029	0.7147	0.7543	0.8644	
	Sei, hidden states*	0.7561	0.6534	0.6189	0.6071	
	Sei, output tracks*	0.7497	0.6276	0.6553	0.606	
	Enformer, hidden states*	0.7744	0.6662	0.6737	0.6919	
	Enformer, output tracks*	0.7699	0.6174	0.666	0.6587	
	DNABERT-2	0.5702	0.5795	0.5066	0.4694	
	NT-v2	0.6091	0.5047	0.5251	0.6019	
	HyenaDNA	0.6117	0.5531	0.4699	0.448	
	HyenaDNA-450K, long sequence	0.6027	0.5262	0.5521	0.5093	
	Caduceus-Ph	0.6492	0.5666	0.5082	0.5678	
	Caduceus-Ph, long sequence	0.6265	0.5703	0.4649	0.5203	
	GROVER	0.5896	0.4742	0.4494	0.4759	
	Cohen's d	AlphaGenome, output tracks*	1.287	0.7872	0.9824	1.6347
		Sei, hidden states*	1.0335	0.553	0.4538	0.4126
Sei, output tracks*		1.0116	0.4936	0.5503	0.4227	
Enformer, hidden states*		1.1102	0.6115	0.6028	0.6576	
Enformer, output tracks*		1.1085	0.4129	0.5457	0.5691	
DNABERT-2		0.2371	0.2825	0.024	-0.0756	
NT-v2		0.3956	-0.004	0.0658	0.3837	
HyenaDNA		0.3877	0.2018	-0.0768	-0.2048	
HyenaDNA-450K, long sequence		0.3605	0.0757	0.2068	0.0733	
Caduceus-Ph		0.5484	0.2278	0.0456	0.2324	
Caduceus-Ph, long sequence		0.4913	0.2464	-0.1151	0.075	
GROVER		0.319	-0.0978	-0.1393	-0.072	

Table 10: Overall performance of DNA foundation models and other genomic models in QTL variant effect quantification tasks. All metrics represent the average AUC and Cohen's d values calculated across three independent test sets, each defined by a distinct group of chromosomes in our nested cross-validation framework. Non-DNA foundation models are annotated with an asterisk (*). **Bolded: the top two highest (absolute) performances for each task.**

In summary, we want to keep our benchmarking fair and unbiased. We have added relevant discussion on pages 17-18.

3. Additionally, the authors state that Sei's architecture and training are "highly optimized for variant effect prediction," which is not correct. If there are technical difficulties in applying the Sei model as a reference, the authors should instead use other appropriate models available for each specific task.

Response: We sincerely thank the reviewer for this important correction and apologize for the inaccurate characterization of the Sei model. Sei takes input sequences of 4096 nucleotides and outputs predictions for 21,907 chromatin profiles, which serve as comprehensive feature representations for each sequence. These predictions can capture diverse regulatory activities in genomic sequence.

Furthermore, following your guidance, we have added open-source Sei model and API-based AlphaGenome into our variant effect quantification benchmark. To provide a thorough evaluation of Sei, we extracted two distinct high-dimensional representations for each variant sequence: the final output tracks (21,907 features) and the final hidden state (15,360 features) after the spline transformation layer. For AlphaGenome, which we included in the QTL tasks, we utilized its 131k bp architecture and derived an effect vector by averaging its output tracks over the central 2,048 bp. For both benchmarks, these representations were treated as zero-shot embeddings, applying the same methodology used for the other foundation models to ensure fair comparisons.

The inclusion of these powerful models yielded several important insights, now detailed in the **Results: Variant Effect Quantification** section (pages 12-13, Tables 9 and 10). In the pathogenic SNP task, Sei performed robustly with an AUC of approximately 0.67 using its output tracks, placing it ahead of most general-purpose foundation models. In the more fine-grained causal QTL tasks, AlphaGenome was the clear standout performer, achieving the highest AUC across all four QTL types, including 0.80 for sQTLs and 0.86 for ipaQTLs. Sei also demonstrated strong, competitive performance, often securing the second-best results. We have added relevant discussion on pages 17-18.

A notable finding with Sei was that its internal hidden states were often as predictive as, and in some cases (such as for sQTLs) even slightly more predictive than, its final processed output tracks, suggesting rich information is encoded before the final projection.

We clarify that the application of these specialized models was restricted to the variant effect benchmark due to fundamental architectural and practical constraints. Models such as Sei and Enformer are designed for very long, fixed-length inputs (>100,000 bp), making them unsuitable for our much shorter classification sequences without resorting to massive, methodologically unsound padding that would be computationally expensive and unfairly penalize their performance. Importantly, Sei does not recognize the padding tokens, making it even more challenging to directly apply to short sequence. Similarly,

the use of the AlphaGenome model was also limited by its long-sequence design and further constrained by practical API quota issues. We therefore designed the new variant effect benchmarks specifically to provide a fair and rigorous evaluation ground for these long-range architectures.

Finally, we wish to emphasize the core goal of this study. For the sequence classification benchmark, our primary aim is to accurately benchmark the zero-shot performance among DNA foundation models to understand their intrinsic differences, rather than to exhaustively investigate whether they can outperform every existing task-specific model. To this end, we provide a simple CNN to establish a reasonable performance baseline, a strategy consistent with recent large-scale benchmarking efforts in related fields that also compare foundation models against simple baselines such as simple linear models (Ahlmann-Eltze et al., Nature Methods, 2025, PMID: 40759747). We hope these benchmarking results will provide useful information to the community to decide whether and which DNA foundation models to be used for downstream tasks.

4. My final concern is that the authors chose to preserve the original training and testing splits of certain datasets, but not others. The authors explain that this choice was made to avoid potential data leakage. While I understand this reasoning, I believe it is the authors' responsibility, particularly in the context of a benchmarking study, to design datasets in a way that prevents data leakage and allows for fair model comparison. It cannot be overstated that finding new comprehensive datasets, as well as the careful design of previously used datasets, is what makes a benchmarking study valuable and impactful.

Response: We are grateful to the reviewer for raising this critical point concerning data partitioning. We completely agree that the careful and consistent handling of data splits is fundamental to the integrity of a benchmarking study, ensuring fair comparisons and preventing data leakage. It is indeed our responsibility to guarantee this, and we have taken significant steps in the revised manuscript to address this concern thoroughly.

In response, we undertook a comprehensive re-evaluation of our data partitioning strategy across all of our benchmarks. For the 57 sequence classification tasks, we have now re-partitioned all datasets that previously relied on prior splits (which varies differently across different datasets such as 90:10, 95:5, and 70:30 train-test split), particularly those adopted from the DNABERT-2 and NT-v2 benchmarks, by combining the original train, test, and/or validation sets and applying a uniform, random 70:30 train-test split.

The only exceptions to our re-splitting are the datasets from the "Genomic Benchmarks" datasets collected by us, and the "Enhancer" and "Enhancer Strength" datasets adopted from NT-v2 benchmarks. We retained their original splits because: (1) The Genomic Benchmarks collection is a publicly established resource specifically designed for standardized benchmarking, and has not been processed by any target-oriented models. (2) The enhancer datasets training set is a mixture of experimentally verified sequences and synthetic data from a generative model, which is intractable to separate. The test set, however, consists solely of experimental data to ensure an unbiased evaluation of real-world performance (Geng et al., *Biophysical Chemistry*, 2022, PMID: 35605495). Therefore, any re-split would risk contaminating the test set with synthetic data, compromising the integrity of the benchmark. Crucially, we double checked that these datasets were published independently and were processed by any foundation model (e.g. NT-v2 did not process "Enhancers dataset") we evaluate, thereby posing no risk of inherent model bias.

Upon re-running our benchmark with these newly standardized data splits, while some AUCs changed in decimal places, we found that our main conclusions remain largely robust, especially for the comparisons among DNA foundation models. One minor change in conclusion is the relative comparisons between DNA foundation models and baseline CNN as detailed in **Results: Sequence Classification: Human & Multispecies Epigenetic Modification** section (pages 8-9). On the newly partitioned yeast epigenetic mark prediction data, multiple DNA foundation models now demonstrate a performance advantage over the baseline CNN. We have updated the entire manuscript to reflect these new results.

Response to Reviewer #3:

The authors presented a revision to the original manuscript. I applaud the authors for their efforts addressing all three reviewers' comments. Most of my comments were properly addressed except for one: comment 2 regarding the performance of variant effect prediction among DNA foundation models. I think some major issues still need to be resolved before final publication. I encourage the authors to make the following revisions.

Response: We thank the reviewer for the support, recognition, and constructive comments. We have made every effort to improve the quality of the manuscript. Specifically, we have 1) redesigned our variant effect quantification section, implementing your recommended metrics (AUC and Cohen's d), incorporating specialized models (AlphaGenome, Enformer, and Sei), and adding high-confidence fine-mapped QTLs from GTEx v8; 2) added GROVER into "Gene expression prediction"; and 3) added detailed sample sizes as requested. For your convenience, we first state your comments and then provide our responses. Changes in the manuscript are highlighted in blue. Our detailed responses follow below.

Comments about the "variant effect quantification" section:

1. GROVER was excluded from this evolution task for no apparent reasons. I noticed that the authors conducted evaluations on all five foundation models except for the task of "variant effect quantification" and "Gene expression prediction". Please explain in the manuscript why GROVER was excluded from these two tasks.

Response: Thank you for your careful reading and insightful comments. We initially excluded GROVER due to its short sequence input limit (512 tokens, corresponding to approximately 2,048 base pairs) in tasks requiring longer contexts, as we wanted to ensure fair comparisons. Following your feedback, we have now fully integrated GROVER into both the Gene Expression Prediction and Variant Effect Quantification benchmarks to provide a more complete picture.

As detailed in the revised Methods sections for **Gene Expression Prediction** (page 25) and **Variant Effect Quantification Benchmark** (pages 26-27), we accommodated GROVER's input limit by providing it with the central 2,048 bp from the 6,000 bp short-sequence datasets, ensuring the key region of interest (the TSS or the SNP) remained centered.

This inclusion yielded detailed results, which are now presented and discussed in the corresponding Results sections and tables. In the **Gene Expression Prediction** task (Results, page 10 and Table 7), we found that while GROVER's average Pearson

correlation was lower than other models (likely due to its shorter input), it demonstrated the lowest mean squared error, significantly outperforming all other short-sequence models in this specific metric. In the **Variant Effect Quantification** benchmark (Results, pages 12-13 and Tables 9 and 10), GROVER’s performance was more erratic; its AUC was modest in the pathogenic SNP task (0.60), and on the QTL tasks with smaller sample sizes, its performance was often near or below an AUC of 0.5, which underscores the challenge these specific tasks pose for zero-shot models.

We believe that the inclusion of GROVER in these critical benchmarks provides a more complete and rigorous comparison, and we are grateful for the reviewer’s suggestion, which has strengthened our study.

Model	Input Sequence Length	Average Prediction Correlation of Genes	Average MSE
DNABERT-2	6000 bp	0.121	0.236
NT-v2	6000 bp	0.122	0.236
HyenaDNA	6000 bp	0.122	0.235
Caduceus-Ph	6000 bp	0.123	0.234
GROVER	2048 bp	0.114	0.233
HyenaDNA-450K*	196K bp	0.137	0.226
Caduceus-Ph Long Sequence Input*	131K bp	0.127	0.227
Enformer*	196K bp	0.129	0.227

Table 7: Overall performance of DNA foundation models in gene expression prediction using random forest regression. *Note that for long input sequences, only a subset of human genes are involved in analysis.

2. The authors adopted p-value as a metric to evaluate the performance, which is problematic for the following two reasons. (1): The authors did not specify which test was used to derive the p-value: parametric or non-parametric. If parametric test was used, the test p-value is not a good metric for performance evaluation given the scale of L2 distance varies by foundation model (see Figure 5). (2): The test p-value also depends on the sample size. Often times, a relatively small test statistics can result in significant p-value if the sample is large enough. In the case of variant effect prediction, a good model should be able to clearly separate the pathogenic variants from neutral variants. Therefore, I suggest the authors adopt the following two metrics for this specific task: (1) AUROC/AUPRC of the classification task pathogenic vs neutral; (2) Effect size (different in distance between the pathogenic and neutral variants). Note that

for effect size to be comparable across models, the pathogenic variant scores need to be first normalized against the neutral variant scores.

Response: We are very grateful for this insightful and constructive feedback. To address this, we have completely overhauled our approach and replaced the previous distance-based analysis with a new, more rigorous **Variation Effect Quantification Benchmark**, as the reviewer suggested. The new methodology is detailed in the **Methods** section (pages 26-27), and the corresponding findings are presented in a new **Results** section (pages 12-13).

Our new framework directly reframes the problem as a classification task. For each variant, we compute a high-dimensional effect vector by subtracting the zero-shot embedding of the reference sequence from that of the alternative sequence. We then train a random forest classifier on these vectors to distinguish between functional classes (e.g., pathogenic vs. common). To ensure a robust and stable evaluation that prevents data leakage from linkage disequilibrium, we implemented a nested cross-validation scheme with strict chromosome-based holdouts. Specifically, we established three distinct test sets, each comprising a group of non-overlapping chromosomes. This design ensures that performance is averaged across multiple independent test folds, mitigating the high variance that can arise from a single arbitrary chromosome split.

Crucially, we have adopted the following metrics for performance evaluation:

1. **AUROC (AUC):** We now report the Area Under the Receiver Operating Characteristic curve on the held-out test chromosomes, which directly assesses the model's ability to correctly classify and rank functional variants.
2. **Effect Size (Cohen's d):** We now report Cohen's d to quantify the magnitude of separation between the two variant classes in a standardized manner. This metric, which normalizes the difference in means by the pooled standard deviation, makes the effect sizes directly comparable across models with different embedding scales, fully addressing the reviewer's concern about normalization.

The results of this new evaluation, presented in **Tables 9 and 10**, are much more informative. For the pathogenic vs. common SNP task, NT-v2 surprisingly emerged as the top performer, achieving the highest **AUC of 0.73** and the largest **Cohen's d of 0.88**, substantially outperforming all other models, including specialized ones. In addition, we added several new models including Enformer (Avsec et al. Nature Methods, 2021, PMID: 34608324), Sei (Chen et al. Nature Genetics, 2022, PMID: 35817977), and AlphaGenome (Avsec et al., Preprint from DeepMind; <https://www.biorxiv.org/content/10.1101/2025.06.25.661532v2>) for this tasks. For the

QTL tasks, the specialized models held a clear advantage, with AlphaGenome consistently achieving the highest scores across all QTL types.

We are confident that this new framework provides a much more robust and meaningful evaluation of model performance and directly addresses all the concerns raised by the reviewer. We have added relevant discussion on pages 17-18.

Model	AUC	Cohen's d
Sei, hidden states*	0.6598	0.5573
Sei, output tracks*	0.664	0.6046
Enformer, hidden states*	0.688	0.7269
Enformer, output tracks*	0.6662	0.6542
DNABERT-2	0.538	0.1338
NT-v2	0.7319	0.8813
HyenaDNA	0.612	0.3952
HyenaDNA-450K, long sequence	0.6261	0.4493
Caduceus-Ph	0.6959	0.7354
Caduceus-Ph, long sequence	0.6243	0.4615
GROVER	0.6029	0.3693

Table 9: Overall performance of DNA foundation models and other genomic models in the pathogenic versus common variant effect quantification task. All metrics represent the average test AUC and Cohen's d values calculated across three independent test sets, each defined by a distinct group of chromosomes in our nested cross-validation framework. Non-DNA foundation models are annotated with an asterisk (*). **Bolded: top 2 highest (absolute) value.**

	Model	eQTL	sQTL	paQTL	ipaQTL
AUC	AlphaGenome, output tracks*	0.8029	0.7147	0.7543	0.8644
	Sei, hidden states*	0.7561	0.6534	0.6189	0.6071
	Sei, output tracks*	0.7497	0.6276	0.6553	0.606
	Enformer, hidden states*	0.7744	0.6662	0.6737	0.6919
	Enformer, output tracks*	0.7699	0.6174	0.666	0.6587
	DNABERT-2	0.5702	0.5795	0.5066	0.4694
	NT-v2	0.6091	0.5047	0.5251	0.6019
	HyenaDNA	0.6117	0.5531	0.4699	0.448
	HyenaDNA-450K, long sequence	0.6027	0.5262	0.5521	0.5093
	Caduceus-Ph	0.6492	0.5666	0.5082	0.5678
	Caduceus-Ph, long sequence	0.6265	0.5703	0.4649	0.5203
	GROVER	0.5896	0.4742	0.4494	0.4759

Cohen's d					
	AlphaGenome, output tracks*	1.287	0.7872	0.9824	1.6347
	Sei, hidden states*	1.0335	0.553	0.4538	0.4126
	Sei, output tracks*	1.0116	0.4936	0.5503	0.4227
	Enformer, hidden states*	1.1102	0.6115	0.6028	0.6576
	Enformer, output tracks*	1.1085	0.4129	0.5457	0.5691
	DNABERT-2	0.2371	0.2825	0.024	-0.0756
	NT-v2	0.3956	-0.004	0.0658	0.3837
	HyenaDNA	0.3877	0.2018	-0.0768	-0.2048
	HyenaDNA-450K, long sequence	0.3605	0.0757	0.2068	0.0733
	Caduceus-Ph	0.5484	0.2278	0.0456	0.2324
	Caduceus-Ph, long sequence	0.4913	0.2464	-0.1151	0.075
	GROVER	0.319	-0.0978	-0.1393	-0.072

Table 10: Overall performance of DNA foundation models and other genomic models in QTL variant effect quantification tasks. All metrics represent the average AUC and Cohen's d values calculated across three independent test sets, each defined by a distinct group of chromosomes in our nested cross-validation framework. Non-DNA foundation models are annotated with an asterisk (*). **Bolded: the top two highest (absolute) performances for each task.**

3. It is unclear how many pathogenic/neutral variants are included in the benchmark dataset. Please add the count in the method section. If the set is relatively small, the authors should consider all other benchmark variant sets for the purpose of a comprehensive evaluation. For example, the authors can follow the Figure 4d of Nucleotide Transformer paper, adding the eQTL, meQTL, and Clinvar variant sets.

Response: We agree completely that providing clear sample sizes and utilizing diverse, high-quality variant sets is essential for a comprehensive evaluation.

To address the first point, we have now explicitly stated the sample sizes for all variant datasets in the revised **Methods: Variant Effect Quantification Datasets** section (page 23). For the pathogenic versus common SNP benchmark, we expanded the sample sizes and this now includes 22,222 pathogenic and 17,374 common sequences for the long-sequence task, and 22,239 pathogenic and 17,398 common sequences for the short-sequence task.

Furthermore, in complete agreement with the reviewer's suggestion to incorporate additional, functionally relevant variant sets, we have introduced a second, extensive benchmark. We have adopted the high-quality, fine-mapped causal QTL dataset, which includes precisely the types of variants the reviewer suggested. As detailed on page 23, this new benchmark evaluates models on their ability to distinguish functional QTLs from matched non-causal variants across several regulatory classes, including 1,896

expression QTLs (eQTLs), 540 splicing QTLs (sQTLs), 116 intronic-polyadenylation QTLs (ipaQTLs), and 142 polyadenylation QTLs (paQTLs).

Reviewer #3 (Remarks on code availability):

The GitHub repo is well documented. Results should be reproducible.

Response: Thank you for checking our GitHub repo and support for our study.

NCOMMS-24-51293B Benchmarking DNA Foundation Models for Genomic Sequence Classification

Response to Reviewer #2:

I appreciate that the authors have added Enformer, Sei, and AlphaGenome for the variant effect quantification benchmarking task and performed random data splitting, however, several key concerns remain.

Response: Thank you for your recognition of revisions. We have made every effort to address your remaining concerns as detailed below. We believe these valid concerns largely result from our unclear descriptions and we have further clarified them during our revisions. We sincerely hope you will find our revision satisfactory.

1. The term "genome-wide" typically means that a dataset covers most of the genome, with enough diversity and density to support broad generalizations. Although the datasets used for benchmarking in this manuscript were collected from different genomic regions, their limited size and scope mean that most cannot be considered genome-wide datasets. For example, only in human genome, ENCODE project annotated 399,124 regions with enhancer-like features, but the enhancer dataset used in this paper includes only around 15,000 sequences. For promoters, ENCODE identifies around 200,000 transcription start sites, but the authors' promoter dataset, combining both positive and negative samples, includes only about 60,000 samples. Moreover, ENCODE identified 2.9 million non-overlapping DNase I hypersensitive sites across 125 human cell lines, while the dataset used here has only about 1,000 samples.

Response: We thank the reviewer for this insightful comment and fully agree that the short sequence-classification datasets used in our benchmark are not *genome-wide* in the strict sense of exhaustive coverage. As the reviewer notes, collections such as our enhancer, promoter, and DNase I datasets represent only a small fraction of the corresponding regulatory elements cataloged by large consortia such as ENCODE. We have provided sample size, class label distribution and sequence length information in Supplementary Table 2. We have clarified this limitation in the Discussion (page 18), and now explicitly explained that these benchmarks capture curated windows around elements (say, enhancer) of interest rather than covering all available elements in the entire genome in Methods section (page 20, 21). We have also removed the genome-wide annotation in Supplementary Text 2 in our last revision, as we recognize that our previous definition of the term "genome-wide" may cause misunderstandings.

Furthermore, we would like to clarify in detail why these 57 benchmark datasets are non-exhaustive. The limited scope is a methodological choice made by the original authors constructing these datasets, to construct scientifically rigorous and meaningful classification tasks. Instead of attempting to catalog every genomic instance of an element, which would introduce noise and redundancy, the authors employed specific sampling strategies for both positive and negative sequences.

For the positive sequences, the sampling was non-exhaustive due to three main reasons. First, to prioritize high-confidence data, the sequences were intentionally sampled from high-quality, curated sources. For example, the promoter datasets from DNABERT-2 are derived from the Eukaryotic Promoter Database (EPD), which includes only experimentally validated Transcription Start Sites (TSSs), thereby filtering out a vast number of lower-confidence or computationally predicted sites (Zhou 2024, arXiv:2306.15006). Second, to reduce redundancy and bias, standard bioinformatics filtering was applied. The promoter datasets for specific human cell lines (*Promoter GM12878*, *Promoter HUVEC*, *Promoter HeLa-S3*, *Promoter NHEK*) explicitly remove redundant sequences with >80% similarity using CD-HIT, ensuring a more diverse training set (Zhang 2022, PMID: 36161334). The enhancer datasets apply even stricter filtering, with one removing all sequence pairs having more than 20% similarity to ensure the model learns generalizable patterns rather than memorizing a few homologous families (Geng 2022, PMID: 35605495). Third, to ensure fair benchmarking, some datasets are non-exhaustive because they were adopted from pre-existing, influential benchmark studies. In the case of the *DNase I* dataset, the positive set represents only a fraction of all DNase I hypersensitive sites because it provides a standardized “challenge set” for direct and fair head-to-head comparison against prior computational methods (Noble 2005, PMID: 15961476, as cited by Liu 2016, PMID: 27153623).

The sampling strategies of negative sequences are equally sophisticated, as a naive negative set drawn randomly from the genome would make the classification task trivially easy. Therefore, these datasets employ “hard negative” sampling. The splice site prediction datasets, for example, constructed a heterogeneous negative set that includes not only random exon and intron sequences but also false positive sequences containing the canonical GT/AG dinucleotide that are not true splice sites (Scalzitti 2021, PMID: 34814826; Wang 2019, PMID: 31881982). Similarly, some promoter identification datasets sample their negatives from specific genomic regions with similar characteristics, like fragments downstream of first exons, or even by explicitly selecting non-promoter sequences with the highest possible sequence similarity to the positive samples (Zhang 2022, PMID: 36161334; Grešová 2023, PMID: 37127596). To control potential confounders, datasets like *Enhancers Cohn*, *Enhancers Ensembl*, and *Open chromatin regions* sample length-matched negatives that are guaranteed to be non-overlapping with any known positive elements (Grešová 2023, PMID: 37127596). In summary, these purposeful sampling strategies are precisely what make these benchmarks valuable; they create a rigorous and reproducible test of a model's true discriminative power.

Although these datasets were curated with sound scientific reasoning, we recognize the reviewer's broader point about the limitations of their scope. To complement short sequence classification tasks, during our revisions, we have added genome-scale evaluations including Gene Expression Prediction and Variant Effect Quantification on long sequence, so that long sequence-context DNA foundation models are evaluated in settings that match their designs.

Lastly, we wish to share our original motivation of conducting this benchmarking study for DNA foundation models. Because many prior papers evaluated models on slightly different tasks and protocols, our aim was to provide a consistent, head-to-head comparison across widely used DNA foundation models. This motivation let us collect 57 sequence-classification benchmark

datasets from literature and to evaluate models under a unified zero-shot framework. We totally agree that building a more comprehensive, genome-wide benchmark would be valuable; however, such work is out of the scope of our current study, and we sincerely hope we could leave this to future work for the following two reasons. First, our original submission (with specifically focused on these 57 classification benchmarks without any newly added section, such as pre-training comparison, gene expression prediction and variant effect quantification tasks) already gained 7 citations before formal publication, indicating that these original 57 benchmarking tasks itself is meaningful for the community. Second, processing raw genomic data to create exhaustive, genome-wide benchmarks requires extensive efforts, and extensive quality control. Many of our chosen datasets, on the other hand, have become popular benchmarks used to evaluate computational methods in relevant tasks. For instance, the DNase I hypersensitive sites identification dataset (originally from Noble 2005, PMID: 15961476) served as benchmark of recent methods (Jin 2024, PMID: 38964595; Lyu 2018, PMID: 30598079). This is similar for the 5mC and 6mA identification (originally from Lv 2020, PMID: 32240948; as cited by Yu 2021, PMID: 34601568) and enhancer identification (originally from Liu 2016, PMID: 26476782; as cited by Li 2023, PMID: 37113248). Given this strong precedent, we thus focus on relying on these widely used datasets and ensure a fair head-to-head comparison among different DNA foundation models.

We sincerely hope you will find our explanations and revisions satisfactory.

2. Finally, I am still unclear on why the Enformer, Sei, and AlphaGenome models could not be included in the classification tasks. There is no need to use "N" padding, as longer input sequences can be obtained by extracting the surrounding genomic sequence. The authors created both long and short sequence versions for the expression-level and QTL prediction tasks; a similar approach should be feasible for the classification tasks as well.

Response: We thank the reviewer for this thoughtful comment and agree that, in principle, longer input sequences could be generated by extracting surrounding genomic regions. We recognize that this is a limitation of our current work, and we have clarified it in the Discussion section (page 18). We have also revised the description on Page 22 and 24 to emphasize that these 57 classification tasks are short sequence to let readers clearly see our rationale and limitations. Our decision not to implement this approach was based on the fundamental incompatibility between the flexible input length requirements of DNA foundation models and the fixed length input requirements of Enformer, Sei, and AlphaGenome, which creates several practical and conceptual constraints.

First, as we detailed in our response to Comment 1, these sequence classification datasets were curated to distinguish local patterns within short, fixed windows against carefully matched, non-overlapping negatives. Extending these sequences would alter the nature of the biological question being asked. Using classifying promoter as an example, if we expand each sequence length (e.g., 196kb to match Enformer and 4096 to match Sei contexts), positives may now encompass promoters plus nearby exons, introns, and enhancers from multiple genes;

negatives are much harder to guarantee remain free of promoters and matched sequence length, GC content, etc. Additionally, the fixed length input requirements of Enformer, Sei, and AlphaGenome made it impractical to include only promoter regions since the length of promoters vary across genes. This practical concern makes it challenging to extending the sequence length directly.

Second, several datasets consist of experimentally curated sequences with fixed lengths that are part of the original assay design (for example, 41 bp methylation windows in Jin 2022 and 251 bp promoter regions in Grešová 2023). Such curated windows are widely used and accepted as standard benchmarks for evaluating DNA foundation models because their predefined lengths and biological contexts allow direct comparison across studies. Preserving this design ensures fairness and consistency with prior work, but it also inevitably limits the scope of our study.

Lastly, most of these datasets are distributed without the original genomic coordinates, which makes reliable reconstruction of longer sequences challenging. This differs from our expression-level and QTL tasks, which we built directly from primary genomic coordinates and where we could freely generate both short and long sequence versions. While the practical issue makes it hard for us to reconstruct these benchmarking datasets, we believe curating these widely used benchmarking tasks used in the community itself is needed as researchers can now obtain curated datasets through our repo instead of downloading from many places.

That said, we fully acknowledge that this decision limits the inclusion of models such as Enformer, Sei, and AlphaGenome. We view this as an important limitation and an opportunity for future work, where benchmark designs that naturally accommodate long-context models can be developed without losing biological relevance or comparability.

Response to Reviewer #3:

All my comments were properly addressed by the author. I have no further comments. I commend the authors for their good work.

Reviewer #3 (Remarks on code availability):

The GitHub repo is well documented. Results should be reproducible.

Response: Thank you so much for supporting our work and careful review. We truly appreciate it.